# On the Power of Federated Learning for Online Sparse Linear Regression with Decentralized Data

## Abstract

In this paper, we study the necessity of federated learning (FL) for online linear regression with decentralized data. Previous work proved that FL is unnecessary for minimizing regret in full information setting, while we prove that it can be necessary if only limited attributes of each instance are observed. We call this problem online sparse linear regression with decentralized data (OSLR-DecD). We propose a federated algorithm for OSLR-DecD, and prove a lower bound on the regret of any noncooperative algorithm. In the case of $d = o(M)$, the upper bound on the regret of our algorithm is smaller than the lower bound, demonstrating the necessity of FL, in which $M$ is the number of clients and $d$ is the dimension of data. When $M = 1$, we give the first lower bound on the regret and improve previous upper bounds. We invent three new techniques including an any-time federated online mirror descent with negative entropy regularization, a paradigm for client-server collaboration with privacy protection, and a reduction from online sparse linear regression to prediction with limited advice for establishing the lower bound on the regret, some of which might be of independent interest.

## 1 Introduction

For many real-world applications involving sequential decision making, such as online recommendations (Zhou et al., 2021), real-time environmental protection (Hong & Chae, 2022), mobile keyboard prediction (Hard et al., 2018; Ramaswamy et al., 2019), the data is generated from geographically dispersed edge devices (Gogineni et al., 2023; Patel et al., 2023), such as phones, tablets, sensors and so on. Due to the privacy constraint, the data must be stored on local devices and can not be shared with others (McMahan et al., 2017; Kairouz et al., 2021; Li et al., 2023). The decentralized essence of the data motivates us to formulate these sequential decision problems as online learning with decentralized data. Assuming that there are $M$ clients such as phones, tablets or sensors. At each round $t = 1, 2, ..., T$ for each $j = 1, ..., M$, the $j$-th client receives an instance $\mathbf{x}_t^{(j)}$. Then a learner chooses a linear hypothesis $f_t^j$ and outputs a prediction $\hat{y}_t^{(j)} = f_t^j(\mathbf{x}_t^{(j)})$. After that the learner receives the true output $y_t^{(j)}$, and suffers a loss $\ell(\hat{y}_t^{(j)}, y_t^{(j)})$. During the $T$ rounds of interaction, the learner aims to design an algorithm that can minimize the following regret,

$$\forall \mathbf{w} \in \mathbb{R}^d, \quad \text{Reg}(\mathbf{w}) = \sum_{j=1}^{M} \sum_{t=1}^{T} \left[ \ell\left( \hat{y}_t^{(j)}, y_t^{(j)} \right) - \ell\left( \mathbf{w}^\top \mathbf{x}_t^{(j)}, y_t^{(j)} \right) \right].$$

In this paper, we consider online linear regression with decentralized data (OLR-DecD), in which $\ell(u, v) = (u - v)^2$ is the square loss function.

A natural approach to solving OLR-DecD is to independently run a copy of an online learning algorithm on the $M$ clients, such as online gradient descent (Zinkevich, 2003). Such a noncooperative algorithm naturally protects the privacy, but suffers a regret increasing linearly with $M$. Another approach is *federated learning* where clients coordinate with a server to learn models without the need to share data. Federated learning has been used to train deep neural networks from decentralized data (McMahan et al., 2017; Karimireddy et al., 2020; Woodworth et al., 2020; Reddi et al., 2021; Wu et al., 2022). However, a pessimistic result is that, the collaboration among clients is actually

unnecessary for OLR-DecD in full information setting (Patel et al., 2023). In other words, federated learning is unnecessary. An intuitive explanation for the pessimistic result is that, federated learning can not provide more information on gradients for the noncooperative algorithm.

Patel et al. (2023) also demonstrated that federated learning may be necessary for OLR-DecD in partial information setting, such as bandit feedback, motivating us considering the problem of online linear regression with limited access to attributes, also called online spare linear regression (OSLR) (Kale, 2014; Foster et al., 2016; Kale et al., 2017; Ito et al., 2017; 2018), in which the learner can only use $b, b < d$, attributes of $\mathbf{x}_t$ at most to make a prediction. OSLR naturally accommodates many real-world constraints on online learning problems, such as capital, labor, computational resource, privacy, and so forth (Cesa-Bianchi et al., 2010; Hazan & Koren, 2012; Jain et al., 2012; Zolghadr et al., 2013). For example, in the task of medical diagnosis of a disease (Cesa-Bianchi et al., 2010), $\mathbf{x}_t$ should contain the results of a large number of medical tests. However, many patients can only pay the cost for several medical tests. Thus $\mathbf{x}_t$ must be sparse. We consider online sparse linear regression with decentralized data (OSLR-DecD). To be specific, at each round $t = 1, 2, ...$, the learner can only use $b$ attributes of $\mathbf{x}_t^{(j)}$ to make the prediction $\hat{y}_t^{(j)}$, $j = 1, ..., M$. After receiving $y_t^{(j)}$, the learner can additionally observe $(b' - b)$ attributes where $b' < d$ (Foster et al., 2016; Kale et al., 2017; Ito et al., 2017). We usually compare the cumulative losses of the learner with any $b$-sparse competitor with bounded norm, and define the regret as follows,

$$\forall \mathbf{w} \in \mathcal{W}_b, \quad \text{Reg}(\mathbf{w}) = \sum_{j=1}^{M} \sum_{t=1}^{T} \left[ \left( \hat{y}_t^{(j)} - y_t^{(j)} \right)^2 - \left( \mathbf{w}^\top \mathbf{x}_t^{(j)} - y_t^{(j)} \right)^2 \right], \tag{1}$$

where $\mathcal{W}_b = \left\{ \mathbf{w} \in \mathbb{R}^d : \|\mathbf{w}\|_0 \le b, \|\mathbf{w}\|_2 \le U \right\}$ and $U$ is a constant. It is natural to ask: *Is federated learning necessary for OSLR-DecD?* To answer the question, it is imperative to prove that there is a federated algorithm whose regret is smaller than that of any noncooperative algorithm.

In this paper, we will answer the question affirmatively. To be specific, we first propose a better algorithm for OSLR, which we called AMRO. Within federated learning framework, we propose a federated AMRO for OSLR-DecD, called FedAMRO, and prove a lower bound on the regret of any noncooperative algorithm. In the case of $d = o(M)$, the upper bound of our federated algorithm is smaller than the lower bound, thereby answering the question affirmatively.

## 1.1 MAIN RESULTS

Our main results are summarized as follows.

(1) **Upper bound on the regret of FedAMRO.** For OSLR-DecD, the expected regret of FedAMRO satisfies

$$\forall \mathbf{w} \in \mathcal{W}_b, \quad \mathbb{E}\left[\text{Reg}(\mathbf{w})\right] = \tilde{O}\left( M\sqrt{T \ln N} + \frac{d-b}{b'-b}\sqrt{MT \ln N} \right), \tag{2}$$

in which $N = \binom{b}{d}$. The dominated term is the second one that only depends on $\sqrt{M}$. Importantly, FedAMRO requires no prior knowledge of the time horizon $T$.

(2) **Lower bound on the regret of any noncooperative algorithm.** In the case of $b = 1$, we prove a $\Omega(M\sqrt{dT/b'})$ lower bound on the regret of any noncooperative algorithm. Compared with the upper bound in (2), in the case of $d = o(M)$, the upper bound is smaller than the lower bound, demonstrating that **federated learning is indeed necessary for OSLR-DecD.**

(3) **Lower and Upper bound on the regret for OSLR.** For OSLR, i.e., $M = 1$, we give the first $\Omega(\sqrt{dT/b'})$ lower bound on the regret. Besides, AMRO achieves an expected regret as follows

$$\forall \mathbf{w} \in \mathcal{W}_b, \quad \mathbb{E}[\text{Reg}(\mathbf{w})] = \tilde{O}\left( \frac{d-b}{b'-b}\sqrt{T \ln N} \right).$$

There is only a gap of order $\tilde{O}(\sqrt{(d-b)/(b'-b) \ln N})$ between the upper bound and the lower bound. Without additional assumptions, the best regret bound for OSLR up to now is $\mathbb{E}[\text{Reg}(\mathbf{w})] = O\left( \frac{d^2}{(b'-b)^2}\sqrt{T \ln N} \right)$ (Foster et al., 2016). AMRO improves the previous regret bound by nearly a factor of $O(\frac{d}{b'-b})$. Besides, AMRO also requires no prior value of $T$.

(4) **Communication Complexity and Computational Complexity of FedAMRO.** The per-round communication complexity of FedAMRO is $O(Md^2)$ bits. The space complexity and per-round time complexity of FedAMRO on each client is only $O(d^2)$, while it is $O(M \cdot N)$ on server.

## 1.2 TECHNICAL CONTRIBUTIONS

In designing AMRO and FedAMRO, and in proving the lower bound, we propose three new techniques, some of which might be of independent interest.

The first one is an any-time online mirror descent (OMD) with negative entropy regularization. AMRO and FedAMRO use the new OMD to update sampling probability. To be specific, we use time-variant decision sets and learning rates in the OMD, thereby eliminating the need on the prior value of $T$. Our technique is more elegant than the standard doubling trick. The regret analysis is also more technical than the well-known OMD with constant decision sets (Bubeck & Cesa-Bianchi, 2012). The time-variant decision sets and regret analysis are also suitable for other variants of OMD, such as optimistic OMD (Rakhlin & Sridharan, 2013).

The second one is a new paradigm for cooperating among clients and server. Most of federated learning algorithms transmit estimators of gradient from clients to server, while our algorithm transmits statistics constructed from estimators of instance (the representation of each example). The new information not only makes communication efficient, but also safeguards against data leakage.

The third one is a non-trivial reduction from OSLR to prediction with limited advice (Seldin et al., 2014) for establishing the first lower bound on the regret. By this lower bound, we further establish the lower bound on the regret of any noncooperative algorithm for OSLR-DecD.

## 2 RELATED WORK

Most of previous work proposed federated learning algorithms for online learning with decentralized data, but without demonstrating their necessity, such as federated online mirror descent (Mitra et al., 2021), communication-efficient federated online gradient descent (Gogineni et al., 2023; Kwon et al., 2023), decentralized federated online learning algorithm (Odeyomi, 2023), communication-efficient federated online multi-kernel learning algorithms (Shen et al., 2021; Ghari & Shen, 2022; Hong & Chae, 2022). For distributed bandit convex optimization, Patel et al. (2023) gave an incomplete answer on the necessity of federated learning, since they did not prove a lower bound on the regret of any noncooperative algorithm. A recent study on online model selection with decentralized data (OMS-DecD) demonstrated that collaboration is essential when clients face computational constraints (Li et al., 2024). Although a federated online mirror descent has been proposed, it necessitates the prior value of $T$. We propose a new federated online mirror descent which does not require the prior value of $T$. The algorithms and results for OMS-DecD are not suitable for OSLR-DecD. The problem that whether federated learning is necessary for OSLR-DecD is still open.

Another related but essentially different problem is distributed online convex optimization that aims to minimize $\text{Reg}^{(i)}(\mathbf{w}) = \sum_{j=1}^{M} \sum_{t=1}^{T} \left[ \ell\left(f_t^{(i)}(\mathbf{x}_t^{(j)}), y_t^{(j)}\right) - \ell\left(\mathbf{w}^\top \mathbf{x}_t^{(j)}, y_t^{(j)}\right) \right]$ for all $i \in [M]$ (Yan et al., 2013; Hosseini et al., 2013; Wan et al., 2024). However, such a goal is not suitable for our problem as the decentralized nature of data and the privacy constraint. To be specific, it is infeasible to evaluate $f_t^{(i)}$ on the other clients as the data from other clients are not visible for $f_t^{(i)}$, thereby computing $\sum_{j=1}^{M} \sum_{t=1}^{T} \ell\left(f_t^{(i)}(\mathbf{x}_t^{(j)}), y_t^{(j)}\right)$ is unfeasible. Besides, the ultimate goal of online learning is to minimize the cumulative losses suffered along its run (Shalev-Shwartz, 2012), i.e., $\sum_{j=1}^{M} \sum_{t=1}^{T} \ell\left(\hat{y}_t^{(j)}, y_t^{(j)}\right)$. Although exploring the necessity of collaboration may be trivial in distributed online convex optimization, but is indeed a non-trivial problem in OLR-DecD.

## 3 PRELIMINARIES AND PROBLEM SETTING

**Notations** Let $\{(\mathbf{x}_t, y_t)\}_{t \in [T]}$ be a sequence of examples, where $\mathbf{x}_t \in \mathcal{X} = \{\mathbf{x} \in \mathbb{R}^d : \|\mathbf{x}\|_2 < +\infty\}$ is an instance, $|y_t| \leq Y$ is the output, and $[T] = \{1, \ldots, T\}$. For any vector $\mathbf{a} \in \mathbb{R}^d$, and any $S = \{s_1, s_2, ..., s_b\} \subset [d]$, we define $\mathbf{a}[S] := (a_{s_1}, a_{s_2}, ..., a_{s_b})$. For any matrix $\mathbf{A} \in \mathbb{R}^{d \times d}$,

$\mathbf{A}[S, S]$ is a sub-matrix formed by the rows and columns indexed by $S$. Let $N = \binom{b}{d}$ and $\Delta_N$ be the $N - 1$ dimensional simplex. For a convex set $\Omega$ and a convex function $\psi$ defined on $\Omega$, the Bregman divergence denoted by $\mathcal{B}_\psi(\cdot, \cdot)$, is defined as follows

$$\forall \mathbf{u}, \mathbf{v} \in \Omega, \quad \mathcal{B}_\psi(\mathbf{u}, \mathbf{v}) = \psi(\mathbf{u}) - \psi(\mathbf{v}) - \langle \nabla \psi(\mathbf{v}), \mathbf{u} - \mathbf{v} \rangle.$$

Next we introduce the pessimistic result that federated learning is unnecessary for OLR-DecD in the full information setting.

**Theorem 1** (Informal result from Theorem 3.2 in Patel et al. (2023)). *For distributed online convex optimization (including online learning with decentralized data) with smooth losses functions (including the square loss function), noncooperative online gradient descent is optimal.*

To avoid the pessimistic result, we consider a type of partial information setting where only limited attributes of each instance can be observed. We call this problem OSLR-DecD defined as follows. Assuming that there are $M$ clients. At any round $t$, each client $j \in [M]$ receives an instance $\mathbf{x}_t^{(j)}$. The learner selects a subset of attributes denoted by $\mathbf{x}_t^{(j)}[S_t^{(j)}]$ where $S_t^{(j)} \subset [d]$ and $|S_t^{(j)}| \leq b$, and makes a prediction $\hat{y}_t^{(j)} = f_t^{(j)}(\mathbf{x}_t^{(j)}[S_t^{(j)}])$ where $f_t^{(j)}$ is a linear hypothesis. After that, the learner observes $y_t^{(j)}$ and additional $(b' - b)$ attributes of $\mathbf{x}_t^{(j)}$. The loss suffered by the learner is $(\hat{y}_t^{(j)} - y_t^{(j)})^2$. Since the learner only uses $b$ attributes to make a prediction, we hope to develop an algorithm that can compete with any $b$-sparse competitor, and define the regret in (1). Due to the privacy constraint, clients can not transfer their raw data, but can share models, gradients or other information without leakaging raw data. In this case of $M = 1$, OSLR-DecD is degenerated to the well-defined OSLR (Foster et al., 2016; Kale et al., 2017).

## 4 A Better Algorithm for OSLR

In this section, we propose an algorithm with a better regret bound for OSLR, which forms the foundation of our federated algorithm in the following section. We first give a high-level explanation on the algorithm. Recalling that we aim to learn the optimal parameter $\mathbf{w}^*$ satisfying $\|\mathbf{w}^*\|_0 \leq b$. An intuitive approach is to partition $[d]$ into $N$ disjoint subsets denoted by $S_i, i \in [N]$, in which $|S_i| = b$ and each element in $S_i$ indexes the corresponding attributes of $\mathbf{x}_t$. Then our algorithm simultaneously learns the support set of $\mathbf{w}^*$ and $\mathbf{w}^*$. To this end, our algorithm will learn a probability distribution over the $N$ subsets, and the optimal hypothesis $f_i^*$ parameterized by $\mathbf{w}_i^*$ with support set $S_i, i \in [N]$. It is obvious that $\mathbf{w}^* \in \{\mathbf{w}_1^*, \ldots, \mathbf{w}_N^*\}$. Next we detail the algorithm.

At round $t$, our algorithm maintains $N$ hypotheses $f_{t,1}, f_{t,2}, \ldots, f_{t,N}$ and a probability distribution $\mathbf{p}_t \in \Delta_N$. For each $i \in [N]$, $f_{t,i}$ is parameterized by $\mathbf{w}_{t,i} \in \mathbb{R}^b$ and uses the attributes indexed by $S_i$ to make a prediction, $f_{t,i}(\mathbf{x}_t) = \langle \mathbf{w}_{t,i}, \mathbf{x}_t[S_i] \rangle$. Our algorithm randomly selects a hypothesis $f_{t,I_t}$ following $\mathbf{p}_t$, observes the attributes indexed by $S_{I_t}$, and outputs $f_{t,I_t}(\mathbf{x}_t[S_{I_t}])$. Since the attributes indexed by $S_i, i \neq I_t$, can not be fully observed, both $\ell(f_{t,i}(\mathbf{x}_t[S_i]))$ and $\nabla_{t,i} := \nabla \ell(f_{t,i}(\mathbf{x}_t[S_i]))$ are unknown. It is necessary to estimate the loss and the gradient, both with low variances.

In order to update $\mathbf{p}_t$, we define a new loss $c_{t,i}$ for evaluating the prediction performance of $f_{t,i}$. Unfolding the square loss function, $c_{t,i}$ and $\nabla_{t,i}$ can be rewritten as follows

$$c_{t,i} := \ell\left(f_{t,i}(\mathbf{x}_t[S_i]), y_t\right) - y_t^2 = \mathbf{w}_{t,i}^\top \left(\mathbf{x}_t \mathbf{x}_t^\top [S_i, S_i]\right) \mathbf{w}_{t,i} - 2y_t \mathbf{w}_{t,i}^\top \mathbf{x}_t[S_i], \tag{3}$$

$$\nabla_{t,i} = 2\left(\mathbf{x}_t \mathbf{x}_t^\top [S_i, S_i]\right) \mathbf{w}_{t,i} - 2y_t \mathbf{x}_t[S_i],$$

in which we can estimate $\mathbf{x}_t \mathbf{x}_t^\top$ and $2y_t \mathbf{x}_t$ by constructing two independent estimators of $\mathbf{x}_t$, denoted by $\tilde{\mathbf{x}}_t$ and $\hat{\mathbf{x}}_t$. To this end, our algorithm samples $\frac{b'-b}{2}$ elements from $[d] \setminus S_{I_t}$ with replacement. We denote this sampled set $\tilde{B}_t$. Our algorithm then repeats the same sampling process to obtain another set, $\hat{B}_t$. Then $\tilde{\mathbf{x}}_t$ and $\hat{\mathbf{x}}_t$ are defined as follows

$$\begin{cases} \tilde{x}_{t,m} = \hat{x}_{t,m} = x_{t,m}, \forall m \in S_{I_t}, \\ \tilde{x}_{t,m} = \dfrac{x_{t,m} - \delta_{t,m}}{\mathbb{P}[m \in \tilde{B}_t]} \cdot \mathbb{I}_{m \in \tilde{B}_t} + \delta_{t,m}, \forall m \notin S_{I_t}, \\ \hat{x}_{t,m} = \dfrac{x_{t,m} - \delta_{t,m}}{\mathbb{P}[m \in \hat{B}_t]} \cdot \mathbb{I}_{m \in \hat{B}_t} + \delta_{t,m}, \forall m \notin S_{I_t}, \end{cases} \tag{4}$$

in which $\delta_{t,m} \in \mathbb{R}$ will be defined later. Both $\tilde{\mathbf{x}}_t$ and $\hat{\mathbf{x}}_t$ are unbiased estimators of $\mathbf{x}_t$. Besides, $\tilde{\mathbf{x}}_t$ is independent of $\hat{\mathbf{x}}_t$, given $\delta_t \in \mathbb{R}^d$. Now the estimators of $c_{t,i}$ and $\nabla_{t,i}$ are defined by

$$
\begin{cases}
\tilde{c}_{t,i} = \mathbf{w}_{t,i}^{\top} \left( \tilde{\mathbf{x}}_t \hat{\mathbf{x}}_t^{\top} [S_i, S_i] \right) \mathbf{w}_{t,i} - y_t \mathbf{w}_{t,i}^{\top} (\tilde{\mathbf{x}}_t + \hat{\mathbf{x}}_t)[S_i], \\
\tilde{\nabla}_{t,i} = 2 \left( \tilde{\mathbf{x}}_t \hat{\mathbf{x}}_t^{\top} [S_i, S_i] \right) \mathbf{w}_{t,i} - y_t (\tilde{\mathbf{x}}_t + \hat{\mathbf{x}}_t)[S_i].
\end{cases}
\tag{5}
$$

Our algorithm updates the probability and the hypothesis following OMD framework,

$$
\begin{cases}
\mathbf{p}_{t+1} = \underset{\mathbf{p} \in \Delta_{t,N}}{\arg\min} \langle \tilde{\mathbf{c}}_t, \mathbf{p} \rangle + \mathcal{B}_{\psi_t}(\mathbf{p}, \mathbf{p}_t), \qquad \Delta_{t,N} = \left\{ \mathbf{p} \in \Delta_N : p_i \geq \frac{\beta_t}{N}, i \in [N] \right\}, \\[2mm]
\mathbf{w}_{t+1,i} = \underset{\mathbf{w} \in \mathcal{W}}{\arg\min} \left\langle \tilde{\nabla}_{t,i}, \mathbf{w} \right\rangle + \mathcal{B}_{\psi_{t,i}}(\mathbf{w}, \mathbf{w}_{t,i}), \qquad \mathcal{W} = \left\{ \mathbf{w} \in \mathbb{R}^b : \|\mathbf{w}\|_2 \leq U \right\}, \\[2mm]
\psi_t(\mathbf{p}) = \sum_{i=1}^{N} \frac{1}{\eta_t} p_i \ln p_i, \qquad\qquad\qquad \psi_{t,i}(\mathbf{w}) = \frac{1}{2\lambda_{t,i}} \|\mathbf{w}\|_2^2, \quad i = 1, ..., N,
\end{cases}
\tag{6}
$$

in which $\beta_t \in (0,1]$ gives a lower bound on the sampling probability. We use the time-varying decision subset $\Delta_{t,N}$ in OMD for the first time, making it possible to use a time-variant learning rate $\eta_t$ and eliminate the need for the prior information of $T$. By the Lagrangian multiplier method, $\mathbf{w}_{t+1,i}$ enjoys a closed form solution as follows,

$$
\mathbf{w}_{t+1,i} = \frac{U}{\max\left\{ U, \|\mathbf{w}_{t,i} - \lambda_{t,i} \tilde{\nabla}_{t,i}\|_2 \right\}} \cdot \left( \mathbf{w}_{t,i} - \lambda_{t,i} \tilde{\nabla}_{t,i} \right).
\tag{7}
$$

However, the time-varying decision subsets make it difficult to solve $\mathbf{p}_{t+1}$. To be specific, $\mathbf{p}_{t+1}$ does not enjoy a closed form. We propose Algorithm 1 for solving $\mathbf{p}_{t+1}$. Due to space limitations, the details of Algorithm 1 are given in the supplementary materials.

---

**Algorithm 1** Solving $\mathbf{p}_{t+1}$

**Input:** $\tilde{\mathbf{c}}_t, \eta_t, \beta_t$
**Initialization:** $\mathcal{A} = \emptyset$
1: $\forall i \in [N], p_{t+1,i} = \frac{p_{t,i} \exp(-\tilde{c}_{t,i} \eta_t)}{\sum_{j=1}^{N} p_{t,j} \exp(-\tilde{c}_{t,j} \eta_t)}$
2: $\mathcal{A} = \{i \in [N], p_{t+1,i} < \frac{\beta_t}{N}\}$ and $\mathcal{A}^{\neg} = [N] \setminus \mathcal{A}$
3: **while** $\mathcal{A} \neq \emptyset$ **do**
4:     $z_t = \frac{\beta_t \cdot \sum_{j \in \mathcal{A}^{\neg}} p_{t,j} \exp(-\tilde{c}_{t,j} \eta_t)}{(N - |\mathcal{A}| \cdot \beta_t)}$
5:     For each $i \in \mathcal{A}$, $p_{t+1,i} = \frac{\beta_t}{N}$
6:     $flag = 0$
7:     **for** $i \in \mathcal{A}^{\neg}$ **do**
8:       $p_{t+1,i} = \frac{p_{t,i} \exp(-\tilde{c}_{t,i} \eta_t)}{\sum_{j \in \mathcal{A}^{\neg}} p_{t,j} \exp(-\tilde{c}_{t,j} \eta_t) + |\mathcal{A}| \cdot z_t}$
9:       **if** $p_{t+1,i} < \frac{\beta_t}{N}$ **then**
10:        $\mathcal{A} = \mathcal{A} \cup \{i\}$
11:        $flag = flag + 1$
12:       **end if**
13:     **end for**
14:     $\mathcal{A}^{\neg} = [N] \setminus \mathcal{A}$
15:     **if** $flag == 0$, **then** $\mathcal{A} = \emptyset$
16: **end while**
17: Return $\mathbf{p}_{t+1}$

**Algorithm 2** AMRO

**Input:** $U, b, b', N = \binom{b}{d}$
**Initialization:** $f_{1,i} = 0, p_{1,i} = \frac{1}{N}, \forall i \in [N]$
1: Divide $[d]$ into $N$ subsets of size $b$, denoted by $S_1, \ldots, S_N$
2: **for** $t = 1, \ldots, T$ **do**
3:     Sample $I_t \in [N]$ following $\mathbf{p}_t$
4:     Output the prediction $\langle \mathbf{w}_{t,I_t}, \mathbf{x}_t[S_{I_t}] \rangle$
5:     Sample $\tilde{B}_t, \hat{B}_t$ from $[d] \setminus S_{I_t}$ independently
6:     Construct $\tilde{\mathbf{x}}_t$ and $\hat{\mathbf{x}}_t$ following (4)
7:     **for** $i \in [N]$ **do**
8:       Construct $\tilde{c}_{t,i}$ and $\tilde{\nabla}_{t,i}$ following (5)
9:     **end for**
10:    Solve $\mathbf{p}_{t+1}$ by Algorithm 1
11:    Compute $\{\mathbf{w}_{t+1,i}\}_{i=1}^{N}$ following (7)
12: **end for**

---

We name this algorithm AMRO (Aggregating online Mirror descent and Resampling for OSLR) and show the pseudo-code in Algorithm 2. Next we give more explanations on AMRO, and compare it with previous algorithms.

(i) In (3), we define $c_{t,i} = \ell\left(f_{t,i}(\mathbf{x}_t[S_i]), y_t\right) - y_t^2$, while the algorithm in Foster et al. (2016) defines $c_{t,i} = \ell\left(f_{t,i}(\mathbf{x}_t[S_i]), y_t\right)$. Subtracting $y_t^2$ from $\ell\left(f_{t,i}(\mathbf{x}_t[S_i]), y_t\right)$ avoids transmitting $y_t^2$ to server in our federated algorithm for privacy protection.

(ii) In (4), previous algorithms use $\delta_{t,m} = 0$ (Foster et al., 2016; Kale et al., 2017; Ito et al., 2017). Our estimators exhibit low variance if $\delta_{t,m}$ provides a good estimate of $x_{t,m}$, as demonstrated in Lemma 1. The estimators will be used in our federated algorithm for privacy protection.

**Lemma 1.** *For all $i \in [N]$ and $\mathbf{x} \in \mathcal{X}$, suppose $\|\mathbf{x}[S_i]\|_2^2 \leq X^2$. For all $t \in [T]$ and $i \in [N]$, let*

$$\|\delta_t[S_i]\|_2^2 \leq X^2, \quad \Xi_{t,i} := \frac{2d - b' - b}{b' - b} \|\mathbf{x}_t[S_i] - \delta_t[S_i]\|_2^2. \tag{8}$$

*Then*

$$\mathbb{E}\left[\tilde{c}_{t,i}^2\right] \leq U^4 \Xi_{t,i}^2 + 2U^2(UX + Y)^2 \Xi_{t,i} + c_{t,i}^2,$$

$$\mathbb{E}\left[\left\|\tilde{\nabla}_{t,i}\right\|_2^2\right] \leq 4U^2 \Xi_{t,i}^2 + 8(UX + Y)^2(\Xi_{t,i} + X^2).$$

*The expectation is taken on the elements in $\tilde{B}_t$ and $\hat{B}_t$.*

There are many approaches to define $\delta_{t,m}$, such as

$$\delta_{t,m} = \frac{\sum_{\tau < t, m \in S_{I_\tau} \cup \tilde{B}_\tau \cup \hat{B}_\tau} x_{\tau,m}}{|\{\tau < t : m \in S_{I_\tau} \cup \tilde{B}_\tau \cup \hat{B}_\tau\}|}. \tag{9}$$

Our estimators can adapt to certain benign environments. If $\delta_{t,m}$ provides a good estimate of $x_{t,m}$ for all $m \in [d]$, such as $(x_{t,m} - \delta_{t,m})^2 = O(\frac{b'-b}{2d-b'-b} x_{t,m}^2)$, then the second-order moments of $\tilde{c}_{t,i}$ and $\hat{\nabla}_{t,i}$ are $O(U^4 X^4)$. In the worst case, that is, $\delta_{t,m}$ is not a good estimator of $x_{t,m}$, the second-order moments are $O(\frac{(d-b)^2}{(b'-b)^2} U^4 X^4)$.

## 5 A FEDERATED AMRO FOR OSLR-DECD

In this section, we propose a federated AMRO within the federated learning framework (McMahan et al., 2017), which is highly non-trivial due to the following two challenges.

The first one is making communication efficient. For each $j \in [M]$, if the client maintains $\mathbf{p}_t^{(j)}$ and $f_{t,1}^{(j)}, \ldots, f_{t,N}^{(j)}$, then the client must transmit estimators of gradient and loss to server and receive a global probability distribution and global models from server. The communication cost is $O(MN)$ bits. It is necessary to limit the communication cost to $O(M \cdot \mathrm{poly}(d))$ bits.

The second one is protecting the raw data of clients. Due to the communication constraint, it is challenging to avoid transmitting estimators of gradient and loss to server and avoiding privacy leakage simultaneously.

Next we give a high-level explanation on the algorithm. We propose two techniques for addressing the challenges. The first one is decoupling prediction and updating. Specifically, clients make predictions, while server aggregates information from clients and updates probability distributions and hypotheses. Our algorithm independently samples a hypothesis for each client, and only sends the selected hypotheses to clients, thereby achieving efficient communication. The second one is a new paradigm for cooperating among clients and server. To be specific, we construct novel statistics (not estimators of loss and gradient) that will be sent by clients. Benefit from our estimators of $\mathbf{x}_t$ in (4), we can enhance the privacy by incorporating random noises into the estimators.

### 5.1 PREDICTION ON CLIENTS

At each round $t$, our algorithm stores a probability distribution $\mathbf{p}_t$ on server. For each $j \in [M]$, server samples $I_t^{(j)} \in [N]$ following $\mathbf{p}_t$, and sends $\mathbf{w}_{t,I_t^{(j)}}$ and $S_{I_t^{(j)}}$ to the client. On the client, our algorithm receives $\mathbf{x}_t^{(j)}$, selects the attributes $\mathbf{x}_t^{(j)}[S_{I_t^{(j)}}]$ and outputs a prediction $\hat{y}_t^{(j)}$ defined by

$$\hat{y}_t^{(j)} = f_{t,I_t^{(j)}}\left(\mathbf{x}_t^{(j)}[S_{I_t^{(j)}}]\right) = \left\langle \mathbf{w}_{t,I_t^{(j)}}, \mathbf{x}_t^{(j)}[S_{I_t^{(j)}}]\right\rangle. \tag{10}$$

Following AMRO, our algorithm independently samples $\tilde{B}_t^{(j)}, \tilde{B}_t^{(j)}$ from $[N] \setminus S_{I_t^{(j)}}$, and observes $\mathbf{x}_t^{(j)}[\tilde{B}_t^{(j)}], \mathbf{x}_t^{(j)}[\hat{B}_t^{(j)}]$. Let $\mathrm{Ber}(\sigma^{(j)})$ be a Bernoulli distribution which outputs 1 with probability $\sigma^{(j)} \in (0, 1)$. For each $m \in S_{I_t^{(j)}}$, our algorithm independently samples $\tilde{\nu}_{t,m}^{(j)}$ and $\hat{\nu}_{t,m}^{(j)}$ following

$\text{Ber}(\sigma^{(j)})$. Due to the privacy constraint, two estimators of $\mathbf{x}_t^{(j)}$ denoted by $\tilde{\mathbf{x}}_t^{(j)}$ and $\hat{\mathbf{x}}_t^{(j)}$ that are slightly different from the estimators in (4), are defined as follows.

$$\forall m \in S_{I_t^{(j)}}, \quad \tilde{x}_{t,m}^{(j)} = \frac{x_{t,m}^{(j)} - \delta_{t,m}^{(j)}}{\mathbb{P}[\tilde{\nu}_{t,m}^{(j)} = 1]} \cdot \mathbb{I}_{\tilde{\nu}_{t,m}^{(j)}=1} + \delta_{t,m}^{(j)}, \qquad \hat{x}_{t,m}^{(j)} = \frac{x_{t,m}^{(j)} - \delta_{t,m}^{(j)}}{\mathbb{P}[\hat{\nu}_{t,m}^{(j)} = 1]} \cdot \mathbb{I}_{\hat{\nu}_{t,m}^{(j)}=1} + \delta_{t,m}^{(j)},$$

$$\forall m \notin S_{I_t^{(j)}}, \quad \tilde{x}_{t,m}^{(j)} = \frac{x_{t,m}^{(j)} - \delta_{t,m}^{(j)}}{\mathbb{P}[m \in \tilde{B}_t^{(j)}]} \cdot \mathbb{I}_{m \in \tilde{B}_t^{(j)}} + \delta_{t,m}^{(j)}, \quad \hat{x}_{t,m}^{(j)} = \frac{x_{t,m}^{(j)} - \delta_{t,m}^{(j)}}{\mathbb{P}[m \in \hat{B}_t^{(j)}]} \cdot \mathbb{I}_{m \in \hat{B}_t^{(j)}} + \delta_{t,m}^{(j)}.$$

The client only sends $\tilde{\mathbf{x}}_t^{(j)}(\hat{\mathbf{x}}_t^{(j)})^\top$ and $y_t^{(j)}(\hat{\mathbf{x}}_t^{(j)} + \tilde{\mathbf{x}}_t^{(j)})$ to server. Benefiting from the definition of $c_{t,i}$ in (3), the client does not transmit $(y_t^{(j)})^2$ to server. Note that it is hard to reconstruct $y_t^{(j)}$ from $y_t^{(j)}(\hat{\mathbf{x}}_t^{(j)} + \tilde{\mathbf{x}}_t^{(j)})$ as our algorithm does not transmit $\hat{\mathbf{x}}_t^{(j)} + \tilde{\mathbf{x}}_t^{(j)}$. $\delta_t^{(j)}$ serves as random noises added on $\mathbf{x}_t^{(j)}$, making it hard to recovery $(\mathbf{x}_t^{(j)}, y_t^{(j)})$ from the information sent by the client, thereby protecting the privacy. The total communication cost is only $O(Md^2)$ bits. Thus our algorithm solves the two challenges. Besides, the computational complexity on each client is only $O(d^2)$.

## 5.2 FEDERATED UPDATING ON SERVER

After receiving the information from clients, it is easy to construct estimator of loss, denoted by $\tilde{\mathbf{c}}_t^{(j)}$, and estimator of gradient, denoted by $\tilde{\nabla}_{t,i}^{(j)}$, for all $j \in [M]$ on server. By (5), we define

$$\forall i \in [N], \quad \tilde{c}_{t,i}^{(j)} = \mathbf{w}_{t,i}^\top \left( \tilde{\mathbf{x}}_t^{(j)}(\hat{\mathbf{x}}_t^{(j)})^\top [S_i, S_i] \right) \mathbf{w}_{t,i} - \mathbf{w}_{t,i}^\top \cdot y_t^{(j)} \left( \tilde{\mathbf{x}}_t^{(j)} + \hat{\mathbf{x}}_t^{(j)} \right) [S_i],$$

$$\tilde{\nabla}_{t,i}^{(j)} = 2 \left( \tilde{\mathbf{x}}_t^{(j)}(\hat{\mathbf{x}}_t^{(j)})^\top [S_i, S_i] \right) \mathbf{w}_{t,i} - y_t^{(j)} \left( \tilde{\mathbf{x}}_t^{(j)} + \hat{\mathbf{x}}_t^{(j)} \right) [S_i].$$

Then our algorithm averages the estimators and updates the probability and hypotheses as follows

$$\begin{cases} \mathbf{p}_{t+1} = \underset{\mathbf{p} \in \Delta_{t,N}}{\arg\min} \langle \bar{\mathbf{c}}_t, \mathbf{p} \rangle + \mathcal{B}_{\psi_t}(\mathbf{p}, \mathbf{p}_t), & \bar{c}_{t,i} = \frac{1}{M} \sum_{j=1}^M \tilde{c}_{t,i}^{(j)}, \quad i = 1, ..., N, \\[2mm] \mathbf{w}_{t+1,i} = \underset{\mathbf{w} \in \mathcal{W}}{\arg\min} \langle \bar{\nabla}_{t,i}, \mathbf{w} \rangle + \mathcal{B}_{\psi_{t,i}}(\mathbf{p}, \mathbf{w}_{t,i}), & \bar{\nabla}_{t,i} = \frac{1}{M} \sum_{j=1}^M \tilde{\nabla}_{t,i}^{(j)}, \quad i = 1, ..., N, \end{cases} \tag{11}$$

in which $\Delta_{t,N}$, $\mathcal{W}$, $\psi_t$ and $\psi_{t,i}$ follow (6).

We name this algorithm FedAMRO (Federated AMRO) and give the pseudo-code in Algorithm 3. Previous work has proposed a federated online mirror descent, called FOMD-No-LU (Li et al., 2024). There is a critical difference between FedAMRO and FOMD-No-LU. FedAMRO uses a time-variant decision set $\Delta_{t,N}$, while FOMD-No-LU uses a constant decision set $\Delta_N$. Thus, FOMD-No-LU must know the prior information of $T$, while FedAMRO does not. The regret analysis of FedAMRO is also more technical.

# 6 MAIN RESULTS

## 6.1 UPPER BOUND ON THE REGRET OF AMRO

**Theorem 2** (Regret Bound of AMRO). *Let $b' \geq b + 2$, $\beta_t = \frac{1}{t}$, $\delta_t$ satisfy (8) and*

$$\eta_t = \frac{\sqrt{\ln N}}{\sqrt{\xi_t \cdot \ln N + \sum_{\tau=1}^t \sum_{i=1}^N p_{\tau,i} \tilde{c}_{\tau,i}^2}}, \quad \lambda_{t,i} = \frac{U}{\sqrt{1 + \sum_{\tau=1}^t \|\tilde{\nabla}_{\tau,i}\|_2^2}}, \quad i = 1, ..., N,$$

*where $\xi_t = \max\{\max_{\tau \leq t, i \in [N]} \tilde{c}_{\tau,i}^2, 0.01\}$. For any sequence of examples, the expected regret of AMRO satisfies*

$$\forall \mathbf{w} \in \mathcal{W}_b, \mathbb{E}[\text{Reg}(\mathbf{w})]$$

$$= O \left( \sqrt{U^4 \sum_{t=1}^T \sum_{i=1}^N p_{t,i} \Xi_{t,i}^2 + (UX+Y)^4 T \frac{\ln(NT)}{\sqrt{\ln N}}} + \frac{(d-b)^2}{(b'-b)^2} U^2 X^2 \ln(NT) \right).$$

---

**Algorithm 3** FedAMRO

---

**Input:** $U, b, b', N = \binom{b}{d}$
**Initialization:** $f_{1,i} = 0, p_{1,i} = \frac{1}{N}, \forall i \in [N]$
1: Divide $[d]$ into $N$ subsets of size $b$, denoted by $S_1, \ldots, S_N$
2: **for** $t = 1, \ldots, T$ **do**
3:    **for** $j = 1, ..., M$ **do**
4:        Server samples $I_t^{(j)}$ following $\mathbf{p}_t$
5:        Server sends $S_{I_t^{(j)}}$ and $\mathbf{w}_{t,I_t^{(j)}}$ to the client
6:    **end for**
7:    **for** $j = 1, ..., M$ in parallel **do**
8:        Client selects $\mathbf{x}_t^{(j)}[S_{I_t^{(j)}}]$
9:        Client computes $\hat{y}_t^{(j)}$ following (10)
10:        Client samples $\tilde{B}_t^{(j)}$ and $\tilde{B}_t^{(j)}$
11:        Client computes $\hat{\mathbf{x}}_t^{(j)}$ and $\tilde{\mathbf{x}}_t^{(j)}$
12:        Client sends $\tilde{\mathbf{x}}_t^{(j)}(\hat{\mathbf{x}}_t^{(j)})^\top, y_t^{(j)}(\hat{\mathbf{x}}_t^{(j)} + \tilde{\mathbf{x}}_t^{(j)})$ to server
13:    **end for**
14:    Server solves $\mathbf{p}_{t+1}$ by Algorithm 1
15:    Server computes $\{\mathbf{w}_{t+1,i}\}_{i=1}^N$ following (11)
16: **end for**

---

In the worst case, it must be

$$\sum_{t=1}^{T}\sum_{i=1}^{N} p_{t,i}\Xi_{t,i}^2 = O\left(\frac{(d-b)^2}{(b'-b)^2}X^4 \cdot T\right).$$

Then we obtain a $O(\frac{d-b}{b'-b}\sqrt{T}\frac{\ln NT}{\sqrt{\ln N}})$ regret. In certain benign environment, if $\Xi_{t,i} = O(X^2)$, then the regret is $O(\sqrt{T}\frac{\ln NT}{\sqrt{\ln N}})$. In general, $\ln N = O(b\ln d)$. In the cases of $b = r$ or $b = d - r$ where $r$ is independent of $d$, then $\ln N = O(\ln d)$. Table 1 summarizes the regret bounds of AMRO and three previous algorithms, including Alg1 (Foster et al., 2016), the first algorithm in Ito et al. (2017) named Alg2, and the second algorithm in Kale et al. (2017) named Alg3. Alg2 and Alg3 adopt additional assumptions for achieving a $O(\text{poly}(d))$ computational complexity.

Table 1: Regret bounds of AMRO and previous algorithms.

| Algorithm | Regret Bound | Assumption |
|---|---|---|
| Alg1 | $O(\frac{d^2}{(b'-b)^2}\sqrt{T\ln N})$ | No |
| Alg2 | $O(d^{\frac{1}{3}}b^{\frac{4}{3}}T^{\frac{2}{3}} \cdot \text{poly}(\log T))$ | Yes |
| Alg3 | $O(\frac{d}{b'-b}\sqrt{T})$ | Yes |
| AMRO | $O(\frac{d-b}{b'-b}\sqrt{T} \cdot \frac{\ln NT}{\sqrt{\ln N}})$ | No |

### 6.2 UPPER BOUND ON THE REGRET OF FEDAMRO

**Theorem 3** (Regret Bound of FedAMRO). *Assuming that* $\max_{f,(\mathbf{x},y)} \ell(f(\mathbf{x}), y) \leq C$. *Let* $b' \geq b + 2$, $\beta_t = \frac{1}{t}$, $\sigma_j \in (\frac{b'-b}{2(d-b)}, 1)$ *for all* $j \in [M]$, $\delta_{t,m}^{(j)}$ *satisfy (8), and*

$$\eta_t = \frac{\sqrt{\ln N}}{\sqrt{\xi_1 \ln N + \alpha \cdot t}}, \quad \lambda_{t,i} = \frac{U}{\sqrt{\frac{CX^2}{4} + \frac{2\mu_2^2 U^2 X^4}{M} + \frac{(2\mu_2+1)X^2 C}{16M}} \cdot \sqrt{t}}$$

*in which*

$$\alpha = \frac{(C-Y^2)^2}{4} + \frac{2\mu_2^2 U^4 X^4 + \mu_2 U^2 X^2 C + \frac{(C-Y^2)^2}{8}}{M},$$

$$\xi_1 = 4\left(C - Y^2 + (\mu_1 U X)^2 + 2\mu_1 U X Y\right)^2, \quad \mu_1 = \frac{4d - b' - 3b}{b' - b}, \quad \mu_2 = \frac{2d - b' - b}{b' - b}.$$

*For any sequence of examples, the expected regret of FedAMRO satisfies*

$$\forall \mathbf{w} \in \mathcal{W}_b, \quad \mathbb{E}\left[\text{Reg}(\mathbf{w})\right]$$

$$= O\left(M\frac{(d-b)^2}{(b'-b)^2}U^2 X^2 \ln(NT) + MUX\sqrt{(UX+Y)^2 T + \frac{U^2 X^2 (d-b)^2}{M(b'-b)^2}T\frac{\ln(NT)}{\sqrt{\ln N}}}\right).$$

The dominated term is $O(\frac{d-b}{b'-b}\sqrt{MT}\frac{\ln(NT)}{\sqrt{\ln N}})$. If we independently run AMRO on each client, then the regret bound is $O(\frac{d-b}{b'-b}M\sqrt{T}\frac{\ln(NT)}{\sqrt{\ln N}})$. FedAMRO enjoys a better regret bound, as it averages the estimator of losses and gradients over $M$ clients, thereby reducing the variance of estimators by a factor of $M$. However, it is imperative to establish a lower bound on the regret of any noncooperative algorithm in order to demonstrate the necessity of federated learning for OSLR-DecD.

### 6.3 Lower Bound on the Regret of Any Noncooperative Algorithm

**Theorem 4** (Lower Bound). *Let $d \geq 4$, $b = 1$ and $d > b' \geq 1$. The regret of any, possibly randomized, noncooperative algorithm for OSLR-DecD satifies*

$$\sup_{(\mathbf{x}_t^{(j)},y_t^{(j)}),t\in[T],j\in[M]} \mathbb{E}\left[\max_{\mathbf{w}\in\mathcal{W}_1}\text{Reg}(\mathbf{w})\right] \geq \frac{9M}{100}\sqrt{\frac{dT}{b'}},$$

*where the expectation is taken over the internal randomness of algorithm.*

Compared with the upper bound in Theorem 3, in the case of $d = o(M)$ (or $M = \omega(d)$), the lower bound is larger than the upper bound. Thus *FedAMRO is better than all noncooperative algorithms, demonstrating that federated learning is indeed necessary for OSLR-DecD.*

To prove Theorem 4, we establish a non-trivial reduction from OSLR to prediction with limited advice (Seldin et al., 2014) and prove the first lower bound for OSLR. By this lower bound, we further establish the lower bound on the regret of any noncooperative algorithm for OSLR-DecD.

## 7 Experiments

In this section, we aim to verify the following two goals.

**G**1 For OSLR, AMRO enjoys better prediction performance than all of previous algorithms.

**G**2 In the case of $M = \omega(d)$, federated learning is necessary for OSLR-DecD.

We download 6 regression datasets shown in Table 2, from WEKA [1] (Hall et al., 2009), LIBSVM [2] and UCI [3]. We normalize the datasets by setting $y \leftarrow \frac{y-\min_t y_t}{\max_t y_t - \min_t y_t}$ and $\mathbf{x} \leftarrow \min\{1, \frac{4}{\|\mathbf{x}\|_2}\}\cdot\mathbf{x}$. All algorithms are implemented in R on a Windows machine with 2.8 GHz Core(TM) i7-1165G7 CPU. We execute each experiment 10 times with random permutation of all datasets.

### 7.1 Results of OSLR

We compare AMRO with Alg1 (Foster et al., 2016), Alg2 (Ito et al., 2017) and Alg3 (Kale et al., 2017). The three baseline algorithms use $b$ attributes to make predictions, and observes additional $b' - b$ attributes. The values of $b$ and $b'$ for each dataset are given in Table 2. For Alg1, there are two learning rates $\eta_{\text{HEDGE}} = cq\sqrt{\ln(d)/T}$ and $\eta_{\text{OGD}} = cq\sqrt{1/T}$ where $q = \frac{(b'-b)(b'-b-1)}{d(d-1)}$. The original

Table 2: Datasets used in the experiments

| Datasets | #Sample | # Feature | $b$ | $b'$ |
|---|---|---|---|---|
| parkinson | 5,875 | 16 | 2 | 4 |
| cpusmall | 8,190 | 12 | 2 | 4 |
| elevators | 16,590 | 18 | 2 | 4 |
| calhousing | 14,000 | 8 | 2 | 4 |
| bank | 8,192 | 32 | 1 | 3 |
| ailerons | 13,750 | 40 | 1 | 3 |

paper sets $c = 1$, while we tune $c \in \{0.1, 0.5, 1, 5, 10, 100, 500\}$. For Alg2, there is a learning rate $\lambda_t = c\sqrt{t}/q$. The original paper sets $c = 8$, while we tune $c \in \{0.05, 0.1, 0.5, 1, 4, 8, 16\}$. For Alg3, there is a batch size $B = c_1\lfloor(T/d)^{\frac{1}{3}}\rfloor$ and a learning rate $\eta = c_2\sqrt{2\ln(d)/\lfloor T/B\rfloor}$. The original paper sets $c_1 = c_2 = 1$, while we tune $c_1 \in \{1, 10, 50, 100\}$ and $c_2 \in \{0.1, 0.5, 1, 5, 10, 50\}$. For FedAMRO, all of the hyperparameters follow Theorem 2, and $\delta_{t,m}$ follows (9). Besides we multiple by 4 on $\eta_t$. For Alg1, Alg2 and AMRO, we tune $U \in \{0.1, 0.5, 1\}$.

---

[1] https://waikato.github.io/weka-wiki/datasets/

[2] https://www.csie.ntu.edu.tw/~cjlin/libsvm/

[3] http://archive.ics.uci.edu/datasets

Table 3: Experimental Results on OSLR.

| Algorithm | elevators ($b=2$) | | parkinson ($b=2$) | | bank ($b=1$) | |
|---|---|---|---|---|---|---|
| | MSE$\times 10^2$ | Time (s) | MSE$\times 10^2$ | Time (s) | MSE$\times 10^2$ | Time (s) |
| Alg1 | $2.58 \pm 0.01$ | 23.59 | $6.44 \pm 0.09$ | 9.92 | $3.07 \pm 0.01$ | 2.94 |
| Alg2 | $1.06 \pm 0.01$ | 1.82 | $\mathbf{5.80 \pm 0.02}$ | 0.71 | $3.00 \pm 0.02$ | 0.74 |
| Alg3 | $1.68 \pm 0.11$ | 0.33 | $10.67 \pm 0.67$ | 0.15 | $2.82 \pm 0.04$ | 0.21 |
| AMRO | $\mathbf{0.89 \pm 0.12}$ | 21.26 | $\mathbf{5.84 \pm 0.02}$ | 6.66 | $\mathbf{2.35 \pm 0.07}$ | 2.33 |

| Algorithm | cpusmall ($b=2$) | | calhousing ($b=2$) | | ailerons ($b=1$) | |
|---|---|---|---|---|---|---|
| | MSE$\times 10^2$ | Time (s) | MSE$\times 10^2$ | Time (s) | MSE$\times 10^2$ | Time (s) |
| Alg1 | $8.32 \pm 0.22$ | 7.22 | $9.33 \pm 0.06$ | 5.95 | $7.70 \pm 0.57$ | 5.24 |
| Alg2 | $3.86 \pm 0.49$ | 0.61 | $6.08 \pm 0.05$ | 1.14 | $16.15 \pm 2.97$ | 1.00 |
| Alg3 | $3.61 \pm 0.31$ | 0.20 | $5.79 \pm 0.58$ | 0.30 | $7.18 \pm 0.83$ | 0.31 |
| AMRO | $\mathbf{2.82 \pm 0.48}$ | 4.61 | $\mathbf{3.60 \pm 0.23}$ | 4.38 | $\mathbf{5.35 \pm 0.76}$ | 5.44 |

By Table 3, AMRO enjoys the smallest MSE on all datasets. AMRO naturally outperforms Alg1 and Alg3 as it enjoys a smaller regret bound. Alg2 has a smaller regret bound than AMRO but performs worse, likely due to the additional assumptions required by Alg2 are not satisfied on the datasets. Although AMRO has a longer running time compared to Alg2 and Alg3, our focus is not on computationally efficient algorithms for OSLR, but the necessity of federated learning for OSLR-DecD. The experimental results verify **G**1.

## 7.2 Results of OSLR-DecD

We compare FedAMRO with two noncooperative algorithms, i.e, IndAMRO and IndAlg3 which independently runs AMRO and Alg3 on each client without collaboratiton, respectively. We exclude Alg1 and Alg2 due to their unsatisfactory performance for OSLR. Let $M = \lfloor \frac{1}{2} d^{\frac{3}{2}} \rfloor$. In this case, FedAMRO is better than all of noncooperative algorithms. For each dataset, we remove some examples such that $T = \lfloor \frac{T}{M} \rfloor \cdot M$, and uniformly divide the examples onto the $M$ clients. For FedAMRO, all of the hyperparameters follow Theorem 3, except that $\xi_1 = \max\{0.01, \max_{i \le M, \tau \le t-1} \bar{c}_{\tau,i}\}$. Let $\delta_{t,m}^{(j)}$ follow (9) and $\sigma^{(j)} = 0.8$ for all $j \in [M]$. Due to the data normalization, we set $Y = X = 1$, $C = 0.5$. We multiple by 4 on $\eta_t$, and tune $U \in \{0.1, 0.5, 1\}$ for IndAMRO and FedAMRO.

Table 4 summaries the results. FedAMRO is significantly better than the two noncooperative algorithms, demonstrating the necessity of federated learning for OSLR-DecD. Besides, the MSE of FedAMRO is also comparable with the MSE of AMRO in Table 3, showing that collaboration can compensate the degeneration of learning performance induced by decentralizing data. The experimental results verify **G**2. We give more experimental results in the supplementary material.

Table 4: Experimental Results OSLR-DecD.

| Algorithm | elevators ($b=2$) | parkinson ($b=2$) | bank ($b=1$) |
|---|---|---|---|
| | MSE | MSE | MSE |
| IndAlg3 | $0.0205 \pm 0.0006$ | $0.1637 \pm 0.0024$ | $0.0297 \pm 0.0004$ |
| IndAMRO | $0.0228 \pm 0.0003$ | $0.0721 \pm 0.0012$ | $0.0298 \pm 0.0002$ |
| FedAMRO | $\mathbf{0.0119 \pm 0.0015}$ | $\mathbf{0.0657 \pm 0.0023}$ | $\mathbf{0.0260 \pm 0.0003}$ |

| Algorithm | cpusmall ($b=2$) | calhousing ($b=2$) | ailerons ($b=1$) |
|---|---|---|---|
| | MSE | MSE | |
| IndAlg3 | $0.0475 \pm 0.0068$ | $0.0750 \pm 0.0030$ | $0.1270 \pm 0.0077$ |
| IndAMRO | $0.0526 \pm 0.0024$ | $0.0544 \pm 0.0016$ | $0.0887 \pm 0.0026$ |
| FedAMRO | $\mathbf{0.0344 \pm 0.0033}$ | $\mathbf{0.0460 \pm 0.0037}$ | $\mathbf{0.0731 \pm 0.0049}$ |

## 8 Conclusion

In this paper, we have proved that federated learning is necessary for OSLR-DecD, contrasting with the pessimistic result that federated learning is unnecessary for OLR-DecD in full information setting. We proposed a federated algorithm for DecD-OSLR, and proved that its regret bound is smaller than the lower bound of any noncooperative algorithm in the case of $d = o(M)$. The experimental results also verify our theoretical findings. Our work demonstrates that federated learning is necessary for online learning with decentralized data in the case of limited resource.

In the future, it would be interesting to close the gap between the lower and upper bounds on the regret, and explore whether the condition $d = o(M)$ can be relaxed, such as for any value of $d > b$.

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

## A  Algorithm for Solving $\mathbf{p}_{t+1}$

At any round $t$, $\mathbf{p}_{t+1}$ is updated as follows,

$$\mathbf{p}_{t+1} = \operatorname*{arg\,min}_{\mathbf{p} \in \Delta_{t,N}} \left\{ \langle \tilde{\mathbf{c}}_t, \mathbf{p} \rangle + \mathcal{B}_{\psi_t}(\mathbf{p}, \mathbf{p}_t) \right\},$$

$$\Delta_{t,N} = \left\{ \mathbf{p} \in \Delta_N : p_i \geq N^{-1}\beta_t, i \in [N] \right\}, \qquad \psi_t(\mathbf{p}) = \sum_{i=1}^{N} \frac{1}{\eta_t} p_i \ln p_i.$$

For any $\mathbf{p}, \mathbf{q}$, the Bregman divergence is

$$\mathcal{B}_{\psi_t}(\mathbf{p}, \mathbf{q}) = \frac{1}{\eta_t} \sum_{i=1}^{N} p_i \ln \frac{p_i}{q_i} - \frac{1}{\eta_t} \sum_{i=1}^{N} p_i + \frac{1}{\eta_t} \sum_{i=1}^{N} q_i. \tag{12}$$

We use the Lagrangian multiplier method to solve $\mathbf{p}_{t+1}$.

$$\min_{\mathbf{p} \in \Delta_{t,N}} \sum_{i=1}^{N} \tilde{c}_{t,i} p_i + \frac{1}{\eta_t} \sum_{i=1}^{N} p_i \ln \frac{p_i}{p_{t,i}},$$

$$\text{s.t.} \quad \sum_{i=1}^{N} p_i = 1,$$

$$p_i \geq \frac{\beta_t}{N}, \forall i \in [N].$$

The Lagrangian function is

$$L = \sum_{i=1}^{N} \tilde{c}_{t,i} p_i + \frac{1}{\eta_t} \sum_{i=1}^{N} p_i \ln \frac{p_i}{p_{t,i}} + \lambda \left( \sum_{i=1}^{N} p_i - 1 \right) + \sum_{i=1}^{N} \gamma_i \left( \frac{\beta_t}{N} - p_i \right).$$

The KKT conditions are

$$
\begin{cases}
\frac{\mathrm{d}\,L}{\mathrm{d}\,p_i} = 0, \forall i \in [N], \\
\sum_{i=1}^{N} p_i = 1, \\
\lambda \neq 0, \\
\gamma_i \geq 0, \\
\gamma_i \cdot \left( \frac{\beta_t}{N} - p_i \right) = 0, \forall i \in [N].
\end{cases}
$$

By the first two conditions, we have

$$
p_i = p_{t,i} \exp\left( -\left( \tilde{c}_{t,i} + \lambda - \gamma_i \right) \eta_t - 1 \right), \quad \sum_{i=1}^{N} p_{t,i} \exp\left( -\left( \tilde{c}_{t,i} + \lambda - \gamma_i \right) \eta_t - 1 \right) = 1.
$$

Solving for $\exp\left( -\lambda \eta_t - 1 \right)$ yields

$$
\exp\left( -\lambda \eta_t - 1 \right) = \frac{1}{\sum_{i=1}^{N} p_{t,i} \exp\left( -\left( \tilde{c}_{t,i} - \gamma_i \right) \eta_t \right)}.
$$

Thus we have

$$
p_i = \frac{p_{t,i} \exp\left( -\left( \tilde{c}_{t,i} - \gamma_i \right) \eta_t \right)}{\sum_{j=1}^{N} p_{t,j} \exp\left( -\left( \tilde{c}_{t,j} - \gamma_j \right) \eta_t \right)}.
$$

Note that there is not a analytical solution. Next we construct a feasible solution.

Initializing $\gamma_i = 0$ for all $i = 1, ..., N$. Then we have

$$
\tilde{p}_i = \frac{p_{t,i} \exp\left( -\tilde{c}_{t,i} \eta_t \right)}{\sum_{j=1}^{N} p_{t,j} \exp\left( -\tilde{c}_{t,j} \eta_t \right)}.
$$

If $\tilde{p}_i \geq \frac{\beta_t}{N}$ for all $i \in [N]$, then

$$
p_i = \tilde{p}_i, \quad \forall i = 1, ..., N.
$$

Let $\mathcal{A} = \{ i \in [N] : p_i < \frac{\beta_t}{N} \}$ and $\mathcal{A}^{\neg} = \{ 1, ..., N \} \setminus \mathcal{A}$. For all $i \in \mathcal{A}$, we increase the value of $\gamma_i$ such that

$$
p_{t,i} \exp\left( -\left( \tilde{c}_{t,i} - \gamma_i \right) \eta_t \right) = z_t,
$$

$$
\frac{z_t}{\sum_{j \in \mathcal{A}^{\neg}} p_{t,j} \exp\left( -\tilde{c}_{t,j} \eta_t \right) + |\mathcal{A}| \cdot z_t} = \frac{\beta_t}{N}.
$$

Solving for $z_t$ gives

$$
z_t = \frac{\beta_t \cdot \sum_{j \in \mathcal{A}^{\neg}} p_{t,j} \exp\left( -\tilde{c}_{t,j} \eta_t \right)}{(N - |\mathcal{A}| \cdot \beta_t)}
$$

$$
p_i = \frac{\beta_t}{N}, \quad \forall i \in \mathcal{A},
$$

$$
p_i = \frac{p_{t,i} \exp\left( -\tilde{c}_{t,i} \eta_t \right)}{\sum_{j \in \mathcal{A}^{\neg}} p_{t,j} \exp\left( -\tilde{c}_{t,j} \eta_t \right) + |\mathcal{A}| \cdot z_t}, \quad \forall i \in \mathcal{A}^{\neg}.
$$

If there are some $i \in \mathcal{A}^{\neg}$ such that $p_i < \frac{\beta_t}{N}$ then we update

$$
\mathcal{A} \leftarrow \mathcal{A} \cup \left\{ i \in \mathcal{A}^{\neg}, p_i < \frac{\beta_t}{N} \right\},
$$

then we repeat the above precess.

We obtain $\mathbf{p}_{t+1}$ by setting $p_{t+1,i} = p_i$ for all $i \in [N]$. The pseudo-code is shown in Algorithm 1.

## B  PROOF OF LEMMA 1

*Proof.* For any $m \in S_{I_t}$, we have $\tilde{x}_{t,m} = x_{t,m}$ and

$$
\forall m \notin S_{I_t}, \quad \mathbb{E}\left[ \tilde{x}_{t,m} \right] = \mathbb{P}[m \in \tilde{B}_t] \cdot \left( \frac{x_{t,m} - \delta_{t,m}}{\mathbb{P}[m \in \tilde{B}_t]} + \delta_{t,m} \right) + \left( 1 - \mathbb{P}[m \in \tilde{B}_t] \right) \cdot \delta_{t,m} = x_{t,m}.
$$

Thus $\mathbb{E}\left[\tilde{\mathbf{x}}_t\right] = \mathbf{x}_t$. The expectation is taken over $\tilde{B}_t$ and $\hat{B}_t$. We can also prove that $\mathbb{E}\left[\hat{\mathbf{x}}_t\right] = \mathbf{x}_t$.

Recalling that,

$$\forall i \in [N], \quad \tilde{c}_{t,i} = \mathbf{w}_{t,i}^\top \tilde{\mathbf{x}}_t \hat{\mathbf{x}}_t^\top [S_i, S_i] \mathbf{w}_{t,i} - y_t \mathbf{w}_{t,i}^\top (\tilde{\mathbf{x}}_t[S_i] + \hat{\mathbf{x}}_t[S_i])$$

$$= (\mathbf{w}_{t,i}^\top \tilde{\mathbf{x}}_t[S_i] - y_t) \cdot (\mathbf{w}_{t,i}^\top \hat{\mathbf{x}}_t[S_i] - y_t) - y_t^2,$$

$$\tilde{\nabla}_{t,i} = 2\tilde{\mathbf{x}}_t \hat{\mathbf{x}}_t^\top [S_i, S_i] \mathbf{w}_{t,i} - y_t(\tilde{\mathbf{x}}_t + \hat{\mathbf{x}}_t)[S_i]$$

$$= (\mathbf{w}_{t,i}^\top \tilde{\mathbf{x}}_t[S_i] - y_t)\hat{\mathbf{x}}_t[S_i] + (\mathbf{w}_{t,i}^\top \hat{\mathbf{x}}_t[S_i] - y_t)\tilde{\mathbf{x}}_t[S_i].$$

We first show that the estimators are unbiased.

$$\mathbb{E}\left[\tilde{c}_{t,i}\right] = c_{t,i} \cdot \mathbb{I}_{i=I_t} + \mathbb{E}\left[\hat{c}_{t,i}\right] \cdot \mathbb{I}_{i\neq I_t}$$

$$= c_{t,i} \cdot \mathbb{I}_{i=I_t} + \mathbb{E}_{\tilde{B}_t}\left[\left(\mathbf{w}_{t,i}^\top \tilde{\mathbf{x}}_t[S_i] - y_t\right)\right] \cdot \mathbb{E}_{\hat{B}_t}\left[\left(\mathbf{w}_{t,i}^\top \hat{\mathbf{x}}_t[S_i] - y_t\right)\right] \cdot \mathbb{I}_{i\neq I_t} - y_t^2 \cdot \mathbb{I}_{i\neq I_t}$$

$$= c_{t,i} \cdot \mathbb{I}_{i=I_t} + \left[\left(\mathbf{w}_{t,i}^\top \mathbf{x}_t[S_i] - y_t\right)^2 - y_t^2\right] \cdot \mathbb{I}_{i\neq I_t}$$

$$= c_{t,i},$$

where the first equality comes from the fact $\tilde{\mathbf{x}}_t$ is independent of $\hat{\mathbf{x}}_t$. Similarly, we can prove

$$\mathbb{E}\left[\tilde{\nabla}_{t,i}\right] = 2(\mathbf{w}_{t,i}^\top \mathbf{x}_t[S_i] - y_t) \cdot \mathbf{x}_t[S_i] \cdot \mathbb{I}_{i=I_t} + 2\mathbb{E}\left[(\mathbf{w}_{t,i}^\top \hat{\mathbf{x}}_t[S_i] - y_t) \cdot \tilde{\mathbf{x}}_t[S_i]\right] \cdot \mathbb{I}_{i\neq I_t} = \nabla_{t,i}.$$

Next we analyze the second order moment.

$$\mathbb{E}\left[\tilde{c}_{t,i}^2\right] = c_{t,i}^2 \cdot \mathbb{I}_{i=I_t} + \mathbb{E}\left[\tilde{c}_{t,i}^2\right] \cdot \mathbb{I}_{i\neq I_t}$$

$$= c_{t,i}^2 \cdot \mathbb{I}_{i=I_t} + \mathbb{E}_{\tilde{B}_t}\left[\left(\mathbf{w}_{t,i}^\top \tilde{\mathbf{x}}_t[S_i] - y_t\right)^2\right] \cdot \mathbb{E}_{\hat{B}_t}\left[\left(\mathbf{w}_{t,i}^\top \hat{\mathbf{x}}_t[S_i] - y_t\right)^2\right] \cdot \mathbb{I}_{i\neq I_t} -$$

$$2\left(\mathbf{w}_{t,i}^\top \mathbf{x}_t[S_i] - y_t\right)^2 y_t^2 \cdot \mathbb{I}_{i\neq I_t} + y_t^4 \cdot \mathbb{I}_{i\neq I_t}.$$

We just need to analyze $\mathbb{E}_{\hat{B}_t}\left[\left(\mathbf{w}_{t,i}^\top \hat{\mathbf{x}}_t[S_i] - y_t\right)^2\right]$. For simplicity, let

$$\varepsilon_{t,m} = x_{t,m} - \delta_{t,m}.$$

Then

$$\mathbb{E}_{\tilde{B}_t}\left[\left(\mathbf{w}_{t,i}^\top \tilde{\mathbf{x}}_t[S_i] - y_t\right)^2\right]$$

$$= \mathbb{E}_{\tilde{B}_t}\left[\sum_{m \in S_i} w_{t,i,m}^2 \cdot \tilde{x}_{t,m}^2\right] + \mathbb{E}_{\tilde{B}_t}\left[\sum_{m \neq n \in S_i} w_{t,i,m} w_{t,i,n} \tilde{x}_{t,m} \tilde{x}_{t,n}\right] - 2y_t \mathbb{E}_{\tilde{B}_t}\left[\mathbf{w}_{t,i}^\top \tilde{\mathbf{x}}_t[S_i]\right] + y_t^2$$

$$\leq \sum_{m \in S_i \setminus S_{I_t}} w_{t,i,m}^2 \cdot \left(2\frac{d-b}{b'-b}\varepsilon_{t,m}^2 + 2\delta_{t,m}\varepsilon_{t,m} + \delta_{t,m}^2\right) + \sum_{m \in S_i \cap S_{I_t}} w_{t,i,m}^2 \cdot x_{t,m}^2 +$$

$$\sum_{m \neq n \in S_i} w_{t,i,m} w_{t,i,n} x_{t,m} x_{t,n} - 2y_t \mathbf{w}_{t,i}^\top \mathbf{x}_t[S_i] + y_t^2$$

$$= \frac{2d - b' - b}{b' - b} \sum_{m \in S_i \setminus S_{I_t}} w_{t,i,m}^2 \cdot \varepsilon_{t,m}^2 + \left(\mathbf{w}_{t,i}^\top \mathbf{x}_t[S_i] - y_t\right)^2$$

$$\leq \frac{2d - b' - b}{b' - b} \sum_{m \in S_i} w_{t,i,m}^2 \cdot \varepsilon_{t,m}^2 + \left(\mathbf{w}_{t,i}^\top \mathbf{x}_t[S_i] - y_t\right)^2$$

$$= U^2 \Xi_{t,i} + \left(\mathbf{w}_{t,i}^\top \mathbf{x}_t[S_i] - y_t\right)^2,$$

where

$$\Xi_{t,i} = \frac{2d - b' - b}{b' - b} \sum_{m \in S_i} \varepsilon_{t,m}^2, \qquad \mathbb{P}\left[m \in \tilde{B}_t\right] = \frac{b' - b}{2(d - b)}.$$

Thus

$$\mathbb{E}\left[\tilde{c}_{t,i}^2\right] \cdot \mathbb{I}_{i\neq I_t} \leq \left(U^2 \Xi_{t,i} + \left(\mathbf{w}_{t,i}^\top \mathbf{x}_t[S_i] - y_t\right)^2\right)^2 \cdot \mathbb{I}_{i\neq I_t} - 2(\mathbf{w}_{t,i}^\top \mathbf{x}_t[S_i] - y_t)^2 y_t^2 \cdot \mathbb{I}_{i\neq I_t} + y_t^4 \cdot \mathbb{I}_{i\neq I_t}$$

$$\leq \left(U^4 \Xi_{t,i}^2 + 2U^2(UX + Y)^2 \Xi_{t,i}\right) \cdot \mathbb{I}_{i\neq I_t} + c_{t,i}^2 \cdot \mathbb{I}_{i\neq I_t},$$

$$\mathbb{E}\left[\tilde{c}_{t,i}^2\right] \leq c_{t,i}^2 \cdot \mathbb{I}_{i=I_t} + \mathbb{E}\left[\tilde{c}_{t,i}^2 \cdot \mathbb{I}_{i\neq I_t}\right]$$

$$\leq U^4 \Xi_{t,i}^2 + 2U^2(UX + Y)^2 \Xi_{t,i} + c_{t,i}^2,$$

which recoveries the first statement.

The analysis for the estimator of gradient is similar.

$$
\begin{aligned}
\mathbb{E}\left[\left\|\tilde{\nabla}_{t,i}\right\|_2^2\right] =& \|\nabla_{t,i}\|_2^2 \cdot \mathbb{I}_{i=I_t} + \mathbb{E}\left[\left\|\tilde{\nabla}_{t,i}\right\|_2^2\right] \cdot \mathbb{I}_{i\neq I_t} \\
=& \|\nabla_{t,i}\|_2^2 \cdot \mathbb{I}_{i=I_t} + \mathbb{E}\left[\left\|(\mathbf{w}_{t,i}^\top \tilde{\mathbf{x}}_t[S_i] - y_t)\hat{\mathbf{x}}_t[S_i] + (\mathbf{w}_{t,i}^\top \hat{\mathbf{x}}_t[S_i] - y_t)\tilde{\mathbf{x}}_t[S_i]\right\|_2^2\right] \cdot \mathbb{I}_{i\neq I_t} \\
\leq& \|\nabla_{t,i}\|_2^2 \cdot \mathbb{I}_{i=I_t} + 4\mathbb{E}\left[\left\|(\mathbf{w}_{t,i}^\top \tilde{\mathbf{x}}_t[S_i] - y_t)\hat{\mathbf{x}}_t[S_i]\right\|_2^2\right] \cdot \mathbb{I}_{i\neq I_t} \\
=& \|\nabla_{t,i}\|_2^2 \cdot \mathbb{I}_{i=I_t} + 4\mathbb{E}_{\tilde{B}_t}\left[\left(\mathbf{w}_{t,i}^\top \tilde{\mathbf{x}}_t[S_i] - y_t\right)^2\right] \cdot \mathbb{E}_{\hat{B}_t}\left[\|\hat{\mathbf{x}}_t[S_i]\|_2^2\right] \cdot \mathbb{I}_{i\neq I_t} \\
\leq& \|\nabla_{t,i}\|_2^2 \cdot \mathbb{I}_{i=I_t} + 4\left(U^2\Xi_{t,i} + \left(\mathbf{w}_{t,i}^\top \mathbf{x}_t[S_i] - y_t\right)^2\right) \cdot \mathbb{E}_{\hat{B}_t}\left[\|\hat{\mathbf{x}}_t[S_i]\|_2^2\right] \cdot \mathbb{I}_{i\neq I_t}.
\end{aligned}
$$

Note that

$$
\begin{aligned}
\mathbb{E}_{\hat{B}_t}\left[\|\hat{\mathbf{x}}_t[S_i]\|_2^2\right] =& \mathbb{E}_{\hat{B}_t}\left[\sum_{m\in S_i} \hat{x}_{t,m}^2\right] \\
=& \sum_{m\in S_i}\left(\frac{2(d-b)}{b'-b}\varepsilon_{t,m}^2 + 2\delta_{t,m}\varepsilon_{t,m} + \delta_{t,m}^2\right) \\
=& \Xi_{t,i} + \|\mathbf{x}_t[S_i]\|_2^2 \\
\leq& \Xi_{t,i} + X^2.
\end{aligned}
$$

Thus

$$
\begin{aligned}
\mathbb{E}\left[\|\tilde{\nabla}_{t,i}^2\|^2\right] \leq& \|\nabla_{t,i}\|_2^2 \cdot \mathbb{I}_{i=I_t} + 4\left(U^2\Xi_{t,i} + \left(\mathbf{w}_{t,i}^\top \mathbf{x}_t[S_i] - y_t\right)^2\right) \cdot \left(\Xi_{t,i} + X^2\right) \cdot \mathbb{I}_{i\neq I_t} \\
\leq& 4\left(U^2\Xi_{t,i}^2 + 2(UX+Y)^2\Xi_{t,i} + X^2(UX+Y)^2\right) \\
\leq& 4U^2\Xi_{t,i}^2 + 8(UX+Y)^2(\Xi_{t,i} + X^2),
\end{aligned}
$$

which concludes the proof. $\qquad\square$

## C  SOME PROPERTIES OF OMD

**Lemma 2** (Boyd & Vandenberghe, 2004). *Assuming that $\Psi(\cdot) : \Omega \to \mathbb{R}$ is a convex and differential function, and $\Omega$ is a convex domain. Let $f^* = \operatorname{argmin}_{f\in\Omega}\Psi(f)$. Then*

$$
\forall g \in \Omega, \quad \langle \nabla\Psi(f^*), g - f^* \rangle \geq 0.
$$

### C.1  UPDATING SAMPLING PROBABILITY

We rewrite the updating of $\mathbf{p}_{t+1}$ as follows

$$
\mathbf{p}_{t+1} = \operatorname*{arg\,min}_{\mathbf{p}\in\Delta_{t,N}} \Psi_t(\mathbf{p}), \qquad \Psi_t(\mathbf{p}) = \langle \tilde{\mathbf{c}}_t, \mathbf{p} \rangle + \mathcal{B}_{\psi_t}(\mathbf{p}, \mathbf{p}_t).
$$

Taking derivation w.r.t. $\mathbf{p}_{t+1}$ gives

$$
\nabla\Psi_t(\mathbf{p}_{t+1}) = \tilde{\mathbf{c}}_t + \nabla\psi_t(\mathbf{p}_{t+1}) - \nabla\psi_t(\mathbf{p}_t). \tag{13}
$$

By Lemma 2, we have

$$
\forall \mathbf{u} \in \Delta_{t,N}, \quad \langle \nabla\Psi_t(\mathbf{p}_{t+1}), \mathbf{p}_{t+1} - \mathbf{u} \rangle \leq 0.
$$

### C.2  UPDATING HYPOTHESES

We rewrite the updating of $\mathbf{w}_{t+1,i}$ as follows

$$
\forall i \in [N], \quad \mathbf{w}_{t+1,i} = \operatorname*{arg\,min}_{\mathbf{w}\in\mathcal{W}} \Psi_{t,i}(\mathbf{w}), \qquad \Psi_{t,i}(\mathbf{w}) = \left\langle \tilde{\nabla}_{t,i}, \mathbf{w} \right\rangle + \mathcal{B}_{\psi_{t,i}}(\mathbf{w}, \mathbf{w}_{t,i}).
$$

Let $\psi_{t,i}(\mathbf{w}) = \frac{1}{2\lambda_{t,i}}\|\mathbf{w}\|_2^2$. Then

$$\mathcal{B}_{\psi_{t,i}}(\mathbf{w}, \mathbf{w}_{t,i}) = \frac{1}{2\lambda_{t,i}}\|\mathbf{w} - \mathbf{w}_{t,i}\|_2^2.$$

We use the Lagrangian multiplier method to obtain $\mathbf{w}_{t+1,i}$.

$$\forall i \in [N], \quad \mathbf{w}_{t+1,i} = \min\left\{1, \frac{U}{\|\tilde{\mathbf{w}}_{t+1,i}\|_2}\right\} \cdot \tilde{\mathbf{w}}_{t+1,i}, \qquad \tilde{\mathbf{w}}_{t+1,i} = \mathbf{w}_{t,i} - \lambda_{t,i}\tilde{\nabla}_{t,i}.$$

# D  TECHNICAL LEMMAS

In this section, we give some technical lemmas.

**Lemma 3** (Lemma 3.5 in Auer et al. (2002))**.** *Let* $\sigma_1, \sigma_2, \ldots, \sigma_T$ *and* $\xi$ *be non-negative real-number. Then*

$$\sum_{t=1}^{T} \frac{\sigma_t}{\sqrt{\xi + \sum_{\tau=1}^{t}\sigma_\tau}} \le 2\sqrt{\xi + \sum_{t=1}^{T}\sigma_t} - 2\sqrt{\xi}.$$

**Lemma 4.** *Let* $\sigma_1, \sigma_2, \ldots, \sigma_T$ *be non-negative real-number. Then*

$$\sum_{t=1}^{T} \frac{\sigma_t}{\sqrt{\xi_t + \sum_{\tau=1}^{t}\sigma_\tau}} \le 2\sqrt{\xi_T + \sum_{t=1}^{T}\sigma_t},$$

*where* $\xi_t = \max_{\tau \le t} \sigma_\tau$.

*Proof.* For any $a > 0$ and $b > 0$, we have $2\sqrt{a}\sqrt{b} \le a + b$. Let $a = \xi_t + \sum_{\tau=1}^{t}\sigma_\tau$ and $b = \xi_t + \sum_{\tau=1}^{t-1}\sigma_\tau$. Then we have

$$2\sqrt{\xi_t + \sum_{\tau=1}^{t}\sigma_\tau} \cdot \sqrt{\xi_t + \sum_{\tau=1}^{t-1}\sigma_\tau} \le 2\left(\xi_t + \sum_{\tau=1}^{t}\sigma_\tau\right) - \sigma_t.$$

Dividing by $\sqrt{a}$ and rearranging terms yields

$$\frac{1}{2}\frac{\sigma_t}{\sqrt{\xi_t + \sum_{\tau=1}^{t}\sigma_\tau}} \le \sqrt{\xi_t + \sum_{\tau=1}^{t}\sigma_\tau} - \sqrt{\xi_t + \sum_{\tau=1}^{t-1}\sigma_\tau} \le \sqrt{\xi_t + \sum_{\tau=1}^{t}\sigma_\tau} - \sqrt{\xi_{t-1} + \sum_{\tau=1}^{t-1}\sigma_\tau}.$$

Summing over $t = 1, \ldots, T$, we obtain

$$\sum_{t=1}^{T} \frac{\sigma_t}{\sqrt{\xi_t + \sum_{\tau=1}^{t-1}\sigma_\tau}} \le 2\sqrt{\xi_T + \sum_{\tau=1}^{T}\sigma_\tau} - 2\sqrt{\varepsilon_1 + \sigma_1},$$

which concludes the proof. $\qquad\square$

**Lemma 5.** *For any* $x \in (-1, 1)$,

$$\exp(-x) \le 1 - x + x^2.$$

*For any* $x > -1$,

$$\ln(1 + x) \le x.$$

**Lemma 6.** *Let* $0 < \beta_t \le 1$. *For any* $i \in [N]$, *let*

$$\eta_t = \frac{\sqrt{\ln N}}{\sqrt{\max_{\tau \le t, j \in [N]} \tilde{c}_{\tau,j}^2 \ln N + \sum_{\tau=1}^{t}\sum_{j=1}^{N} p_{\tau,j}\tilde{c}_{\tau,j}^2}},$$

$$\mathbf{u}_t \in \Delta_{t,N} = \left\{u_{t,j} = \frac{\beta_t}{N}, j \neq i, u_{t,i} = 1 - \frac{N-1}{N}\beta_t\right\},$$

$$A_t = \left\{j \in [N], \frac{u_{t+1,j}}{\eta_{t+1}} \ge \frac{u_{t,j}}{\eta_t}\right\}.$$

*Then it must be*

(a) $i \in A_t$,

(b)
$$\sum_{j \in A_t} \left( \frac{u_{t+1,j}}{\eta_{t+1}} - \frac{u_{t,j}}{\eta_t} \right) \leq \frac{1}{\eta_{t+1}} - \frac{1}{\eta_t}.$$

*Proof.* Since $\beta_{t+1} < \beta_t$, we have $u_{t+1,i} \geq u_{t,i}$. By $\eta_{t+1} \leq \eta_t$, we can obtain
$$\frac{u_{t+1,i}}{\eta_{t+1}} - \frac{u_{t,i}}{\eta_t} \geq 0.$$

Thus $i \in A_t$. For the second statement, we have
$$\sum_{j \in A_t} \left( \frac{u_{t+1,j}}{\eta_{t+1}} - \frac{u_{t,j}}{\eta_t} \right) = \frac{\frac{|A_t|-1}{N}\beta_{t+1} + 1 - \frac{N-1}{N}\beta_{t+1}}{\eta_{t+1}} - \frac{\frac{|A_t|-1}{N}\beta_t + 1 - \frac{N-1}{N}\beta_t}{\eta_t}$$
$$\leq \left( 1 - \left( \frac{N-1}{N} - \frac{|A_t|-1}{N} \right) \beta_{t+1} \right) \cdot \left( \frac{1}{\eta_{t+1}} - \frac{1}{\eta_t} \right)$$
$$\leq \frac{1}{\eta_{t+1}} - \frac{1}{\eta_t},$$

where we use the fact $|A_t| \leq N$. $\qquad\square$

**Lemma 7.** *Let $\beta_t = \frac{1}{t}$ and $\mathbf{u}_t$ be defined in [Lemma](#) 6. Then*
$$\sum_{t=1}^{T} \langle \mathbf{u}_t, \mathbf{c}_t \rangle - \sum_{t=1}^{T} c_{t,i} \leq \max_{t,j} c_{t,j} \cdot \ln T.$$

*Proof.* Substituting into $\mathbf{u}_t$ gives,
$$\sum_{t=1}^{T} \langle \mathbf{u}_t, \mathbf{c}_t \rangle - \sum_{t=1}^{T} c_{t,i} = \sum_{t=1}^{T} \left( \sum_{j=1}^{N} u_{t,j} c_{t,j} - c_{t,i} \right)$$
$$= \sum_{t=1}^{T} \left( \sum_{j \neq i} \frac{\beta_t}{N} c_{t,j} + \left( 1 - \frac{N-1}{N}\beta_t \right) c_{t,i} - c_{t,i} \right)$$
$$= \sum_{t=1}^{T} \left( \sum_{j \neq i} \frac{\beta_t}{N} c_{t,j} - \frac{N-1}{N}\beta_t c_{t,i} \right)$$
$$\leq \sum_{t=1}^{T} \frac{1}{t} \max_{j} c_{t,j}$$
$$\leq \max_{t,j} c_{t,j} \cdot \ln T,$$

which concludes the proof. $\qquad\square$

**Lemma 8.** *Let $\psi_t = \frac{1}{\eta_t} \sum_{j=1}^{N} p_j \ln p_j$, $\mathbf{p}_t \in \Delta_{t,N}$ and $\mathbf{u}_t$ be defined in [Lemma](#) 6. Assuming that $\eta_{t+1} \leq \eta_t$ for all $t \geq 1$, then*
$$\sum_{t=1}^{T} [\mathcal{B}_{\psi_t}(\mathbf{u}_t, \mathbf{p}_t) - \mathcal{B}_{\psi_t}(\mathbf{u}_t, \mathbf{p}_{t+1})] \leq \ln \left( \frac{N}{\beta_T} \right) \cdot \frac{1}{\eta_T}.$$

*Proof.* By the definition of Bregman divergence, we have

$$\sum_{t=1}^{T} [\mathcal{B}_{\psi_t}(\mathbf{u}_t, \mathbf{p}_t) - \mathcal{B}_{\psi_t}(\mathbf{u}_t, \mathbf{p}_{t+1})] \overset{(12)}{=} \sum_{t=1}^{T} \left[ \frac{1}{\eta_t} \sum_{i=1}^{N} u_{t,i} \ln \frac{u_{t,i}}{p_{t,i}} - \frac{1}{\eta_t} \sum_{i=1}^{N} u_{t,i} \ln \frac{u_{t,i}}{p_{t+1,i}} \right]$$

$$= \sum_{i=1}^{N} \left[ \sum_{t=1}^{T} \left( \frac{1}{\eta_t} u_{t,i} \ln \frac{1}{p_{t,i}} - \frac{1}{\eta_t} u_{t,i} \ln \frac{1}{p_{t+1,i}} \right) \right]$$

$$\leq \sum_{i=1}^{N} \left[ \sum_{t=1}^{T-1} \ln \frac{1}{p_{t+1,i}} \cdot \left( \frac{u_{t+1,i}}{\eta_{t+1}} - \frac{u_{t,i}}{\eta_t} \right) + \frac{u_{1,i}}{\eta_1} \ln \frac{1}{p_{1,i}} \right]$$

$$\leq \sum_{t=1}^{T-1} \sum_{i \in A_t} \ln \frac{1}{p_{t+1,i}} \cdot \left( \frac{u_{t+1,i}}{\eta_{t+1}} - \frac{u_{t,i}}{\eta_t} \right) + \frac{\ln N}{\eta_1}$$

$$\leq \ln \left( \frac{N}{\beta_T} \right) \cdot \sum_{t=1}^{T-1} \left( \frac{1}{\eta_{t+1}} - \frac{1}{\eta_t} \right) + \frac{\ln N}{\eta_1} \quad \text{(Lemma 6)}$$

$$\leq \ln \left( \frac{N}{\beta_T} \right) \cdot \frac{1}{\eta_T},$$

which concludes the proof. $\qquad \square$

## E  PROOF OF THEOREM 2

We first explain how the core part of the regret analysis of AMRO is different from previous work (Shalev-Shwartz, 2012; Foster et al., 2016). There are two core parts in the regret analysis of AMRO. The first one is to analyze the regret defined w.r.t. a sequence of time-variant competitors, i.e., $\mathbf{u}_1 \in \Delta_{1,N}, \mathbf{u}_2 \in \Delta_{2,N}, \ldots, \mathbf{u}_T \in \Delta_{T,N}$ owing to the utilization of time-varying decision sets $\Delta_{1,N}, \ldots, \Delta_{T,N}$. To this end, we require a crucial lemma that carefully controls the sum of the difference of Bregman divergence (please see Lemma 8), i.e.,

$$\sum_{t=1}^{T} [\mathcal{B}_{\psi_t}(\mathbf{u}_t, \mathbf{p}_t) - \mathcal{B}_{\psi_t}(\mathbf{u}_t, \mathbf{p}_{t+1})].$$

The analysis differs from standard analysis of OMD with constant decision $\Delta_N$ and constant learning rate (Shalev-Shwartz, 2012), which just controls a easier term defined as follows,

$$\sum_{t=1}^{T} [\mathcal{B}_{\psi}(\mathbf{u}, \mathbf{p}_t) - \mathcal{B}_{\psi}(\mathbf{u}, \mathbf{p}_{t+1})] = \mathcal{B}_{\psi}(\mathbf{u}, \mathbf{p}_1) - \mathcal{B}_{\psi}(\mathbf{u}, \mathbf{p}_{T+1}) \leq \mathcal{B}_{\psi}(\mathbf{u}, \mathbf{p}_1).$$

The second core part in the regret analysis is to bound the variances of estimators of gradient and loss (please see Lemma 1), enabling our algorithm to improve the regret bound in Foster et al. (2016) by a factor at least $O(d/b)$ and adapt to certain benign environments. We aim to prove that the variances depending on $\|\mathbf{x}_t - \delta_t\|_2^2$ where $\delta_t$ is an optimistic estimator of $\mathbf{x}_t$ using $\mathbf{x}_1, \mathbf{x}_2, \ldots, \mathbf{x}_{t-1}$. The variance bounds in Lemma 1 are data-dependent.

*Proof.* For any $\mathbf{w}' \in \mathcal{W}_b$, there exists a $\mathbf{w} \in \mathcal{W}$ and $i \in [N]$ such that

$$\sum_{t=1}^{T} (\langle \mathbf{w}', \mathbf{x}_t \rangle - y_t)^2 = \sum_{t=1}^{T} (\langle \mathbf{w}, \mathbf{x}_t[S_i] \rangle - y_t)^2.$$

With out loss of generality, assuming that the competitor $\mathbf{w}' \in \mathcal{W}_b$ uses the attributes indexed by elements in $S_i$ to make predictions. We decompose the regret as follows

$$
\begin{aligned}
\mathrm{Reg}(\mathbf{w}) =& \sum_{t=1}^{T}(\hat{y}_t - y_t)^2 - \sum_{t=1}^{T}(\langle \mathbf{w}, \mathbf{x}_t[S_i] \rangle - y_t)^2 \\
=& \sum_{t=1}^{T} \left[ (\langle \mathbf{w}_{t,I_t}, \mathbf{x}_t[S_{I_t}] \rangle - y_t)^2 - (\langle \mathbf{w}_{t,i}, \mathbf{x}_t[S_i] \rangle - y_t)^2 \right] + \\
& \sum_{t=1}^{T} \left[ (\langle \mathbf{w}_{t,i}, \mathbf{x}_t[S_i] \rangle - y_t)^2 - (\langle \mathbf{w}, \mathbf{x}_t[S_i] \rangle - y_t)^2 \right] \\
=& \underbrace{\sum_{t=1}^{T} \left[ c_{t,I_t} - c_{t,i} \right]}_{R_1} + \underbrace{\sum_{t=1}^{T} \left[ (\langle \mathbf{w}_{t,i}, \mathbf{x}_t[S_i] \rangle - y_t)^2 - (\langle \mathbf{w}, \mathbf{x}_t[S_i] \rangle - y_t)^2 \right]}_{R_2},
\end{aligned}
$$

where $c_{t,i} = \mathbf{w}_{t,i}^\top (\mathbf{x}_t \mathbf{x}_t^\top [S_i, S_i]) \mathbf{w}_{t,i} - 2 y_t \mathbf{w}_{t,i}^\top \mathbf{x}_t[S_i]$.

ANALYZING $R_1$

We first analyze an approximation of $R_1$, that is

$$
\sum_{t=1}^{T} \langle \mathbf{p}_t - \mathbf{u}_t, \mathbf{c}_t \rangle,
$$

where $\mathbf{u}_1, \mathbf{u}_2, \ldots, \mathbf{u}_T$ follow Lemma 6.

To start with, we analyze the instantaneous regret.

$$
\begin{aligned}
\langle \mathbf{p}_t - \mathbf{u}_t, \mathbf{c}_t \rangle =& \langle \mathbf{p}_t - \mathbf{u}_t, \tilde{\mathbf{c}}_t \rangle + \langle \mathbf{p}_t - \mathbf{u}_t, \mathbf{c}_t - \tilde{\mathbf{c}}_t \rangle \\
=& \underbrace{\langle \mathbf{p}_{t+1} - \mathbf{u}_t, \tilde{\mathbf{c}}_t \rangle}_{\Xi_{1,1}} + \underbrace{\langle \mathbf{p}_t - \mathbf{p}_{t+1}, \tilde{\mathbf{c}}_t \rangle}_{\Xi_{1,2}} + \langle \mathbf{p}_t - \mathbf{u}_t, \mathbf{c}_t - \tilde{\mathbf{c}}_t \rangle.
\end{aligned}
$$

Next we give an upper bound on $\Xi_{1,1}$.

$$
\begin{aligned}
\mathcal{B}_{\psi_t}(\mathbf{u}_t, \mathbf{p}_t) - \mathcal{B}_{\psi_t}(\mathbf{u}_t, \mathbf{p}_{t+1}) - \mathcal{B}_{\psi_t}(\mathbf{p}_{t+1}, \mathbf{p}_t) =& \langle \nabla \psi_t(\mathbf{p}_t) - \nabla \psi_t(\mathbf{p}_{t+1}), \mathbf{p}_{t+1} - \mathbf{u}_t \rangle \\
=& \langle \tilde{\mathbf{c}}_t + \nabla \psi_t(\mathbf{p}_t) - \nabla \psi_t(\mathbf{p}_{t+1}) - \tilde{\mathbf{c}}_t, \mathbf{p}_{t+1} - \mathbf{u}_t \rangle \\
\overset{(13)}{=}& \langle \tilde{\mathbf{c}}_t, \mathbf{p}_{t+1} - \mathbf{u}_t \rangle - \langle \nabla \Psi_t(\mathbf{p}_{t+1}), \mathbf{p}_{t+1} - \mathbf{u}_t \rangle \\
\geq& \langle \tilde{\mathbf{c}}_t, \mathbf{p}_{t+1} - \mathbf{u}_t \rangle, \quad \text{(Lemma 2)}
\end{aligned}
$$

where $\mathbf{p}_{t+1} \in \Delta_{t,N}$ and $\mathbf{u}_t \in \Delta_{t,N}$.

Similarly, $\Xi_{1,2}$ can be rewritten by

$$
\begin{aligned}
\mathcal{B}_{\psi_t}(\mathbf{p}_t, \mathbf{p}_t) - \mathcal{B}_{\psi_t}(\mathbf{p}_t, \mathbf{p}_{t+1}) - \mathcal{B}_{\psi_t}(\mathbf{p}_{t+1}, \mathbf{p}_t) =& \langle \nabla \psi_t(\mathbf{p}_t) - \nabla \psi_t(\mathbf{p}_{t+1}), \mathbf{p}_{t+1} - \mathbf{p}_t \rangle \\
=& \langle \tilde{\mathbf{c}}_t, \mathbf{p}_{t+1} - \mathbf{p}_t \rangle.
\end{aligned}
$$

Taking expectation w.r.t. $\{\tilde{B}_t, \hat{B}_t\}_{t=1}^{T}$ gives

$$
\begin{aligned}
& \mathbb{E}\left[ \sum_{t=1}^{T} \langle \mathbf{p}_t - \mathbf{u}_t, \mathbf{c}_t \rangle \right] \\
\leq& \mathbb{E}\left[ \sum_{t=1}^{T} \left[ \mathcal{B}_{\psi_t}(\mathbf{u}_t, \mathbf{p}_t) - \mathcal{B}_{\psi_t}(\mathbf{u}_t, \mathbf{p}_{t+1}) \right] \right] + \mathbb{E}\left[ \sum_{t=1}^{T} \mathcal{B}_{\psi_t}(\mathbf{p}_t, \mathbf{p}_{t+1}) \right] + \sum_{t=1}^{T} \mathbb{E}\left[ \langle \mathbf{p}_t - \mathbf{u}_t, \mathbf{c}_t - \tilde{\mathbf{c}}_t \rangle \right] \\
\leq& \mathbb{E}\left[ \frac{\ln\left(\frac{N}{\beta_T}\right)}{\eta_T} \right] + \mathbb{E}\left[ \sum_{t=1}^{T} \mathcal{B}_{\psi_t}(\mathbf{p}_t, \mathbf{p}_{t+1}) \right], \quad \text{(Lemma 8)}
\end{aligned}
$$

in which $\mathbb{E}[\tilde{\mathbf{c}}_t] = \mathbf{c}_t$. We further analyze the second term. Let

$$\nabla \psi_t(\tilde{\mathbf{p}}_{t+1}) = \nabla \psi_t(\mathbf{p}_t) - \tilde{\mathbf{c}}_t.$$

Then $\mathbf{p}_{t+1}$ can be equivalently defined as follows

$$\mathbf{p}_{t+1} = \arg\min_{\mathbf{p} \in \Delta_{t,N}} \mathcal{B}_{\psi_t}(\mathbf{p}, \tilde{\mathbf{p}}_{t+1}).$$

By Lemma 5, we can obtain

$$
\begin{aligned}
\mathbb{E}\left[\sum_{t=1}^{T} \mathcal{B}_{\psi_t}(\mathbf{p}_t, \mathbf{p}_{t+1})\right] \leq & \mathbb{E}\left[\sum_{t=1}^{T} \mathcal{B}_{\psi_t}(\mathbf{p}_t, \tilde{\mathbf{p}}_{t+1})\right] \\
\overset{(12)}{=} & \mathbb{E}\left[\sum_{t=1}^{T} \frac{1}{\eta_t}\left(\sum_{j=1}^{N} p_{t,j} \ln \frac{p_{t,j}}{\tilde{p}_{t+1,j}} - 1 + \sum_{j=1}^{N} \tilde{p}_{t+1,j}\right)\right] \\
= & \mathbb{E}\left[\sum_{t=1}^{T} \frac{1}{\eta_t}\left(\sum_{i=1}^{N} p_{t,j} \eta_t \tilde{c}_{t,j} - 1 + \sum_{j=1}^{N} p_{t,j} \exp(-\eta_t \tilde{c}_{t,j})\right)\right] \\
\leq & \mathbb{E}\left[\sum_{t=1}^{T} \frac{\sum_{j=1}^{N} p_{t,j}\left(1 - \eta_t \tilde{c}_{t,j} + \eta_t^2 \tilde{c}_{t,j}^2\right) - 1}{\eta_t} + \langle \mathbf{p}_t, \tilde{\mathbf{c}}_t \rangle\right] \\
\leq & \sum_{t=1}^{T} \eta_t \cdot \sum_{j=1}^{N} p_{t,j} \tilde{c}_{t,j}^2,
\end{aligned}
$$

where

$$
\eta_t = \frac{\sqrt{\ln N}}{\sqrt{\max_{\tau \leq t, j \in [N]} \tilde{c}_{\tau,j}^2 \ln N + \sum_{\tau=1}^{t} \sum_{j=1}^{N} p_{\tau,j} \tilde{c}_{\tau,j}^2}},
$$

$$
\tilde{c}_{t,j} = \left(\mathbf{w}_{t,j}^\top \tilde{\mathbf{x}}_t[S_j] - y_t\right) \cdot \left(\mathbf{w}_{t,j}^\top \hat{\mathbf{x}}_t[S_j] - y_t\right) - y_t^2.
$$

By Lemma 4, we have

$$
\begin{aligned}
& \mathbb{E}\left[\sum_{t=1}^{T}\langle \mathbf{p}_t - \mathbf{u}_t, \mathbf{c}_t \rangle\right] \\
\leq & 3 \frac{\ln(NT)}{\sqrt{\ln N}} \cdot \sqrt{\mathbb{E}\left[\max_{t \leq T, j \in [N]} \tilde{c}_{t,j}^2 \ln N + \sum_{t=1}^{T} \sum_{j=1}^{N} p_{t,j} \tilde{c}_{t,j}^2\right]} \\
\leq & 3 \frac{\ln(NT)}{\sqrt{\ln N}} \cdot \sqrt{(\mu_1 U X + Y)^4 \ln N + \sum_{t=1}^{T} \sum_{j=1}^{N} p_{t,j} \mathbb{E}\left[\tilde{c}_{t,j}^2\right]} \\
\leq & 3 \ln(NT)(\mu_1 U X + Y)^2 + 3 \frac{\ln(NT)}{\sqrt{\ln N}} \cdot \sqrt{\sum_{t=1}^{T} \sum_{j=1}^{N} p_{t,j}\left(U^4 \Xi_{t,j}^2 + 2 U^2 (U X + Y)^2 \Xi_{t,j} + c_{t,j}^2\right)},
\end{aligned}
$$

where $\mu_1 = \frac{4d - b' - 3b}{b' - b}$ and

$$
\mathbb{E}\left[\max_{t \leq T, j \in [N]} \tilde{c}_{t,j}^2\right] \leq \mathbb{E}\left[(\mu_1 U X + Y)^4\right] \leq (\mu_1 U X + Y)^4.
$$

By Lemma 7, taking expectation w.r.t. $\{I_t\}_{t=1}^{T}$ gives

$$
\begin{aligned}
\mathbb{E}[R_1] =& \mathbb{E}\left[\sum_{t=1}^{T}\langle \mathbf{p}_t, \mathbf{c}_t\rangle - \sum_{t=1}^{T} c_{t,i}\right] \\
=& \mathbb{E}\left[\sum_{t=1}^{T}\langle \mathbf{p}_t - \mathbf{u}_t, \mathbf{c}_t\rangle + \sum_{t=1}^{T}\langle \mathbf{u}_t, \mathbf{c}_t\rangle - \sum_{t=1}^{T} c_{t,i}\right] \\
\leq& 3\frac{\ln(NT)}{\sqrt{\ln N}}\cdot\sqrt{\sum_{t=1}^{T}\sum_{j=1}^{N} p_{t,j}\left(U^2\Xi_{t,j} + (UX+Y)^2\right)^2} + 3\ln(NT)\cdot(\mu_1 UX+Y)^2 + \\
& (UX+Y)^2\cdot\ln T.
\end{aligned}
$$

### ANALYZING $R_2$

Similar to the analysis of $R_1$, we can obtain

$$
\begin{aligned}
R_2 & \\
\leq& \sum_{t=1}^{T}\langle\nabla_{t,i}, \mathbf{w}_{t,i} - \mathbf{w}\rangle \\
=& \sum_{t=1}^{T}\langle\tilde{\nabla}_{t,i}, \mathbf{w}_{t,i} - \mathbf{w}\rangle + \sum_{t=1}^{T}\langle\mathbf{w}_{t,i} - \mathbf{w}, \nabla_{t,i} - \tilde{\nabla}_{t,i}\rangle \\
\leq& \sum_{t=1}^{T}\left[\mathcal{B}_{\psi_{t,i}}(\mathbf{w}, \mathbf{w}_{t,i}) - \mathcal{B}_{\psi_{t,i}}(\mathbf{w}, \mathbf{w}_{t+1,i})\right] + \sum_{t=1}^{T}\mathcal{B}_{\psi_{t,i}}(\mathbf{w}_{t,i}, \mathbf{w}_{t+1,i}) + \sum_{t=1}^{T}\langle\mathbf{w}_{t,i} - \mathbf{w}, \nabla_{t,i} - \tilde{\nabla}_{t,i}\rangle.
\end{aligned}
$$

Recalling that

$$
\lambda_{t,i} = \frac{U}{\sqrt{1 + \sum_{\tau=1}^{t}\|\tilde{\nabla}_{\tau,i}\|_2^2}}.
$$

By Lemma 3, we have

$$
\sum_{t=1}^{T}\mathcal{B}_{\psi_{t,i}}(\mathbf{w}_{t,i}, \mathbf{w}_{t+1,i}) = \sum_{t=1}^{T}\frac{1}{2\lambda_{t,i}}\|\mathbf{w}_{t,i} - \mathbf{w}_{t+1,i}\|_2^2 \leq \sum_{t=1}^{T}\frac{\lambda_{t,i}}{2}\|\tilde{\nabla}_{t,i}\|_2^2 \leq U\sqrt{\sum_{t=1}^{T}\|\tilde{\nabla}_{t,i}\|_2^2},
$$

and

$$
\begin{aligned}
\sum_{t=1}^{T}[\mathcal{B}_{\psi_{t,i}}(\mathbf{w}, \mathbf{w}_{t,i}) - \mathcal{B}_{\psi_{t,i}}(\mathbf{w}, \mathbf{w}_{t+1,i})] =& \sum_{t=1}^{T}\frac{1}{\lambda_{t,i}}\|\mathbf{w} - \mathbf{w}_{t,i}\|_2^2 - \frac{1}{\lambda_{t,i}}\|\mathbf{w} - \mathbf{w}_{t+1,i}\|_2^2 \\
\leq& \sum_{t=1}^{T-1}\|\mathbf{w} - \mathbf{w}_{t+1,i}\|_2^2\cdot\left(\frac{1}{\lambda_{t+1,i}} - \frac{1}{\lambda_{t,i}}\right) + \frac{\|\mathbf{w} - \mathbf{w}_{1,i}\|_2^2}{\lambda_1} \\
\leq& \frac{2U^2}{\lambda_{T,i}} \\
=& 2U\sqrt{1 + \sum_{\tau=1}^{T}\|\tilde{\nabla}_{\tau,i}\|_2^2}.
\end{aligned}
$$

Taking expectation w.r.t. $\tilde{B}_t, \hat{B}_t, t = 1, ..., T$ gives

$$
\mathbb{E}[R_2] \leq 3U\sqrt{\sum_{t=1}^{T}\mathbb{E}\left[\|\tilde{\nabla}_{t,i}\|_2^2\right]} + 2U \leq 3U\sqrt{\sum_{t=1}^{T}\left(4U^2\Xi_{t,i}^2 + 8(UX+Y)^2(\Xi_{t,i}+X^2)\right)} + 2U,
$$

where the last inequality comes from Lemma 1.

Combining the upper bound on $\mathbb{E}[R_1]$ and $\mathbb{E}[R_2]$ gives

$$\mathbb{E}[\text{Reg}(\mathbf{w})]$$

$$\leq 3U\sqrt{\sum_{t=1}^{T}\left(4U^2\Xi_{t,i}^2 + 8(UX+Y)^2(\Xi_{t,i}+X^2)\right)}+$$

$$\frac{\ln(NT)}{\sqrt{\ln N}}\sqrt{\sum_{t=1}^{T}\sum_{j=1}^{N}p_{t,j}\left(U^2\Xi_{t,j}+(UX+Y)^2\right)^2}+$$

$$3\ln(NT)\cdot(\mu UX+Y)^2+(UX+Y)^2\cdot\ln T + 2U$$

$$=O\left(\sqrt{\sum_{t=1}^{T}U^2\Xi_{t,i}^2+(UX+Y)^2X^2T}+\sqrt{U^4\sum_{t=1}^{T}\sum_{j=1}^{N}p_{t,j}\Xi_{t,j}^2+(UX+Y)^4T}\frac{\ln(NT)}{\sqrt{\ln N}}+\right.$$

$$\left.\frac{(d-b)^2}{(b'-b)^2}U^2X^2\ln(NT)+(UX+Y)^2\cdot\ln T\right)$$

$$=O\left(\sqrt{U^4\sum_{t=1}^{T}\sum_{j=1}^{N}p_{t,j}\Xi_{t,j}^2+(UX+Y)^4T}\frac{\ln(NT)}{\sqrt{\ln N}}+\right.$$

$$\left.\frac{(d-b)^2}{(b'-b)^2}U^2X^2\ln(NT)+(UX+Y)^2\cdot\ln T\right),$$

which concludes the proof. □

# F  PROOF OF THEOREM 3

## F.1  TECHNICAL LEMMAS

We first introduce some technical lemmas.

**Lemma 9** (Theorem 1 in Li et al. (2024)). *Assuming that* $l_{t,i}^{(j)}(\mathbf{w}) = \left(\mathbf{w}^\top\mathbf{x}_t^{(j)}[S_i]-y_t^{(j)}\right)^2$, $t \in [T], j \in [M]$. *Let* $\nabla_{t,i}^{(j)} = \nabla_{\mathbf{w}_{t,i}}l_{t,i}^{(j)}(\mathbf{w}_{t,i})$ *and* $\tilde{\nabla}_{t,i}^{(j)}$ *be an estimator of* $\nabla_{t,i}^{(j)}$. *At any round* $t \in [T]$, *let* $\mathbf{q}_{t+1}$ *and* $\mathbf{r}_{t+1}$ *be two auxiliary decisions defined as follows,*

$$\nabla_{\mathbf{q}_{t+1}}\psi_{t,i}(\mathbf{q}_{t+1}) = \nabla_{\mathbf{w}_{t,i}}\psi_{t,i}(\mathbf{w}_{t,i})-2\sum_{j=1}^{M}\frac{\tilde{\nabla}_{t,i}^{(j)}-\nabla_{t,i}^{(j)}}{M},$$

$$\nabla_{\mathbf{r}_{t+1}}\psi_{t,i}(\mathbf{r}_{t+1}) = \nabla_{\mathbf{w}_{t,i}}\psi_{t,i}(\mathbf{w}_{t,i})-\frac{2}{M}\sum_{j=1}^{M}\nabla_{t,i}^{(j)}.$$

*Then FedAMRO guarantees that,*

$$\forall\mathbf{w}\in\Omega,\quad \sum_{t=1}^{T}\sum_{j=1}^{M}\frac{l_{t,i}^{(j)}(\mathbf{w}_{t,i})-l_{t,i}^{(j)}(\mathbf{w})}{M}$$

$$\leq\sum_{t=1}^{T}\left[\mathcal{B}_{\psi_{t,i}}(\mathbf{w},\mathbf{w}_{t,i})-\mathcal{B}_{\psi_{t,i}}(\mathbf{w},\mathbf{w}_{t+1,i})\right]+\sum_{t=1}^{T}\frac{\mathcal{B}_{\psi_{t,i}}(\mathbf{w}_{t,i},\mathbf{r}_{t+1})}{2}+\sum_{t=1}^{T}\frac{\mathcal{B}_{\psi_{t,i}}(\mathbf{w}_{t,i},\mathbf{q}_{t+1})}{2}+$$

$$\frac{1}{M}\sum_{t=1}^{T}\sum_{j=1}^{M}\left\langle\tilde{\nabla}_{t,i}^{(j)}-\nabla_{t,i}^{(j)},\mathbf{w}_{t,i}-\mathbf{w}\right\rangle.$$

**Lemma 10.** *Assuming that* $l_t^{(j)}(\mathbf{p}) = \langle\mathbf{p},\mathbf{c}_t^{(j)}\rangle$, $t \in [T], j \in [M]$, *where* $\mathbf{p} \in \Delta_{t,N}$. *Let* $\tilde{\mathbf{c}}_t^{(j)}$ *be an estimator of* $\mathbf{c}_t^{(j)}$. *At any round* $t \in [T]$, *let* $\mathbf{q}_{t+1}$ *and* $\mathbf{r}_{t+1}$ *be two auxiliary decisions defined as*

*follows,*

$$\nabla_{\mathbf{q}_{t+1}} \psi_t(\mathbf{q}_{t+1}) = \nabla_{\mathbf{p}_t} \psi_t(\mathbf{p}_t) - 2 \sum_{j=1}^{M} \frac{\tilde{\mathbf{c}}_t^{(j)} - \mathbf{c}_t^{(j)}}{M},$$

$$\nabla_{\mathbf{r}_{t+1}} \psi_t(\mathbf{r}_{t+1}) = \nabla_{\mathbf{u}_t} \psi_t(\mathbf{p}_t) - \frac{2}{M} \sum_{j=1}^{M} \mathbf{c}_t^{(j)}.$$

*Then FedAMRO guarantees that,*

$$\forall \{\mathbf{u}_t \in \Delta_{t,N}\}_{t=1}^{T}, \quad \sum_{t=1}^{T} \sum_{j=1}^{M} \frac{l_t^{(j)}(\mathbf{p}_t^{(j)}) - l_t^{(j)}(\mathbf{u}_t)}{M}$$

$$\leq \sum_{t=1}^{T} \left[ \mathcal{B}_{\psi_t}(\mathbf{u}_t, \mathbf{p}_t) - \mathcal{B}_{\psi_t}(\mathbf{u}_t, \mathbf{p}_{t+1}) + \frac{\mathcal{B}_{\psi_t}(\mathbf{p}_t, \mathbf{r}_{t+1})}{2} \right] +$$

$$\sum_{t=1}^{T} \frac{\mathcal{B}_{\psi_t}(\mathbf{p}_t, \mathbf{q}_{t+1})}{2} + \frac{1}{M} \sum_{t=1}^{T} \sum_{j=1}^{M} \left\langle \mathbf{c}_t^{(j)} - \tilde{\mathbf{c}}_t^{(j)}, \mathbf{p}_t - \mathbf{u}_t \right\rangle.$$

*Proof of Lemma 10.* Let $\bar{\mathbf{c}}_t = \frac{1}{M} \sum_{j=1}^{M} \tilde{\mathbf{c}}_t^{(j)}$ and

$$\Psi_t(\mathbf{p}) = \langle \bar{\mathbf{c}}_t, \mathbf{p} \rangle + \mathcal{B}_{\psi_t}(\mathbf{p}, \mathbf{p}_t),$$
$$\mathbf{p}_{t+1} = \underset{\mathbf{p} \in \Delta_{t,N}}{\arg\min} \Psi_t(\mathbf{p}).$$

For any $\mathbf{u}_t \in \Delta_{t,N}$,

$$\langle \bar{\mathbf{c}}_t, \mathbf{p}_{t+1} - \mathbf{u}_t \rangle = \langle \nabla \Psi_t(\mathbf{p}_{t+1}) - \nabla \psi_t(\mathbf{p}_{t+1}) + \nabla \psi_t(\mathbf{p}_t), \mathbf{p}_{t+1} - \mathbf{u}_t \rangle$$
$$\leq \langle \nabla \psi_t(\mathbf{p}_t) - \nabla \psi_t(\mathbf{p}_{t+1}), \mathbf{p}_{t+1} - \mathbf{u}_t \rangle$$
$$= \mathcal{B}_{\psi_t}(\mathbf{u}_t, \mathbf{p}_t) - \mathcal{B}_{\psi_t}(\mathbf{u}_t, \mathbf{p}_{t+1}) - \mathcal{B}_{\psi_t}(\mathbf{p}_{t+1}, \mathbf{p}_t).$$

Then we give a lower bound.

$$\langle \bar{\mathbf{c}}_t, \mathbf{p}_{t+1} - \mathbf{u}_t \rangle$$

$$= \frac{1}{M} \sum_{j=1}^{M} \left[ \left\langle \mathbf{c}_t^{(j)}, \mathbf{p}_{t+1} - \mathbf{u}_t \right\rangle + \left\langle \tilde{\mathbf{c}}_t^{(j)} - \mathbf{c}_t^{(j)}, \mathbf{p}_{t+1} - \mathbf{u}_t \right\rangle \right]$$

$$= \frac{1}{M} \sum_{j=1}^{M} \left\langle \mathbf{c}_t^{(j)}, \mathbf{p}_t^{(j)} - \mathbf{u}_t \right\rangle + \underbrace{\frac{1}{M} \sum_{j=1}^{M} \left\langle \mathbf{c}_t^{(j)}, \mathbf{p}_{t+1} - \mathbf{p}_t \right\rangle}_{\Xi_1} + \underbrace{\frac{1}{M} \sum_{j=1}^{M} \left\langle \tilde{\mathbf{c}}_t^{(j)} - \mathbf{c}_t^{(j)}, \mathbf{p}_{t+1} - \mathbf{u}_t \right\rangle}_{\Xi_2},$$

where $\mathbf{p}_t^{(j)} = \mathbf{p}_t$.

Next we analyze $\Xi_1$ and $\Xi_2$.

To analyze $\Xi_1$, we introduce an auxiliary variable $\mathbf{r}_{t+1}$ defined as follows

$$\nabla_{\mathbf{r}_{t+1}} \psi_t(\mathbf{r}_{t+1}) = \nabla_{\mathbf{p}_t} \psi_t(\mathbf{p}_t) - \frac{2}{M} \sum_{j=1}^{M} \mathbf{c}_t^{(j)}.$$

Then we have

$$\Xi_1 = \frac{1}{2} \left\langle \frac{2}{M} \sum_{j=1}^{M} \mathbf{c}_t^{(j)}, \mathbf{p}_{t+1} - \mathbf{p}_t \right\rangle$$

$$= \frac{1}{2} \left\langle \nabla_{\mathbf{u}_t} \psi_t(\mathbf{p}_t) - \nabla_{\mathbf{r}_{t+1}} \psi_t(\mathbf{r}_{t+1}), \mathbf{p}_{t+1} - \mathbf{p}_t \right\rangle$$

$$= \frac{\mathcal{B}_{\psi_t}(\mathbf{p}_{t+1}, \mathbf{r}_{t+1}) - \mathcal{B}_{\psi_t}(\mathbf{p}_{t+1}, \mathbf{p}_t) - \mathcal{B}_{\psi_t}(\mathbf{p}_t, \mathbf{r}_{t+1})}{2}$$

$$\geq -\frac{1}{2} \left( \mathcal{B}_{\psi_t}(\mathbf{p}_{t+1}, \mathbf{p}_t) + \mathcal{B}_{\psi_t}(\mathbf{p}_t, \mathbf{r}_{t+1}) \right).$$

Before analyzing $\Xi_2$, we also introduce an auxiliary variable $\mathbf{q}_{t+1}$ defined as follows

$$\nabla_{\mathbf{q}_{t+1}}\psi_t(\mathbf{q}_{t+1}) = \nabla_{\mathbf{P}_t}\psi_t(\mathbf{p}_t) - \frac{2}{M}\sum_{j=1}^{M}\left(\tilde{\mathbf{c}}_t^{(j)} - \mathbf{c}_t^{(j)}\right).$$

Now we can analyze $\Xi_2$. We have

$$\Xi_2 = \frac{1}{2}\left\langle \frac{2}{M}\sum_{j=1}^{M}\left(\tilde{\mathbf{c}}_t^{(j)} - \mathbf{c}_t^{(j)}\right), \mathbf{p}_{t+1} - \mathbf{p}_t \right\rangle + \underbrace{\left\langle \frac{1}{M}\sum_{j=1}^{M}\left(\tilde{\mathbf{c}}_t^{(j)} - \mathbf{c}_t^{(j)}\right), \mathbf{p}_t - \mathbf{u}_t \right\rangle}_{\Xi_3}$$

$$= \frac{1}{2}\left\langle \nabla_{\mathbf{p}_t}\psi_t(\mathbf{p}_t) - \nabla_{\mathbf{q}_{t+1}}\psi_t(\mathbf{q}_{t+1}), \mathbf{p}_{t+1} - \mathbf{p}_t \right\rangle + \Xi_3$$

$$= \frac{\mathcal{B}_{\psi_t}(\mathbf{p}_{t+1}, \mathbf{q}_{t+1}) - \mathcal{B}_{\psi_t}(\mathbf{p}_{t+1}, \mathbf{p}_t) - \mathcal{B}_{\psi_t}(\mathbf{p}_t, \mathbf{q}_{t+1})}{2} + \Xi_3$$

$$\geq -\frac{1}{2}\left(\mathcal{B}_{\psi_t}(\mathbf{p}_{t+1}, \mathbf{p}_t) + \mathcal{B}_{\psi_t}(\mathbf{p}_t, \mathbf{q}_{t+1})\right) + \Xi_3.$$

Combining the lower bound and upper bound gives

$$\sum_{j=1}^{M}\frac{\left\langle \mathbf{c}_t^{(j)}, \mathbf{p}_t^{(j)} - \mathbf{u}_t \right\rangle}{M} \leq \mathcal{B}_{\psi_t}(\mathbf{u}_t, \mathbf{p}_t) - \mathcal{B}_{\psi_t}(\mathbf{u}_t, \mathbf{p}_{t+1}) - \Xi_3 + \frac{\mathcal{B}_\psi(\mathbf{p}_t, \mathbf{q}_{t+1})}{2} + \frac{\mathcal{B}_\psi(\mathbf{p}_t, \mathbf{r}_{t+1})}{2}.$$

Using the convexity of $l_t^{(j)}$ concludes the proof. $\qquad\square$

Next we give a similar result with Lemma 1.

**Lemma 11.** *Let*
$$\tilde{\nu}_t^{(j)} := (\tilde{\nu}_{t,m}^{(j)})_{m \in S_{I_t^{(j)}}}, \quad \hat{\nu}_t^{(j)} := (\hat{\nu}_{t,m}^{(j)})_{m \in S_{I_t^{(j)}}}. \tag{14}$$

*Assuming that $\max_{f,(\mathbf{x},y)} \ell(f(\mathbf{x}), y) \leq C$. Let $\mu_2 = \frac{2d - b' - b}{b' - b}$. For any $i \in [N]$,*

$$\mathbb{E}\left[\tilde{c}_{t,i}^{(j)}\right] = c_{t,i}^{(j)},$$

$$\mathbb{E}\left[\tilde{\nabla}_{t,i}^{(j)}\right] = \nabla_{t,i}^{(j)},$$

$$\mathbb{E}\left[(\tilde{c}_{t,i}^{(j)})^2\right] \leq 16\mu_2^2 U^4 X^4 + 8\mu_2 U^2 X^2 C + (C - Y^2)^2,$$

$$\mathbb{E}\left[\left\|\tilde{\nabla}_{t,i}^{(j)}\right\|_2^2\right] \leq 32\mu_2^2 U^2 X^4 + (2\mu_2 + 1)X^2 C.$$

*The expectation is taken on the elements in $\tilde{B}_t^{(j)}, \hat{B}_t^{(j)}, \tilde{\nu}_t^{(j)}$ and $\hat{\nu}_t^{(j)}$.*

*Proof of Lemma 11.* For any $I_t^{(j)} \in [N]$, we have,

$$\forall m \in S_{I_t^{(j)}}, \ \mathbb{E}\left[\tilde{x}_{t,m}^{(j)}\right] = \mathbb{P}[\tilde{v}_{t,m}^{(j)} = 1]\left(\frac{x_{t,m}^{(j)} - \delta_{t,m}^{(j)}}{\mathbb{P}[\tilde{v}_{t,m}^{(j)} = 1]} + \delta_{t,m}^{(j)}\right) + \left(1 - \mathbb{P}[\tilde{v}_{t,m}^{(j)} = 1]\right)\delta_{t,m}^{(j)} = x_{t,m}^{(j)},$$

$$\forall m \notin S_{I_t^{(j)}}, \ \mathbb{E}\left[\tilde{x}_{t,m}^{(j)}\right] = x_{t,m}^{(j)}.$$

Thus $\mathbb{E}\left[\tilde{\mathbf{x}}_t^{(j)}\right] = \mathbf{x}_t^{(j)}$. Similarly, it can be proved that $\mathbb{E}\left[\hat{\mathbf{x}}_t^{(j)}\right] = \mathbf{x}_t^{(j)}$. Next we show that the estimators are unbiased.

$$\mathbb{E}\left[\tilde{c}_{t,i}^{(j)}\right] = \mathop{\mathbb{E}}_{\tilde{B}_t^{(j)}, \tilde{\nu}_t^{(j)}}\left[\left(\mathbf{w}_{t,i}^\top \tilde{\mathbf{x}}_t^{(j)}[S_i] - y_t^{(j)}\right)\right] \cdot \mathop{\mathbb{E}}_{\hat{B}_t^{(j)}, \hat{\nu}_t^{(j)}}\left[\left(\mathbf{w}_{t,i}^\top \hat{\mathbf{x}}_t^{(j)}[S_i] - y_t^{(j)}\right)\right] - y_t^{(j)} \cdot y_t^{(j)}$$

$$= \left(\mathbf{w}_{t,i}^\top \mathbf{x}_t^{(j)}[S_i] - y_t^{(j)}\right)^2 - y_t^{(j)} \cdot y_t^{(j)}$$

$$= c_{t,i}^{(j)},$$

where the first equality comes from that $\tilde{\mathbf{x}}_t^{(j)}$ is independent of $\hat{\mathbf{x}}_t^{(j)}$. Similarly, we can prove

$$\mathbb{E}\left[\tilde{\nabla}_{t,i}^{(j)}\right] = 2\mathbb{E}\left[(\mathbf{w}_{t,i}^\top \hat{\mathbf{x}}_t^{(j)}[S_i] - y_t^{(j)}) \cdot \hat{\mathbf{x}}_t^{(j)}[S_i]\right] = \nabla_{t,i}^{(j)}.$$

Then we analyze the second order moment.

$$\mathbb{E}\left[(\tilde{c}_{t,i}^{(j)})^2\right] = \underset{\tilde{B}_t^{(j)},\tilde{\nu}_t^{(j)}}{\mathbb{E}}\left[\left(\mathbf{w}_{t,i}^\top \tilde{\mathbf{x}}_t^{(j)}[S_i] - y_t^{(j)}\right)^2\right] \cdot \underset{\hat{B}_t^{(j)},\hat{\nu}_t^{(j)}}{\mathbb{E}}\left[\left(\mathbf{w}_{t,i}^\top \hat{\mathbf{x}}_t^{(j)}[S_i] - y_t^{(j)}\right)^2\right] -$$
$$2(\mathbf{w}_{t,i}^\top \mathbf{x}_t^{(j)}[S_i] - y_t^{(j)})^2 (y_t^{(j)})^2 + (y_t^{(j)})^4.$$

For simplicity, let

$$\varepsilon_{t,m}^{(j)} = x_{t,m}^{(j)} - \delta_{t,m}^{(j)}.$$

Then

$$\underset{\tilde{B}_t^{(j)},\tilde{\nu}_t^{(j)}}{\mathbb{E}}\left[\left(\mathbf{w}_{t,i}^\top \tilde{\mathbf{x}}_t^{(j)}[S_i] - y_t^{(j)}\right)^2\right]$$

$$= \underset{\tilde{B}_t^{(j)},\tilde{\nu}_t^{(j)}}{\mathbb{E}}\left[\sum_{m\in S_i} w_{t,i,m}^2 \cdot \tilde{x}_{t,m}^{(j)} \cdot \tilde{x}_{t,m}^{(j)}\right] + \underset{\tilde{B}_t^{(j)},\tilde{\nu}_t^{(j)}}{\mathbb{E}}\left[\sum_{m\neq n\in S_i} w_{t,i,m} w_{t,i,n} \tilde{x}_{t,m}^{(j)} \tilde{x}_{t,n}^{(j)}\right] -$$

$$2y_t^{(j)} \underset{\tilde{B}_t^{(j)},\tilde{\nu}_t^{(j)}}{\mathbb{E}}\left[\mathbf{w}_{t,i}^\top \tilde{\mathbf{x}}_t^{(j)}[S_i]\right] + y_t^{(j)} \cdot y_t^{(j)}$$

$$\leq \sum_{m\in S_i} w_{t,i,m}^2 \left[\frac{2(d-b)}{b'-b}(\varepsilon_{t,m}^{(j)})^2 + 2\delta_{t,m}^{(j)}\varepsilon_{t,m}^{(j)} + (\delta_{t,m}^{(j)})^2\right] +$$

$$\sum_{m\neq n\in S_i} w_{t,i,m} w_{t,i,n} x_{t,m}^{(j)} x_{t,n}^{(j)} - 2y_t^{(j)} \mathbf{w}_{t,i}^\top \mathbf{x}_t^{(j)}[S_i] + (y_t^{(j)})^2$$

$$= \frac{2d-b'-b}{b'-b} \sum_{m\in S_i} w_{t,i,m}^2 \cdot (\varepsilon_{t,m}^{(j)})^2 + \left(\mathbf{w}_{t,i}^\top \mathbf{x}_t^{(j)}[S_i] - y_t^{(j)}\right)^2$$

$$\leq 4\frac{2d-b'-b}{b'-b} U^2 X^2 + \left(\mathbf{w}_{t,i}^\top \mathbf{x}_t^{(j)}[S_i] - y_t^{(j)}\right)^2$$

$$\leq 4\frac{2d-b'-b}{b'-b} U^2 X^2 + C.$$

Thus

$$\mathbb{E}\left[(\tilde{c}_{t,i}^{(j)})^2\right] \leq \left(4\frac{2d-b'-b}{b'-b} U^2 X^2 + C\right)^2 - 2CY^2 + Y^4$$

$$= 16\frac{(2d-b'-b)^2}{(b'-b)^2} U^4 X^4 + 8\frac{2d-b'-b}{b'-b} U^2 X^2 C + (C-Y^2)^2.$$

The analysis for the gradient is similar.

$$\mathbb{E}\left[\left\|\tilde{\nabla}_{t,i}^{(j)}\right\|_2^2\right] \leq 4\mathbb{E}\left[\left\|(\mathbf{w}_{t,i}^\top \tilde{\mathbf{x}}_t^{(j)}[S_i] - y_t^{(j)})\hat{\mathbf{x}}_t^{(j)}[S_i]\right\|_2^2\right]$$

$$= 4\underset{\tilde{B}_t^{(j)},\tilde{\nu}_t^{(j)}}{\mathbb{E}}\left[\left(\mathbf{w}_{t,i}^\top \tilde{\mathbf{x}}_t^{(j)}[S_i] - y_t^{(j)}\right)^2\right] \cdot \underset{\hat{B}_t^{(j)},\hat{\nu}_t^{(j)}}{\mathbb{E}}\left[\|\hat{\mathbf{x}}_t^{(j)}[S_i]\|_2^2\right]$$

$$\leq 4\left(4\frac{2d-b'-b}{b'-b} U^2 X^2 + C\right) \cdot \underset{\hat{B}_t^{(j)},\hat{\nu}_t^{(j)}}{\mathbb{E}}\left[\|\hat{\mathbf{x}}_t^{(j)}[S_i]\|_2^2\right]$$

$$= 4\left(4\frac{2d-b'-b}{b'-b} U^2 X^2 + C\right) \cdot \left(2\frac{2d-b'-b}{b'-b} + 1\right) X^2$$

$$= 32\frac{(2d-b'-b)^2}{(b'-b)^2} U^2 X^4 + \left(2\frac{2d-b'-b}{b'-b} + 1\right) X^2 C,$$

which concludes the proof. $\qquad\square$

*Proof of Theorem 2.* We decompose the regret as follows

$$\text{Reg}(\mathbf{w}) = \sum_{j=1}^{M} \sum_{t=1}^{T} \left(\hat{y}_t^{(j)} - y_t^{(j)}\right)^2 - \sum_{j=1}^{M} \sum_{t=1}^{T} \left(\langle \mathbf{w}, \mathbf{x}_t^{(j)}[S_i]\rangle - y_t^{(j)}\right)^2$$

$$= \sum_{j=1}^{M} \sum_{t=1}^{T} \left[\left(\hat{y}_t^{(j)} - y_t\right)^2 - \left(\mathbf{w}_{t,i}^\top \mathbf{x}_t^{(j)}[S_i] - y_t^{(j)}\right)^2\right] +$$

$$\sum_{j=1}^{M} \sum_{t=1}^{T} \left[\left(\langle \mathbf{w}_{t,i}, \mathbf{x}_t^{(j)}[S_i]\rangle - y_t\right)^2 - (\langle \mathbf{w}, \mathbf{x}_t[S_i]\rangle - y_t)^2\right]$$

$$= \underbrace{\sum_{j=1}^{M} \sum_{t=1}^{T} \left[c_{t,I_t}^{(j)} - c_{t,i}^{(j)}\right]}_{R_1} + \underbrace{\sum_{j=1}^{M} \sum_{t=1}^{T} \left[\left(\mathbf{w}_{t,i}^\top \mathbf{x}_t[S_i] - y_t\right)^2 - \left(\mathbf{w}^\top \mathbf{x}_t[S_i] - y_t\right)^2\right]}_{R_2},$$

where $\hat{y}_t^{(j)} = \left(\mathbf{w}_{t,I_t}^\top \mathbf{x}_t^{(j)}[S_{I_t}] - y_t\right)^2$. Next we separately analyze $R_1$ and $R_2$.

## F.2 Analyzing $R_1$

Let $\mathbf{u} = (u_1, ..., u_d)$ in which $u_i = 1$ and $u_k = 0$ for all $k \neq i$. Let $\mathbf{u}_t = (u_{t,1}, ..., u_{t,d})$ where $u_{t,k} = N^{-1}\beta_t$ for all $k \neq i$ and $u_{t,i} = 1 - \frac{N-1}{N}\beta_t$. Taking expectation w.r.t. $\{\tilde{\nu}_t^{(j)}, \hat{\nu}_t^{(j)}, \tilde{B}_t^{(j)}, \hat{B}_t^{(j)}\}_{j \in [M], t \in [T]}$ (see (14)) yields

$$\frac{1}{M}\mathbb{E}\left[\sum_{j=1}^{M} \sum_{t=1}^{T} \langle \mathbf{p}_t - \mathbf{u}, \mathbf{c}_t^{(j)}\rangle\right]$$

$$= \mathbb{E}\left[\sum_{j=1}^{M} \sum_{t=1}^{T} \frac{\langle \mathbf{p}_t - \mathbf{u}_t, \mathbf{c}_t^{(j)}\rangle}{M}\right] + \mathbb{E}\left[\sum_{j=1}^{M} \sum_{t=1}^{T} \frac{\langle \mathbf{u}_t - \mathbf{u}, \mathbf{c}_t^{(j)}\rangle}{M}\right]$$

(Lemma 10 + Lemma 7)

$$\leq \mathbb{E}\left[\sum_{t=1}^{T} \mathcal{B}_{\psi_t}(\mathbf{u}_t, \mathbf{p}_t) - \mathcal{B}_{\psi_t}(\mathbf{u}_t, \mathbf{p}_{t+1}) + \frac{\mathcal{B}_{\psi_t}(\mathbf{p}_t, \mathbf{r}_{t+1})}{2}\right] + \mathbb{E}\left[\sum_{t=1}^{T} \frac{\mathcal{B}_{\psi_t}(\mathbf{p}_t, \mathbf{q}_{t+1})}{2}\right] + \max_{t,j,i} c_{t,i}^{(j)} \cdot \ln T$$

(Lemma 8)

$$\leq \ln(NT)\mathbb{E}\left[\frac{1}{\eta_T}\right] + \frac{1}{2}\mathbb{E}\left[\sum_{t=1}^{T} \mathcal{B}_{\psi_t}(\mathbf{p}_t, \mathbf{r}_{t+1})\right] + \frac{1}{2}\mathbb{E}\left[\sum_{t=1}^{T} \mathcal{B}_{\psi_t}(\mathbf{p}_t, \mathbf{q}_{t+1})\right] + (UX + Y)^2 \cdot \ln T,$$

where the first inequality also uses the following fact,

$$\frac{1}{M}\mathbb{E}\left[\sum_{t=1}^{T} \sum_{j=1}^{M} \left\langle \mathbf{c}_t^{(j)} - \tilde{\mathbf{c}}_t^{(j)}, \mathbf{p}_t - \mathbf{u}_t\right\rangle\right] = 0.$$

We just need to analyze $\mathbb{E}\left[\sum_{t=1}^{T} \mathcal{B}_{\psi_t}(\mathbf{p}_t, \mathbf{r}_{t+1})\right]$ and $\mathbb{E}\left[\mathcal{B}_{\psi_t}(\mathbf{p}_t, \mathbf{q}_{t+1})\right]$. By Lemma 10, we introduce two new notations

$$\mathbf{v}_t = \frac{2}{M}\sum_{j=1}^{M} \mathbf{c}_t^{(j)}, \quad \mathbf{z}_t = 2\sum_{j=1}^{M} \frac{\tilde{\mathbf{c}}_t^{(j)} - \mathbf{c}_t^{(j)}}{M}.$$

It is obvious that

$$-2Y^2 \leq v_{t,i} \leq 2C - 2Y^2,$$
$$-2(\mu_1^2 U^2 X^2 + 2\mu_1 UXY) - 2C + 2Y^2 \leq z_{t,i},$$
$$z_{t,i} \leq 2(\mu_1^2 U^2 X^2 + 2\mu_1 UXY),$$

where $\mu_1 = \frac{4d-b'-3b}{b'-b}$. We define $\eta_t$ as follows

$$\eta_t = \frac{\sqrt{\ln N}}{\sqrt{\xi_1 \cdot \ln N + \alpha \cdot t}},$$

$$\alpha = \frac{(C-Y^2)^2}{4} + \frac{2\mu_2^2 U^4 X^4 + \mu_2 U^2 X^2 C + \frac{(C-Y^2)^2}{8}}{M},$$

$$\xi_1 = 4\left(C - Y^2 + \mu_1^2 U^2 X^2 + 2\mu_1 UXY\right)^2.$$

We further analyze

$$\mathbb{E}\left[\sum_{t=1}^{T} \mathcal{B}_{\psi_t}(\mathbf{p}_t, \mathbf{r}_{t+1})\right] \overset{(12)}{=} \mathbb{E}\left[\sum_{t=1}^{T} \frac{1}{\eta_t}\left(\sum_{i=1}^{N} p_{t,i} \ln \frac{p_{t,i}}{r_{t+1,i}} - 1 + \sum_{i=1}^{N} r_{t+1,i}\right)\right]$$

$$= \mathbb{E}\left[\sum_{t=1}^{T} \frac{1}{\eta_t}\left(\sum_{i=1}^{N} p_{t,i}\eta_t v_{t,j} - 1 + \sum_{i=1}^{N} p_{t,i}\exp(-\eta_t v_{t,j})\right)\right]$$

(Lemma 5)

$$\leq \mathbb{E}\left[\sum_{t=1}^{T} \frac{\sum_{i=1}^{N} p_{t,i}\left(1 - \eta_t v_{t,i} + \eta_t^2 v_{t,i}^2\right) - 1}{\eta_t} + \langle \mathbf{p}_t, \mathbf{v}_t\rangle\right]$$

$$= \mathbb{E}\left[\sum_{t=1}^{T}\sum_{i=1}^{N} p_{t,i}\eta_t v_{t,i}^2\right]$$

$$\leq \sum_{t=1}^{T} 4(C-Y^2)^2 \eta_t,$$

and

$$\mathbb{E}\left[\sum_{t=1}^{T} \mathcal{B}_{\psi_t}(\mathbf{p}_t, \mathbf{q}_{t+1})\right] \overset{(12)}{=} \mathbb{E}\left[\sum_{t=1}^{T} \frac{1}{\eta_t}\left(\sum_{i=1}^{N} p_{t,i} \ln \frac{p_{t,i}}{q_{t+1,i}} - 1 + \sum_{i=1}^{N} q_{t+1,i}\right)\right]$$

$$= \mathbb{E}\left[\sum_{t=1}^{T} \frac{1}{\eta_t}\left(\sum_{i=1}^{N} p_{t,i}\eta_t z_{t,i} - 1 + \sum_{i=1}^{N} p_{t,i}\exp(-\eta_t z_{t,i})\right)\right]$$

(Lemma 5)

$$\leq \mathbb{E}\left[\sum_{t=1}^{T} \frac{\sum_{i=1}^{N} p_{t,i}\left(1 - \eta_t z_{t,i} + \eta_t^2 z_{t,i}^2\right) - 1}{\eta_t} + \langle \mathbf{p}_t, \mathbf{z}_t\rangle\right]$$

$$= \mathbb{E}\left[\sum_{t=1}^{T} \eta_t \sum_{i=1}^{N} p_{t,i} z_{t,i}^2\right]$$

$$= \mathbb{E}\left[\sum_{t=1}^{T} \eta_t \cdot \sum_{i=1}^{N} p_{t,i}\frac{4}{M^2}\left(\sum_{j=1}^{M}(\tilde{c}_{t,i}^{(j)} - c_{t,i}^{(j)})^2\right)\right] +$$

$$\sum_{t=1}^{T} \mathbb{E}\left[\eta_t \cdot \sum_{i=1}^{N} \frac{4p_{t,i}}{M^2}\left(\sum_{k\neq j}(\tilde{c}_{t,i}^{(j)} - c_{t,i}^{(j)})(\tilde{c}_{t,i}^{(k)} - c_{t,i}^{(k)})\right)\right]$$

$$= \mathbb{E}\left[\sum_{t=1}^{T} \eta_t \cdot \sum_{i=1}^{N} p_{t,i}\frac{4}{M^2}\left(\sum_{j=1}^{M}\left(\tilde{c}_{t,i}^{(j)}\right)^2 - \left(c_{t,i}^{(j)}\right)^2\right)\right]$$

(Lemma 11)

$$\leq \sum_{t=1}^{T} \eta_t \frac{16\mu_2^2 U^4 X^4 + 8\mu_2 U^2 X^2 C + (C-Y^2)^2}{M}.$$

Substituting into the two upper bounds yields

$$
\frac{1}{M}\mathbb{E}\left[\sum_{j=1}^{M}\sum_{t=1}^{T}\langle \mathbf{p}_t - \mathbf{u}, \mathbf{c}_t^{(j)}\rangle\right] \leq \mathbb{E}\left[\frac{\ln(NT)}{\eta_T} + 8\sum_{t=1}^{T}\eta_t \alpha\right] + (UX+Y)^2 \cdot \ln T
$$

$$
\text{(Lemma 3)}
$$

$$
\leq \frac{\ln(NT)}{\sqrt{\ln N}}\sqrt{\xi_1 \ln N + \alpha T} + 16\sqrt{\alpha T \ln N} + (UX+Y)^2 \cdot \ln T
$$

$$
\leq \sqrt{\xi_1}\ln(NT) + (UX+Y)^2 \ln T + 17\frac{\ln(NT)}{\sqrt{\ln N}}\sqrt{\alpha T}.
$$

Taking expectation w.r.t. $\{I_t^{(j)}\}_{t\leq T, j\in[M]}$ yields

$$
\frac{1}{M}\mathbb{E}[R_1] = \frac{1}{M}\mathbb{E}\left[\sum_{j=1}^{M}\sum_{t=1}^{T}\langle \mathbf{p}_t - \mathbf{u}, \mathbf{c}_t^{(j)}\rangle\right] \leq \sqrt{\xi_1}\ln(NT) + (UX+Y)^2 \ln T + 17\frac{\ln(NT)}{\sqrt{\ln N}}\sqrt{\alpha T}.
$$

### F.3 Analyzing $R_2$

Taking expectation w.r.t. $\{\tilde{\nu}_t^{(j)}, \hat{\nu}_t^{(j)}, \tilde{B}_t^{(j)}, \tilde{B}_t^{(j)}\}_{t\in[T], j\in[M]}$ gives

$$
\frac{1}{M}\mathbb{E}[R_2]
$$
$$
\leq \mathbb{E}\left[\sum_{t=1}^{T}\left[\mathcal{B}_{\psi_{t,i}}(\mathbf{w}, \mathbf{w}_{t,i}) - \mathcal{B}_{\psi_{t,i}}(\mathbf{w}, \mathbf{w}_{t+1,i})\right]\right] + \mathbb{E}\left[\sum_{t=1}^{T}\frac{\mathcal{B}_{\psi_t}(\mathbf{w}_{t,i}, \mathbf{r}_{t+1})}{2} + \sum_{t=1}^{T}\frac{\mathcal{B}_{\psi_t}(\mathbf{w}_{t,i}, \mathbf{q}_{t+1})}{2}\right] +
$$
$$
\frac{1}{M}\mathbb{E}\left[\sum_{t=1}^{T}\sum_{j=1}^{M}\left\langle \tilde{\nabla}_{t,i}^{(j)} - \nabla_{t,i}^{(j)}, \mathbf{w}_{t,i} - \mathbf{w}\right\rangle\right] \quad \text{(Lemma 9)}
$$
$$
\leq \mathbb{E}\left[\frac{2U^2}{\lambda_{T,i}} + \sum_{t=1}^{T}\frac{\mathcal{B}_{\psi_t}(\mathbf{w}_{t,i}, \mathbf{r}_{t+1})}{2} + \frac{\mathcal{B}_{\psi_t}(\mathbf{w}_{t,i}, \mathbf{q}_{t+1})}{2}\right].
$$

We define $\lambda_{t,i}$ as follows

$$
\lambda_{t,i} = \frac{U}{\sqrt{\frac{CX^2}{4} + \frac{2\mu_2^2 U^2 X^4 + \frac{(2\mu_2+1)X^2 C}{16}}{M}} \cdot \sqrt{t}},
$$
$$
\mu_2 = \frac{2d - b' - b}{b' - b}.
$$

We further have

$$\mathbb{E}\left[\sum_{t=1}^{T}\mathcal{B}_{\psi_{t,i}}(\mathbf{w}_{t,i},\mathbf{r}_{t+1}) + \sum_{t=1}^{T}\mathcal{B}_{\psi_{t,i}}(\mathbf{w}_{t,i},\mathbf{r}_{t+1})\right]$$

$$=\mathbb{E}\left[\sum_{t=1}^{T}\frac{\|\mathbf{w}_{t,i}-\mathbf{r}_{t+1}\|_2^2}{2\lambda_{t,i}} + \sum_{t=1}^{T}\frac{\|\mathbf{w}_{t,i}-\mathbf{q}_{t+1}\|_2^2}{2\lambda_{t,i}}\right]$$

$$=2\mathbb{E}\left[\sum_{t=1}^{T}\lambda_{t,i}\left[\left\|\frac{1}{M}\sum_{j=1}^{M}\nabla_{t,i}^{(j)}\right\|_2^2 + \left\|\sum_{j=1}^{M}\frac{\tilde{\nabla}_{t,i}^{(j)}-\nabla_{t,i}^{(j)}}{M}\right\|_2^2\right]\right]$$

$$\leq 2\mathbb{E}\left[\sum_{t=1}^{T}\lambda_{t,i}\left[\sum_{j=1}^{M}\frac{\left\|\nabla_{t,i}^{(j)}\right\|_2^2}{M} + \sum_{j=1}^{M}\frac{\left\|\tilde{\nabla}_{t,i}^{(j)}-\nabla_{t,i}^{(j)}\right\|_2^2}{M^2}\right]\right] +$$

$$2\sum_{t=1}^{T}\mathbb{E}\left[\lambda_{t,i}\cdot\frac{1}{M^2}\sum_{j\neq k}\left(\tilde{\nabla}_{t,i}^{(j)}-\nabla_{t,i}^{(j)}\right)\left(\tilde{\nabla}_{t,i}^{(k)}-\nabla_{t,i}^{(k)}\right)\right]$$

$$\leq 2\mathbb{E}\left[\sum_{t=1}^{T}\lambda_{t,i}\left(\frac{1}{M}\sum_{j=1}^{M}\left\|\nabla_{t,i}^{(j)}\right\|_2^2 + \frac{1}{M^2}\sum_{j=1}^{M}\left\|\tilde{\nabla}_{t,i}^{(j)}\right\|_2^2\right)\right]$$

(Lemma 11)

$$\leq 32\sum_{t=1}^{T}\lambda_{t,i}\left(\frac{CX^2}{4} + \frac{2\mu_2^2U^2X^4 + \frac{(2\mu_2+1)X^2C}{16}}{M}\right)$$

(Lemma 3)

$$\leq 64U\sqrt{\frac{CX^2}{4} + \frac{2\mu_2^2U^2X^4 + \frac{(2\mu_2+1)X^2C}{16}}{M}}\cdot\sqrt{T}.$$

Substituting into the upper bound of $\mathbb{E}[R_2]$ yields

$$\frac{1}{M}\mathbb{E}[R_2] \leq \frac{2U^2}{\lambda_{T,i}} + 64U\sqrt{\frac{CX^2}{4} + \frac{2\mu_2^2U^2X^4 + \frac{(2\mu_2+1)X^2C}{16}}{M}}\cdot\sqrt{T}$$

$$\leq 66U\sqrt{\frac{CX^2}{4} + \frac{2\mu_2^2U^2X^4 + \frac{(2\mu_2+1)X^2C}{16}}{M}}\cdot\sqrt{T}.$$

### F.4 COMBINING $R_1$ AND $R_2$

Taking expectation w.r.t. $\{I_t^{(j)}\}_{t\leq T,j\in[M]}$ yields

$$\frac{1}{M}\mathbb{E}[\text{Reg}(\mathbf{w})] = \frac{1}{M}\mathbb{E}[R_1] + \frac{1}{M}\mathbb{E}[R_2]$$

$$\leq \sqrt{\xi_1}\ln(NT) + (UX+Y)^2\ln T + 17\frac{\ln(NT)}{\sqrt{\ln N}}\sqrt{\alpha T} +$$

$$66U\sqrt{\frac{CX^2}{4} + \frac{2\mu_2^2U^2X^4 + \frac{(2\mu_2+1)X^2C}{16}}{M}}\cdot\sqrt{T}.$$

Note that $C = O(UX+Y)^2$. Omitting the constant terms concludes the proof. $\qquad\square$

## G   PROOF OF THEOREM 4

We first prove a lower bound on the regret of OSLR, and then prove Theorem 4.

### G.1 LOWER BOUND OF ANY ALGORITHM FOR OSLR

Let $b = 1$ and $d > b' \geq b$. Assuming that the $h$-th attribute is the optimal one. We construct $(\mathbf{x}_t^h, y_t^h), t = 1, ..., T$ as follows.

$$\begin{cases} \forall i \neq h, x_{t,i}^h = 0, \\ x_{t,h}^h = y_t^h \cdot b_t, \\ \mathbb{P}[y_t^h = 1] = \mathbb{P}[y_t^h = 0] = \dfrac{1}{2}, \\ \mathbb{P}[b_t = 1] = \dfrac{1}{2} + \varepsilon, \quad \mathbb{P}[b_t = 0] = \dfrac{1}{2} - \varepsilon, \end{cases} \tag{15}$$

where $\varepsilon < \frac{1}{2}$. Let $\mathcal{A}$ be any algorithm that only uses one attribute to make prediction, i.e., $b = 1$, but can observe additional $b' - 1$ attributes. At each round $t$, denote by $I_t \in [d]$ the index of attributes selected by $\mathcal{A}$. The algorithm uses the constant predictor $w_{t,I_t}$ to give a prediction $w_{t,I_t} x_{t,I_t}^h$.

For any $i \in [d]$, let $l(w_{t,i}) = \left(w_{t,i} x_{t,i}^h - y_t^h\right)^2$ be the loss of $w_{t,i}$. By (15), it is easy to prove

$$\mathbb{P}[l(w_{t,i}) = 1] = \mathbb{P}[l(w_{t,i}) = 0] = \frac{1}{2}, \quad \forall i \neq h,$$
$$l(w_{t,h}) = (w_{t,h} b_t - 1)^2 \leq (w_h^* x_{t,h}^h - y_t^h)^2,$$

where $w_h^* = 1$. We can also prove

$$\mathbb{P}[(w_h^* x_{t,h}^h - y_t^h)^2 = 0] = \mathbb{P}[b_t = 1] = \frac{1}{2} + \varepsilon,$$
$$\mathbb{P}[(w_h^* x_{t,h}^h - y_t^h)^2 = 1] = \mathbb{P}[b_t = 0] = \frac{1}{2} - \varepsilon.$$

For any $w_{1,I_1}, w_{2,I_2}, \ldots, w_{T,I_T}$ generated by $\mathcal{A}$, it is easy to construct another predictors $\bar{w}_{1,I_1}, \bar{w}_{2,I_2}, \ldots, \bar{w}_{T,I_T}$ which are never worse than $\{w_{t,I_t}\}_{t=1}^T$. To be specific, we define $\bar{w}_{t,I_t}$ as follows,

$$\bar{w}_{t,I_t} = \begin{cases} w_{t,I_t} & \text{if } I_t \neq h, \\ w_h^* = 1 & \text{otherwise.} \end{cases} \tag{16}$$

It is easy to prove that

$$\begin{aligned} \max_{\mathbf{w} \in \mathcal{W}_1} \text{Reg}(\mathbf{w}) &= \sum_{t=1}^T (w_{t,I_t} x_{t,I_t}^h - y_t^h)^2 - \min_{\mathbf{w} \in \mathcal{W}_1} \sum_{t=1}^T (\mathbf{w}^\top \mathbf{x}_t^h - y_t^h)^2 \\ &\geq \sum_{t=1}^T (w_{t,I_t} x_{t,I_t}^h - y_t^h)^2 - \min_{w \in \mathbb{R}} \sum_{t=1}^T (w x_{t,h}^h - y_t^h)^2 \\ &\geq \sum_{t=1}^T (\bar{w}_{t,I_t} x_{t,I_t}^h - y_t^h)^2 - \sum_{t=1}^T (w_h^* x_{t,h}^h - y_t^h)^2. \end{aligned}$$

Thus we just need to prove the lower bound of regret induced by $\bar{w}_{1,I_1}, \bar{w}_{2,I_2}, \ldots, \bar{w}_{T,I_T}$. It is easy to show that

$$(\bar{w}_{t,I_t} \cdot x_{t,I_t}^h - y_t^h)^2 = y_t^2, \quad \forall I_t \neq h, \ \bar{w}_{t,I_t} \in \mathbb{R},$$
$$(\bar{w}_{t,I_t} \cdot x_{t,I_t}^h - y_t^h)^2 = (b_t - 1)^2, \quad I_t = h.$$

The loss is independent of the value of $\bar{w}_{t,I_t}$, but only depends on the index $I_t$. The algorithm can observe additional $b' - 1$ attributes, implying observing additional $b' - 1$ losses.

Next we reduce OSLR to a problem of prediction with limited advice (Seldin et al., 2014; Thune & Seldin, 2018). Let each coordinate $i \in [d]$ be an arm and $l_{t,i}$ be the loss induced by selecting the $i$-th arm, defined by

$$l_{t,i} = y_t^2 \in \{0, 1\}, \quad \forall i \neq h,$$
$$l_{t,h} = (b_t - 1)^2 \in \{0, 1\}.$$

It is obvious that

$$\mathbb{P}[l_{t,i} = 1] = \mathbb{P}[l_{t,i} = 0] = \frac{1}{2}, \quad \forall i \neq h,$$

$$\mathbb{P}[l_{t,h} = 0] = \frac{1}{2} + \varepsilon, \quad \mathbb{P}[l_{t,h} = 1] = \frac{1}{2} - \varepsilon.$$

$h$ is the unknown and optimal arm. At each round $t$, any algorithm selects an arm $I_t \in [d]$ and suffers the loss $l_{t,I_t}$. After that the algorithm selects additional $b' - 1$ arms, denoted by $I'_{t,1}, I'_{t,2}, ..., I'_{t,b'-1}$, and observe their losses.

We mainly follow the proof of Theorem 2 in Seldin et al. (2014). At each round $t$, let $O_t = \{I_t, I'_{t,1}, I'_{t,2}, ..., I'_{t,b'-1}\}$. Assuming that the algorithm is deterministic, that is, $O_t$ is determined by $\{O_\tau\}_{\tau=1}^{t-1}$ and the observed losses. To start with, we uniformly sample the best arm $h \in [d]$. Let $\mathbb{E}_h[\cdot]$ and $\mathbb{P}_h[\cdot]$ separately be the expectation and probability operator conditioned on the $h$-th attribute being the optimal one. Let $N_{T,i} = \sum_{t=1}^{T} \mathbb{I}_{I_t = i}$ and $\mathbf{q} = (q_1, q_2, ..., q_d)$ in which $q_i = \frac{N_{T,i}}{T}$. We random sample an arm $J \sim \mathbf{q}$. Then we have $\mathbb{P}_h[J = h] = \mathbb{E}_h[\frac{N_{T,i}}{T}]$. Under the condition of $h$ is the best arm, the expected regret satisfies

$$\mathbb{E}_h\left[\sum_{t=1}^{T}(\bar{w}_{t,I_t}x_{t,I_t}^h - y_t^h)^2 - \sum_{t=1}^{T}(w_h^* x_{t,h}^h - y_t^h)^2\right] = \mathbb{E}_h\left[\sum_{t=1}^{T}\varepsilon \cdot \mathbb{I}_{I_t \neq h}\right]$$

$$= \varepsilon \cdot (T - \mathbb{E}_h[N_{T,h}])$$

$$= \varepsilon T \cdot (1 - \mathbb{P}_h[J = h]).$$

Taking expectation w.r.t. $h$ yields

$$\frac{1}{d}\sum_{h=1}^{d}\mathbb{E}_h\left[\sum_{t=1}^{T}(\bar{w}_{t,I_t}x_{t,I_t}^h - y_t^h)^2 - \sum_{t=1}^{T}(w_h^* x_{t,h}^h - y_t^h)^2\right] \geq \varepsilon T \cdot \left(1 - \frac{1}{d}\sum_{h=1}^{d}\mathbb{P}_h[J = h]\right).$$

Next we aim to prove an upper bound on $\frac{1}{d}\sum_{h=1}^{d}\mathbb{P}_h[J = h]$.

We define a fictitious 0-th arm. If $h = 0$, then we define

$$\mathbb{P}[l_{t,i} = 1] = \mathbb{P}[l_{t,i} = 0] = \frac{1}{2}, \quad \forall i \in [d].$$

Let $\mathbb{P}_0[\cdot]$ and $\mathbb{E}_0[\cdot]$ separately be the expectation and probability operator conditioned on the 0-th arm being the optimal one. By Pinsker's inequality we have:

$$\frac{1}{d}\sum_{h=1}^{d}\mathbb{P}_h[J = h] \leq \frac{1}{d}\sum_{h=1}^{d}\mathbb{P}_0[J = h] + \frac{1}{d}\sum_{h=1}^{d}\sqrt{\frac{1}{2}\text{KL}(\mathbb{P}_0, \mathbb{P}_h)} \leq \frac{1}{d} + \sqrt{\frac{1}{2d}\sum_{h=1}^{d}\text{KL}(\mathbb{P}_0, \mathbb{P}_h)}.$$

By the proof of Theorem 2 in Seldin et al. (2014), we have

$$\sum_{h=1}^{d}\text{KL}(\mathbb{P}_0, \mathbb{P}_h) \leq \text{KL}(\frac{1}{2} - \varepsilon, \frac{1}{2}) \cdot b'T \leq 4\varepsilon^2 b'T.$$

Let $\varepsilon = \frac{\sqrt{d}}{4\sqrt{b'T}}$. We have

$$\varepsilon T \cdot \left(1 - \frac{1}{d}\sum_{h=1}^{d}\mathbb{P}_h[J = h]\right) \geq \varepsilon T \cdot \left(1 - \frac{1}{d} - \sqrt{\frac{1}{2d}\sum_{h=1}^{d}\text{KL}(\mathbb{P}_0, \mathbb{P}_h)}\right)$$

$$\geq \varepsilon T \cdot \left(1 - \frac{1}{d} - \sqrt{\frac{2\varepsilon^2}{d}b'T}\right)$$

$$\geq 0.09\sqrt{\frac{d}{b'}}T.$$

Then the worst-case regret of any deterministic algorithm for OSLR satisfies

$$
\sup_{(\mathbf{x}_t, y_t), t \in [T]} \max_{\mathbf{w} \in \mathcal{W}_1} \mathrm{Reg}(\mathbf{w}) \geq \sup_{h \in [d], (\mathbf{x}_t^h, y_t^h), t \in [T]} \max_{\mathbf{w} \in \mathcal{W}_1} \mathrm{Reg}(\mathbf{w})
$$

$$
\geq \sup_{h \in [d], (\mathbf{x}_t^h, y_t^h), t \in [T]} \left[ \sum_{t=1}^{T} (\bar{w}_{t,I_t} x_{t,I_t}^h - y_t^h)^2 - \sum_{t=1}^{T} (w_h^* x_{t,h}^h - y_t^h)^2 \right]
$$

$$
\geq \mathbb{E}_{h \in [d], (\mathbf{x}_t^h, y_t^h), t \in [T]} \left[ \sum_{t=1}^{T} (\bar{w}_{t,I_t} x_{t,I_t}^h - y_t^h)^2 - \sum_{t=1}^{T} (w_h^* x_{t,h}^h - y_t^h)^2 \right]
$$

$$
= \frac{1}{d} \sum_{h=1}^{d} \mathbb{E}_h \left[ \sum_{t=1}^{T} (\bar{w}_{t,I_t} x_{t,I_t}^h - y_t^h)^2 - \sum_{t=1}^{T} (w_h^* x_{t,h}^h - y_t^h)^2 \right]
$$

$$
\geq \varepsilon T \cdot \left( 1 - \frac{1}{d} \sum_{h=1}^{d} \mathbb{P}_h \left[ J = h \right] \right)
$$

$$
\geq 0.09 \sqrt{\frac{d}{b'} T}.
$$

### G.2 Lower Bound of Any Noncooperative Algorithm for OSLR-DecD

We uniformly sample a same $h \in [d]$ for all clients, and construct $(\mathbf{x}_t^{h,(j)}, y_t^{h,(j)})$ following (15). In this way, the optimal attribute is same for the data on all clients. Let $w_{1,I_1^{(j)}}^{(j)}, w_{2,I_2^{(j)}}^{(j)}, \ldots, w_{T,I_T^{(j)}}^{(j)}$ be the predictors generated by $\mathcal{A}$ on the $j$-th client. We also construct another predictors $\bar{w}_{1,I_1^{(j)}}^{(j)}, \bar{w}_{2,I_2^{(j)}}^{(j)}, \ldots, \bar{w}_{T,I_T^{(j)}}^{(j)}$ following (16), which are never worse than $\{w_{t,I_t^{(j)}}^{(j)}\}_{t=1}^{T}$. Following the proof in previous subsection, it is easy to prove that

$$
\sup_{\{(\mathbf{x}_t^{(j)}, y_t^{(j)})\}_{t=1}^{T}, j \in [M]} \max_{\mathbf{w} \in \mathcal{W}_1} \mathrm{Reg}(\mathbf{w})
$$

$$
= \sup_{\{(\mathbf{x}_t^{(j)}, y_t^{(j)})\}_{t=1}^{T}, j \in [M]} \left[ \sum_{j=1}^{M} \sum_{t=1}^{T} (w_{t,I_t^{(j)}}^{(j)} x_{t,I_t^{(j)}}^{(j)} - y_t^{(j)})^2 - \min_{\mathbf{w} \in \mathcal{W}_1} \sum_{j=1}^{M} \sum_{t=1}^{T} (\mathbf{w}^\top \mathbf{x}_t^{(j)} - y_t^{(j)})^2 \right]
$$

$$
\geq \sup_{\substack{h, \{(\mathbf{x}_t^{h,(j)}, y_t^{h,(j)})\}_{t=1}^{T}, \\ j \in [M]}} \left[ \sum_{j=1}^{M} \sum_{t=1}^{T} (w_{t,I_t^{(j)}}^{(j)} x_{t,I_t^{(j)}}^{h,(j)} - y_t^{h,(j)})^2 - \sum_{j=1}^{M} \sum_{t=1}^{T} (w_h^* \cdot x_{t,h}^{h,(j)} - y_t^{h,(j)})^2 \right]
$$

$$
\geq \mathbb{E}_{\substack{h, \{(\mathbf{x}_t^{h(j)}, y_t^{h(j)})\}_{t=1}^{T}, \\ j \in [M]}} \left[ \sum_{j=1}^{M} \sum_{t=1}^{T} (w_{t,I_t^{(j)}}^{(j)} x_{t,I_t^{(j)}}^{h,(j)} - y_t^{h,(j)})^2 - \sum_{j=1}^{M} \sum_{t=1}^{T} (w_h^* \cdot x_{t,h}^{h,(j)} - y_t^{h,(j)})^2 \right]
$$

$$
= \sum_{j=1}^{M} \frac{1}{d} \sum_{h=1}^{d} \mathbb{E}_h \left[ \sum_{t=1}^{T} (w_{t,I_t^{(j)}}^{(j)} x_{t,I_t^{(j)}}^{h,(j)} - y_t^{h,(j)})^2 - \sum_{t=1}^{T} (w_h^* \cdot \mathbf{x}_{t,h}^{h,(j)} - y_t^{h,(j)})^2 \right]
$$

$$
\geq 0.09 M \sqrt{\frac{d}{b'} T}.
$$

We can obtain a randomized algorithm by sampling from a class of deterministic algorithms. Thus the same lower bound holds for any algorithm, that is,

$$
\sup_{\{(\mathbf{x}_t^{(j)}, y_t^{(j)})\}_{t=1}^{T}, j \in [M]} \mathbb{E} \left[ \max_{\mathbf{w} \in \mathcal{W}_1} \mathrm{Reg}(\mathbf{w}) \right] \geq 0.09 M \sqrt{\frac{d}{b'} T},
$$

where the expectation is taken over the internal randomness of algorithm. Up to now, we conclude the proof.

Table 5: Experimental Results on Verifying the Efficiency of Estimators of $\mathbf{x}_t$.

| Algorithm | elevators ($b = 2$) | parkinson ($b = 2$) | bank ($b = 1$) |
| --- | --- | --- | --- |
| | MSE | MSE | MSE |
| AMRO-zero | $0.0134 \pm 0.0036$ | $0.0586 \pm 0.0004$ | $0.0235 \pm 0.0004$ |
| AMRO-random | $0.0136 \pm 0.0049$ | $0.0588 \pm 0.0013$ | $0.0241 \pm 0.0008$ |
| AMRO | $\mathbf{0.0089 \pm 0.0012}$ | $0.0584 \pm 0.0002$ | $0.0235 \pm 0.0007$ |

| Algorithm | cpusmall ($b = 2$) | calhousing ($b = 2$) | ailerons ($b = 1$) |
| --- | --- | --- | --- |
| | MSE | MSE | MSE |
| AMRO-zero | $0.0478 \pm 0.0068$ | $0.0470 \pm 0.0036$ | $\mathbf{0.0544 \pm 0.0128}$ |
| AMRO-random | $0.0532 \pm 0.0085$ | $0.0491 \pm 0.0088$ | $0.0621 \pm 0.0225$ |
| AMRO | $\mathbf{0.0282 \pm 0.0048}$ | $\mathbf{0.0360 \pm 0.0023}$ | $\mathbf{0.0535 \pm 0.0076}$ |

Table 6: Experimental Results on Verifying the Efficiency of $\delta^{(j)}$.

| Algorithm | elevators ($b = 2$) | parkinson ($b = 2$) | bank ($b = 1$) |
| --- | --- | --- | --- |
| | MSE | MSE | MSE |
| FedAMRO-0.9 | $0.0118 \pm 0.0010$ | $0.0655 \pm 0.0028$ | $0.0263 \pm 0.0005$ |
| FedAMRO-0.5 | $0.0125 \pm 0.0021$ | $0.0674 \pm 0.0055$ | $0.0263 \pm 0.0005$ |
| FedAMRO-0.1 | $0.0125 \pm 0.0021$ | $0.0679 \pm 0.0068$ | $0.0262 \pm 0.0004$ |

| Algorithm | cpusmall ($b = 2$) | calhousing ($b = 2$) | ailerons ($b = 1$) |
| --- | --- | --- | --- |
| | MSE | MSE | MSE |
| FedAMRO-0.9 | $0.0336 \pm 0.0029$ | $0.0447 \pm 0.0025$ | $0.0729 \pm 0.0038$ |
| FedAMRO-0.5 | $0.0333 \pm 0.0031$ | $0.0466 \pm 0.0054$ | $0.0724 \pm 0.0036$ |
| FedAMRO-0.1 | $0.0358 \pm 0.0024$ | $0.0483 \pm 0.0023$ | $0.0723 \pm 0.0045$ |

## H  MORE EXPERIMENTS

We verify that our estimators in (4) can induce better performance in certain benign environment. To be specific, we instantiate AMRO with three types of $\delta_{t,m}$, i.e.,

- AMRO: $\delta_{t,m}$ follows (9), i.e,

$$\delta_{t,m} = \sum_{\tau < t, m \in \tilde{B}_\tau \cup \hat{B}_\tau} \frac{x_{\tau,m}}{|\{\tau < t : m \in \tilde{B}_\tau \cup \hat{B}_\tau\}|}.$$

- AMRO-zero: $\delta_{t,m} = 0$

- AMRO-random: We first sample $\delta_{t,m} \in [-0.5, 0.5]$ for all $m = 1, .., d$. Then we normalize $\delta_t$ by

$$\delta_t \leftarrow \min\left\{1, \frac{4}{\|\delta_t\|_2}\right\} \cdot \delta_t.$$

The other hyper-parameters of AMRO-zero and AMRO-random follow AMRO.

The results are shown in Table 5. Overall, AMRO performs best on most of datasets, verifying that it is possible to estimate the value of each attribute using the observed values of each attribute. Especially, on the *elevators*, *cpusmall* and *calhousing* datasets, AMRO is much better than AMRO-zero and AMRO-random. On the *parkinson*, *bank* and *ailerons* datasets, the three algorithms enjoy similar prediction performance, showing that it is hard to precisely estimate the values of each attribute without any prior information. In other words, if we can obtain some prior information, it is possible to construct better estimator than the estimators in (9). The experimental results verify the superiority of our estimators in (4).

We further verify that how the values of $\sigma^{(j)}, j \in [M]$ influence the learning performance of FedAMRO. To be specific, we instantiate FedAMRO with three types of $\sigma^{(j)}$, i.e.,

- FedAMRO-0.9: $\sigma^{(j)} = 0.9$.
- FedAMRO-0.5: $\sigma^{(j)} = 0.5$.

- FedAMRO-0.1: $\sigma^{(j)} = 0.1$.

The other hyper-parameters of FedAMRO keep unchanged.

The results are shown in Table 6. Overall, as the value of increases, FedAMRO's prediction performance improves. Recalling that

$$\tilde{x}_{t,m}^{(j)} = \frac{x_{t,m}^{(j)} - \delta_{t,m}^{(j)}}{\delta^{(j)}} \cdot \mathbb{I}_{\tilde{\nu}_{t,m}^{(j)}} + \delta_{t,m}^{(j)}, \quad \forall m \in S_{I_t^{(j)}}.$$

as the value of increases, $\tilde{x}_{t,m}^{(j)}$ would contain more information of $x_{t,m}^{(j)}$, which may increase the risk of privacy leaking. Thus $\sigma_j$ balances the privacy and utility of FedAMRO.

