# OpenReview forum: "On the Power of Federated Learning for Online Sparse Linear Regression with Decentralized Data"
_ICLR.cc/2025/Conference — Submitted to ICLR 2025_

### Official Review · Reviewer_193r · 2024-10-29

**Soundness:** 3
**Presentation:** 3
**Contribution:** 3
**Rating:** 6
**Confidence:** 2

**Summary:**

This paper delves into the importance of federated learning (FL) in the context of online sparse linear regression (OSLR) with decentralized data (DecD). The authors challenge the previous consensus that FL is unnecessary for minimizing regret in full information settings by demonstrating that FL becomes essential when only limited attributes of each instance are observable.

The paper introduces the problem of online sparse linear regression with decentralized data (OSLR-DecD), where data is scattered across multiple clients, and each client can only observe a limited number of attributes for each instance. The authors propose a federated algorithm called FedAMRO for OSLR-DecD and establish a lower bound on the regret of any noncooperative algorithm. They prove that in scenarios where the dimensionality of data (d) is much smaller than the number of clients (M), the upper bound on the regret of FedAMRO is smaller than the lower bound for noncooperative algorithms, thus highlighting the necessity of FL.

**Strengths:**

1. An upper bound on the regret for FedAMRO, showing its effectiveness in OSLR-DecD.

2. A lower bound on the regret for any noncooperative algorithm, reinforcing the advantage of federated learning in OSLR-DecD when d is of smaller order than M.

3. The first lower and upper bounds on the regret for OSLR when M equals 1.

4. Three new techniques: any-time federated online mirror descent with negative entropy regularization, a client-server collaboration paradigm with privacy protection, and a reduction from online sparse linear regression to prediction with limited advice for establishing regret bounds.

**Weaknesses:**

1. The paper assumes that the dimensionality of data (d) is much smaller than the number of clients (M) to establish the necessity of federated learning. This assumption might not hold in real-world scenarios where the data can be high-dimensional or the number of clients limited. Whether this result can generalize to  high-dimensional sparse regression method ? such as lasso?

2. The experimental section of the paper validates the theoretical findings on a limited number of datasets. The generalization of the results to other types of data or different problem domains is not extensively tested. A more rigorous experimental validation involving a broader and more diverse set of datasets would strengthen the paper's conclusions.

**Questions:**

1. How does the proposed Federated Algorithm for Online Sparse Linear Regression with Decentralized Data (FedAMRO) scale with the increase in the number of clients (M) and the dimensionality of data (d), especially when both M and d are large? What are the potential bottlenecks, and how might they be addressed?

2. How robust is the proposed FedAMRO algorithm to data heterogeneity across clients?

Because I am not a professional researcher in the field of federated optimization, I find the conclusions of this article quite interesting. In the sparse case, it actually draws a contrary conclusion to the general setting, so I give it a score of 6.

---

> ### Author Response · Authors · 2024-11-22
>
> We thank the reviewer for detailed comments and valuable questions. The questions involve several valuable further work. Next we answer the questions.
>
> **[Question 1]** The paper assumes that $d$ is much smaller than the number of clients ($M$) to establish the necessity of federated learning. This assumption might not hold in real-world scenarios where the data can be high-dimensional or $M$ limited. Whether this result can generalize to high-dimensional sparse regression method? such as lasso?
>
> **[Answer]** We first clarify that the condition $d\ll M$ can be easily satisfied in the cross-device federated learning setting, in which the number of clients (such as phones, tablets) can easily go beyond millions (please refer to Section 1.2 in [1]). It is left as an important future work to study whether our algorithm can be extend to any value of $d$, that is, $d>1$. If not, the development of new algorithms is necessary.
>
> Next we answer the question. **We believe that algorithms based on high-dimensional sparse regression methods, like LASSO, could yield results better than ours**, given the advantageous properties of LASSO for sparse regression in high-dimensional spaces [2]. However, **these algorithms may require additional assumptions on the problem**, such as the restricted isometry condition [3], the restricted eigenvalue condition [4] an so forth. In conclusion, it is intriguing to establish new algorithms rooted in LASSO for OSLR-DecD.
>
> **[Question 2]** How does the proposed FedAMRO algorithm scale with the increase in $M$ and $d$, especially when both $M$ and $d$ are large? What are the potential bottlenecks, and how might they be addressed?
>
> **[Answer]** We will explain the scalability of FedAMRO from the statistical property (i.e., regret bound), computational complexity and communication complexity.
>
> First, we consider the statistical property. By Theorem 3, the regret bound of FedAMRO is
> $$
> \tilde{O}\left(M\frac{bd^2}{(b’-b)^2}\ln(d)+M\sqrt{bT\ln(d)}+\frac{d-b}{b’-b}\sqrt{bMT\ln(d)}\right),
> $$
> in which $\tilde{O}(\cdot)$ hides the polylogarithmic factors in $T$. The regret bound depends on $O(M\sqrt{T})$ and $O(d^2+d\sqrt{T})$. By Theorem 3.2 in [5], the dependence on $M$ is optimal. However, it is unclear whether the dependence on $d$ is optimal. **The bottleneck comes from that the variances on the estimators of loss and gradient are $O(d^2)$ in the worst case**. To address this issue, **we may consider some benign environments where the instances on all clients are predictable**, making it possible to construct estimators with variances much smaller than $O(d^2)$ (please see the remark below Lemma 1).
>
> Next we consider the computational and communication complexity. In line 111-113, the space complexity and per-round time complexity is $O(d^2)$ on each client and $O(Nd^b)$ on server. The communication complexity is $O(Md^2)$. The computational complexity on server is prohibitive. **The bottleneck comes from that the server maintains $O(d^b)$ models for each client to obtaining small regret bound**. However, it has been proved that there is not a $O(\mathrm{poly}(d))$ algorithm that can achieve a $o(T)$ regret for online sparse linear regression unless $\mathbf{NP}\subseteq \mathbf{BPP}$ [6]. Therefore, **to address this issue, we must pose more assumptions on the problem**, such as restricted isometry condition [7]. It is interesting to develop polynomial-time federated algorithms in the future.
>
> **[Question 3]** How robust is the proposed FedAMRO algorithm to data heterogeneity across clients?
>
> **[Answer]** Our results implicitly necessitate that the distributions across all client are similar, as the learning performance is evaluated by static regret (please see line 42-44), that is there is a single optimal model (i.e., ${\bf w}$) for all clients. **If the data heterogeneity is significant across clients, then we may use a new form of regret similar to the tracking regret in online learning [8] to evaluate the learning performance**, in which the optimal model on each client is different from others, i.e., ${\bf w}^{(j)}, j=1,...,M$. Studying the necessity of federated learning and developing algorithms in scenarios characterized by high data heterogeneity would be intriguing future work.
>
> **References**
>
> [1] Wang et al. A Field Guide to Federated Optimization. CoRR, 2021.
>
> [2] Raskutti et al. Restricted Eigenvalue Properties for Correlated Gaussian Designs. JMLR, 2010.
>
> [3] Emmanuel Candes and Terence Tao. Decoding by Linear Programming, TIT, 2005.
>
> [4] Bickel et al. Simultaneous analysis of lasso and dantzig selector. Annals of statistics, 2009.
>
> [5] Patel et al. Federated Online and Bandit Convex Optimization. ICML, 2023.
>
> [6] Foster et al. Online Sparse Linear Regression. COLT, 2016.
>
> [7] Kale et al. Adaptive Feature Selection: Computationally Efficient Online Sparse Linear
> Regression under RIP. ICML, 2017.
>
> [8] Daniely et al. Strongly Adaptive Online Learning. ICML, 2015.

---

### Official Review · Reviewer_8VCG · 2024-11-01

**Soundness:** 3
**Presentation:** 1
**Contribution:** 2
**Rating:** 5
**Confidence:** 3

**Summary:**

This paper addresses the challenge of online linear regression with partial feedback in a decentralized data environment (OSLR-DecD). The authors explore the role of federated learning in this context, proposing and proving the upper bound of the regret for a federated algorithm named FedAMRO for OSLR-DecD. They establish a regret lower bound for any noncooperative algorithm, showing that, in cases where $d=o(M)$, the regret upper bound of FedAMRO is smaller than this lower bound. This result underscores the effectiveness and importance of the federated learning approach within the decentralized, partial-feedback setting.

**Strengths:**

1. This paper addresses the problem of online linear regression with partial information feedback, which is a topic with broad applicability across real-world scenarios where data is decentralized and feedback is limited.

2. By proving both the upper bound of the regret under the federated setting and the lower bound of the regret under the noncooperative setting for the problem, the authors provide a thorough and well-rounded analysis, and answer the question of federated learning in this problem.

**Weaknesses:**

A primary weakness of this paper is the lack of clarity in conveying the intuition behind the core analytical ideas. The authors introduce numerous complex notations early on, often before providing a clear explanation of the algorithms’ key ideas, leaving many of the notations insufficiently defined. Additionally, the absence of a full algorithm description in the main text makes it challenging to grasp the algorithm’s foundational approach. The paper also lacks a proof outline or sketch, making it difficult to follow the reasoning behind the results. Sections 4 and 5 introduce extensive variable definitions for the pseudocode, yet both the algorithm and analysis remain inadequately explained. Furthermore, certain parts of the problem definition in the introduction and Sections 3.1 and 3.2 appear repetitive, occupying space that could be more effectively used to clarify the algorithm and its analysis.

**Questions:**

1. Could the authors give some intuition or high-level explanation of the algorithm design in Algorithms 2 and 3?
2. Could the authors explain how the core part of the analysis of Algorithm 2 is different from previous work?

---

> ### Author Response · Authors · 2024-11-22
>
> We sincerely thank the reviewer for constructive suggestions. We have revised our paper by incorporating more high-level explanations on our algorithms and theoretical analyses. Next we answer the questions.
>
> **[Question 1]** Could the authors give some intuition or high-level explanation of the algorithm design in Algorithms 2 and 3?
>
> **[Answer]** **We first give a high-level explanation on Algorithm 2**.  Recalling that we aim to learn the optimal parameter ${\bf w}^\ast$ satisfying $\Vert{\bf w}^\ast\Vert_0\leq b$. An intuitive approach is to partition $[d]$ into $N$ disjoint subsets denoted by $S_i,i\in[N]$, in which $\vert S_i\vert=b$ and each element in $S_i$ indexes the corresponding attributes of ${\bf x}_t$. In this work, our focus is on studying whether federated learning is effective for OSLR-DecD rather than developing computationally efficient algorithms. Then our algorithm simultaneously learns the support set of ${\bf w}^\ast$ and ${\bf w}^\ast$. To this end, our algorithm will learn a probability distribution over the $N$ subsets, and the optimal hypothesis $f^\ast_i$ parameterized by ${\bf w}^\ast_i$ with support set $S_i$, $i\in[N]$. It is obvious that ${\bf w}^\ast\in\{{\bf w}^\ast_1,\ldots,{\bf w}^\ast_N\}$.
>
> **Next we give a high-level explanation on Algorithm 3**. We proposed Algorithm 3 by extending upon Algorithm 2 within the federated learning framework. **The extension is non-trivial due to the tension of ensuring efficient communication and maintaining privacy protection.** We propose two techniques for addressing the challenge. The first one is decoupling prediction and updating. Specifically, clients make predictions, while server aggregates information from clients and updates probability distributions and hypotheses. Our algorithm independently samples a hypothesis for each client, and only sends the selected hypotheses to clients, thereby achieving efficient communication. The second one is a new paradigm for cooperating among clients and server. To be specific, we construct novel statistics (not estimators of loss and gradient) that will be sent by clients. Benefit from our estimators of ${\bf x}_t$  in Eq. (4), we can enhance the privacy by incorporating random noises into the estimators.
>
> **[Question 2]** Could the authors explain how the core part of the analysis of Algorithm 2 is different from previous work?
>
> **[Answer]**
> There are two core parts in the regret analysis of Algorithm 2 that are different from previous work [1,2].
>
> **The first one is to analyze the regret defined w.r.t. a sequence of time-variant competitors, i.e., $u_1 \in \Delta_{1,N}, u_2\in\Delta_{2,N},\ldots, u_T\in \Delta_{T,N}$ owing to the utilization of time-varying decision sets** $\Delta_{1,N},\ldots,\Delta_{T,N}$. To this end, we require a crucial lemma that carefully controls the sum of the difference of Bregman divergence (please see Lemma 8), i.e.,
> $$
> \sum^T_{t=1}[B_{\psi_t}( u_t,p_t) -B_{\psi_t}(u_t,p_{t+1})].
> $$
> The analysis differs from standard analysis of OMD with constant decision $\Delta_N$ and constant learning rate [1], which just controls a easier term defined as follows,
>
> $$
> \sum^T_{t=1}[B_{\psi}( u,p_t) -B_{\psi}(u,p_{t+1})]
> =B_{\psi}( u,p_1)-B_{\psi}( u,p_{T+1})\leq B_{\psi}( u,p_1).
> $$
>
> **The second core part in the regret analysis is to bound the variances of estimators of gradient and loss (please see Lemma 1)**, enabling our algorithm to improve the regret bound in [2] by a factor at least $O(d/b)$ and adapt to certain benign environments. We aim to prove that the variances depending on $\Vert {\bf x}_t -\delta_t\Vert^2_2$ where $\delta_t$ is an optimistic estimator of ${\bf x}_t$ using ${\bf x}_s$, $s=1,2,\ldots,t-1$. The variance bounds in Lemma 1 are data-dependent.
>
> **References**
>
> [1] Shai Shalev-Shwartz. Online Learning and Online Convex Optimization. Foundations and Trends in Machine Learning, 2011.
>
> [2] Foster et al. Online sparse linear regression. COLT, 2016.

---

### Official Review · Reviewer_MVgQ · 2024-11-03

**Soundness:** 2
**Presentation:** 3
**Contribution:** 2
**Rating:** 3
**Confidence:** 4

**Summary:**

This paper studies the problem of federated learning for online sparse linear regression (OSLR) with decentralized data. The authors first propose a new algorithm AMRO for OSLR, which improves the previous best regret, then extend AMRO to federated learning setting, and demonstrates the necessity of federated learning for OSLR by comparing the upper regret bound of FedAMRO and the lower regret bound of noncooperative algorithm.

**Strengths:**

1) This paper studies the problem of online linear regression with decentralized data, and proposes a federated algorithm with theoretical guarantees on its regret bound.
2) The lower bound on the regret of any noncooperative algorithm is also analyzed, which can demonstrate that federated learning is indeed necessary for the studied problem.
3) Experimental results are provided to verify the performance of the proposed algorithm.

**Weaknesses:**

1) In this paper, the definition of regret for federated learning seems a bit strange. I understand that it follows Patel et al., 2023. However, in decentralized learning, the regret is commonly defined as $Reg^{(i)}(\\mathbf{w})=\sum\_{j=1}^M \sum\_{t=1}^T [l(f_t^i(\\mathbf{x}_t^{j}),y\_t^{(j)}) - l(\\mathbf{w}^\top \\mathbf{x}\_t^{(j)}, y_t^{(j)})] $ [1][2][3]. Note that this definition considers the global loss of the learned hypothesis $f_t^i(\cdot)$ of the single client $i$. By contrast, the definition in this paper considers the sum of each local loss, i.e., $\sum\_{j=1}^M l(\\hat{y}\_t^{(j)},y\_t^{(j)})=\sum\_{j=1}^M l(f_t^j(\\mathbf{x}_t^{j}),y\_t^{(j)})$.

2) Moreover, I believe this strange definition of regret is the main reason for the pessimistic result of Patel et al., 2023 (i.e., the collaboration among clients is actually unnecessary in the full information setting). If we adopt the commonly used regret, the pessimistic result is no longer valid. In this case, the motivation of this paper may be questionable, since collaboration among clients is indeed necessary.

3) The literature review is poor. There do exist many related but missed studies, especially the mentioned works on decentralized online convex optimization [1][2][3] and the previous works on online sparse linear regression.

4) It is also unclear why the problem of online sparse linear regression with decentralized learning should be investigated.

[1] Yan et al. Distributed Autonomous Online Learning: Regrets and Intrinsic Privacy-Preserving Properties. IEEE Transactions on Knowledge and Data Engineering, 2013.

[2] Hosseini et al. Online distributed optimization via dual averaging. In CDC, 2013.

[3] Wan et al. "Nearly Optimal Regret for Decentralized Online Convex Optimization". In COLT, 2024.

**Questions:**

1) In this paper, the derived regret bound is related to $b$. Should different values of $b$ be tested on a dataset to verify its impact on loss?
2) When extending AMRO to FedAMRO, you modified $\\hat{\\mathbf{x}}$ and $\\widetilde{\\mathbf{x}}$ due to privacy constraints. Are there any theoretical results on privacy analysis?
3) Why does FedAMRO update $\\mathbf{w}\_{t+1,i}$ on the server? It seems that the communication costs can be reduced by updating it on the clients, calculating the cost, and then transmitting it to the server.

---

> ### Author Response · Authors · 2024-11-22
> **Addressing the reviewer’s concerns (Part 1)**
>
> We thank the reviewer for valuable comments and suggestions. We first try to address the reviewer’s concerns as follows and then answer the questions.
>
> **[Weakness 1]** In this paper, the definition of regret for federated learning seems a bit strange. I understand that it follows Patel et al., 2023. However, in decentralized learning, the regret is commonly defined as $Reg^{(i)}({\bf w})$. This definition considers the global loss of the learned hypothesis $f^i_t(\cdot)$ of the single client $i$. By contrast, the definition in this paper considers the sum of each local loss.
>
> **[Answer]** In this paper, the definition of regret for federated learning aligns closely with much of the previous work on federated online learning [1-4]. Even in distributed online learning, there is regret defined similarly to ours [5-7]. In fact, **our definition of regret for online learning with decentralized data (or federated online learning) is natural and necessary for two reasons.**
>
> (1) Due to the decentralized nature of data and the privacy constraint, it is infeasible to evaluate a model learned on client $i$, denoted by $f^i_t$, on the other clients, as the data from other clients are not visible for $f^i_t$. In this scenario, computing the cumulative losses $\sum^M_{j=1}\sum^T_{t=1}l(f^i_t({\bf x}^j_t),y^j_t)$ is unattainable.
>
> (2) In centralized online learning, the ultimate goal of a learner is to minimize the cumulative losses (or cumulative mistakes) suffered along its run (please see Page 109 in [8]). We introduce the notation of regret in the unrealizable case where the hypothesis space does not contain the true model (please see Page 110 in [8]) and restate the goal as minimizing the regret. A natural extension of the goal for a learner in online learning with decentralized data is to minimize her cumulative losses suffered along its run, i.e, $\sum^M_{j=1}\sum^T_{t=1}l(\hat{y}^j_t,y^j_t)$. In the unrealizable case, we can shift the objective to minimizing the regret, as defined in our paper.
>
> We agree with that minimizing $Reg^{(i)}({\bf w})$ is a standard goal in distributed online convex optimization. However, **due to the privacy constraint and the decentralized essence of data in online learning with decentralized data, it differs from distributed online convex optimization**. Therefore, the regret defined for these two problems should also differ.
>
> **[Weakness 2]** I believe this strange definition of regret is the main reason for the pessimistic result of Patel et al., 2023. If we adopt the commonly used regret, the pessimistic result is no longer valid. In this case, the motivation of this paper may be questionable, since collaboration among clients is indeed necessary.
>
> **[Answer]** The regret defined in this paper for online learning with decentralized data is rational (please see the answer on Weakness 1). Given the pessimistic result established in Patel et al., 2023, the motivation behind this paper is substantial, exploring the conditions under which federated learning is necessary for online linear regression with decentralized data. Both the groundbreaking findings of Petel et al., 2023 and our results provide new insights on the limitations and power of federated learning.

---

> ### Author Response · Authors · 2024-11-22
> **Addressing the reviewer’s concerns (Part 2)**
>
> **[Weakness 3]** The literature review is poor. There do exist many related but missed studies, especially the mentioned works on decentralized online convex optimization and the previous works on online sparse linear regression.
>
> **[Answer]** First, we need to clarify that **the notation of “decentralized” in decentralized online convex optimization emphasizes there is not a centralized server, while it emphasizes the data is generated from geographically dispersed edge devices in online learning with decentralized data** (please see line 30-40). We have focused on the work most related to ours and omitted the less relevant studies due to space limitations, such as distributed online convex optimization. We have discussed the distinctions between our work and distributed online convex optimization, especially regarding the definition of regret, in our revised version.
>
> Furthermore, we need to clarify that the relevant studies on online sparse linear regression (OSLR) were discussed in line 264-269. We also compared the regret bound of our algorithm with those of prior algorithms for OSLR in Table 1. These comparisons suffice to underscore the contributions of our work to OSLR.
>
> **[Weakness 4]** It is also unclear why the problem of online sparse linear regression with decentralized learning should be investigated.
>
> **[Answer]** As mentioned in the answers to Weakness 1 and Weakness 2, **our primary goal is to explore the conditions under which the pessimistic result established by Patel et al. (2023) can be circumvented**. Their findings motivated us to investigate online linear regression in a decentralized data setting with limited information, where algorithms have access only to a subset of attributes for each instance. We refer to this new problem as **online sparse linear regression with decentralized data (OSLR-DecD)**.
>
> As explained in the Introduction, real-world applications often face constraints that prevent algorithms from observing all attributes of an instance, leading to degraded prediction performance. Previous work has formulated online learning problems with limited attribute access as Online Sparse Linear Regression (OSLR). Additionally, data is frequently generated from geographically dispersed edge devices, naturally leading to a decentralized data scenario. Therefore, we aim to **determine whether the prediction performance on a client can be enhanced by leveraging data from multiple clients, which motivates our study of OSLR-DecD**.
>
> In conclusion, OSLR-DecD is a new, well-defined, and challenging problem that is valuable for understanding both the limitations and capabilities of federated learning.
>
> We hope that our clarifications can address the reviewer’s concerns. We are also happy to offer more explanations if needed.
>
> **References**
>
> [1] Abhimanyu Dubey and Alex Pentland. Differentially-Private Federated Linear Bandits. NeurIPS, 2020.
>
> [2] Li et al. Communication Efficient Federated Learning for Generalized Linear Bandits. NeurIPS, 2022.
>
> [3] Huang et al. Federated Linear Contextual Bandits. NeurIPS, 2021.
>
> [4] Olusola T. Odeyomi. Differentially Private Online Federated Learning With Personalization and Fairness. ISIT, 2023.
>
> [5] Kamp et al. Communication-Efficient Distributed Online Prediction by Dynamic Model Synchronization. ECML-PKDD, 2014.
>
> [6] Kamp et al. Communication-Efficient Distributed Online Learning with Kernels. ECML-PKDD, 2016.
>
> [7] Wang et al. Distributed Bandit Learning: Near-Optimal Regret with Efficient Communication. ICLR, 2020.
>
> [8] Shai Shalev-Shwartz. Online Learning and Online Convex Optimization. Foundations and Trends in Machine Learning, 2011.

---

> ### Author Response · Authors · 2024-11-22
> **Answering the reviewer’s questions**
>
> Next we answer the questions.
>
> **[Question 1]** In this paper, the derived regret bound is related to $b$. Should different values of $b$ be tested on a dataset to verify its impact on loss?
>
> **[Answer]** We did not verify different values of $b$ on a single dataset. The experiments aim to verify the following two goals: (1) Our algorithm, AMRO, enjoys better prediction performance than all of previous algorithms for OSLR. (2) In the case of $M = \omega(d)$, federated learning is necessary for OSLR-DecD. Since our theoretical results hold for any $b\geq 1$, it suffices to set a fixed value of $b$ on a dataset.
>
> **[Question 2]** When extending AMRO to FedAMRO, you modified $\hat{\bf x}_t$ and $\tilde{\bf x}_t$ due to privacy constraints. Are there any theoretical results on privacy analysis?
>
> **[Answer]** We did not provide theoretical results on privacy. It would be intriguing to analyze our algorithm or develop new algorithms with rigorous privacy guarantee using the differential private framework. In fact, our algorithm can enhance privacy empirically by introducing Gaussian noises into $\hat{\bf x}_t$ and $\tilde{\bf x}_t$, making it hard to recovery $({\bf x}^{(j)}_t,y^{(j)}_t)$ from the transmitted information
> $y^{(j)}_t\cdot (\hat{\bf x}^{(j)}_t+\tilde{\bf x}^{(j)}_t)$, aligning with the principles of federated learning. By the concentration inequality of Gaussian random variable, it is easy to prove that our regret bounds only increase by a factor of $O(\mathrm{poly}(\ln{T}))$.
>
> **[Question 3]** Why does FedAMRO update $w_{t+1,i}$ on the server? It seems that the communication costs can be reduced by updating it on the clients, calculating the cost, and then transmitting it to the server.
>
> **[Answer]** In line 297-309, we elucidated that it is non-trivial to give a federated version of AMRO, involving the need for efficient communication and preserving privacy. To be specific, **if we update $w_{t+1,i}$ and calculate the loss $c^{(j)}_{t,i}$ for all $i=1,...,N$ on clients, then the computational cost on each client is $O(N)$ and the total communication cost is $O(MN)$, in which $N=O(d^b)$ in general**. To address this issue, our algorithm updates $w_{t+1,i}$ and calculates $c^{(j)}_{t,i}$ on server, resulting in a computational cost of only $O(d^2)$ on each client and a total communication cost of only $O(Md^2)$.

---

> > ### Comment · Reviewer_MVgQ · 2024-11-28
> >
> > Thanks for the authors' responses. I do not agree with the authors' comparison of the decentralized online optimization (studied in this paper) and the distributed online convex optimization (studied in previous studies). Although each local client in the decentralized setting cannot directly access the model and data of other clients, the goal of the decentralized optimization should still be to minimize the global loss functions by the model in each local client (as what the so-called distributed online convex optimization has done). If we only need to compare the model of each local client against the optimal model for local data in this client (and then simply sum the local regret among all clients), one actually can prove that the problem is trivially equal to individually performing the standard online convex optimization on each local client. That actually is why the previous work of Patel et al., 2023 can derive the so-called pessimistic result (i.e., the collaboration among clients is unnecessary). Moreover, one actually can read these previous papers on distributed online convex optimization, and find that their algorithms are actually designed in a totally decentralized way.

---

> > > ### Author Response · Authors · 2024-11-28
> > >
> > > We sincerely appreciate the Reviewer’s feedback on our responses.
> > >
> > > We have carefully read the feedback and identified that the disagreement comes from the goals of online learning with decentralized data (OL-DecD) and distributed online convex optimization (DOCO). The two goals are related, but essentially different. We will provide further explanation as follows.
> > >
> > > For DOCO, following previous work (please see Eq. (4) in [1] and Eq. (2) in [2]), the learner (or algorithm) aims to solve the following problem on the fly,
> > > $$
> > > P_1: \min_{w}\sum^T_{t=1}\sum^M_{i=1}f_{t,i}(w),
> > > $$
> > > where $f_{t,i}$ is a convex loss function. We denote by $L^\ast$ the minimum and $w^\ast$ the optimal solution. **In other words, the learner aims to output a $\hat{w}_i$ that is a good approximation of $w^\ast$ on each client $i$. Besides, the cumulative losses of $\hat{w}_i$ also approximates $L^\ast$ well.** Since the learner must approximate $w^\ast$, the collaboration is necessary. However, this differs for OL-DecD.
> > >
> > > For OL-DecD (or online learning), **the learner aims to minimize the cumulative losses suffered along its run** (please see Page 109 in [3] or see the answer on Weakness 1), that is,
> > > $$
> > > P_2:
> > > \min_{w_{t,i}, t=1,...,T, i=1,...,M}\sum^T_{t=1}\sum^M_{i=1}f_{t,i}(w_{t,i}).
> > > $$
> > > By the definition of regret, we can rewrite the goal as approximating $L^\ast$, i.e., the minimum of $P_1$, rather than producing an approximation of $w^\ast$. **In other words, the learner also aims to solve $P_1$, but is not required to output a $\hat{w}$**.
> > >
> > > Therefore, OL-DecD is related to DOCO but fundamentally distinct. As pointed by the Reviewer, collaboration is unnecessary for OL-DecD in full information setting. However, we proved that collaboration is indeed necessary in partial information setting, such as, online sparse linear regression with decentralized data.
> > >
> > > We hope that our responses can address the Reviewer’s concerns. We are happy to offer more explanations if needed.
> > >
> > > References
> > >
> > > [1] Yan et al. Distributed Autonomous Online Learning: Regrets and Intrinsic Privacy-Preserving Properties. IEEE Transactions on Knowledge and Data Engineering, 2013.
> > >
> > > [2] Hosseini et al. Online distributed optimization via dual averaging. In CDC, 2013.
> > >
> > > [3] Shai Shalev-Shwartz. Online Learning and Online Convex Optimization. Foundations and Trends in Machine Learning, 2011.

---

### Official Review · Reviewer_VNSb · 2024-11-04

**Soundness:** 3
**Presentation:** 3
**Contribution:** 3
**Rating:** 8
**Confidence:** 3

**Summary:**

This paper considers the federated online sparse linear regression problem, where each client can only observe part of the feature vector at each round. The authors propose the Aggregated OMD algorithm and establish the regret bound. They also establish the lower bound for non-cooperative algorithms. Their results reveal an interesting fact: unlike the full-information setting, where non-cooperative algorithms are optimal, federated learning can achieve better performance than non-cooperative algorithms.

**Strengths:**

1. The problem is well-motivated, and the results provide an interesting contrast to federated full-information online linear regression.

2. Technically solid: improvements on regrets of OSLR and a new reduction from sparse linear regression to online prediction with limited advice.

3. Numerical experiments are conducted to illustrate the superior performance of the proposed algorithm.

**Weaknesses:**

1. There is still an $O(\sqrt{d})$ gap between the proposed upper bound and the lower bound.

2. Several explanations in the algorithm design are unclear, as mentioned in the questions section.

**Questions:**

1. In line 265, the authors state that subtracting $y_t^2$ from the loss can enhance privacy protection. However, it seems that the reported information still involves $y_t$. Could the authors explain this point further?

2. In line 282, the authors mention that there are many approaches to define $\delta_{t,m}$. Could the authors clarify which general properties $\delta_{t,m}$ must satisfy to achieve the same regret guarantee in the main theorem?

3. The authors state that their operations in equations (3) and (4) differ from previous algorithm designs (subtracting $y_t^2$ or using non-zero $\delta_{t,m}$). Are these differences intended for privacy protection or to improve regret performance?

---

> ### Author Response · Authors · 2024-11-22
>
> We sincerely thank the Reviewer for offering valuable comments and acknowledging our theoretical and technical contributions. Next we answer the questions.
>
> **[Question 1]** In line 265, the authors state that subtracting $y^2_t$ from the loss can enhance privacy protection. However, it seems that the reported information still involves. Could the authors explain this point further?
>
> **[Answer]** In line 332, our algorithm transmits $y^{(j)}_t\cdot (\hat{\bf x}^{(j)}_t+\tilde{\bf x}^{(j)}_t)$ which contains $y^{(j)}_t$. However, **it is hard to reconstruct $y^{(j)}_t$ from $y^{(j)}_t\cdot (\hat{\bf x}^{(j)}_t+\tilde{\bf x}^{(j)}_t)$, since our algorithm does not transmit $(\hat{\bf x}^{(j)}_t+\tilde{\bf x}^{(j)}_t)$.** Conversely, if $(y^{(j)}_t)^2$ is not subtracted  from the loss, then our algorithm must transmit $(y^{(j)}_t)^2$ to server to compute the loss estimator. It is easy to recover $y^{(j)}_t$ from the reported information $(y^{(j)}_t)^2$.
>
> Therefore, subtracting $y^2_t$ from the loss can enhance privacy protection.
>
> **[Question 2]** In line 282, the authors mention that there are many approaches to define $\delta_{t,m}$. Could the authors clarify which general properties $\delta_{t,m}$ must satisfy to achieve the same regret guarantee in the main theorem?
>
>
> **[Answer]** As indicated in Eq. (8), we only need to ensure that the norm of $\delta_t$ is bounded, i.e., $\Vert \delta_t\Vert_2$ is bounded. This condition is readily met, as we can always normalize $\delta_t$.
>
> **[Question 3]** The authors state that their operations in equations (3) and (4) differ from previous algorithm designs (subtracting $y^2_t$ or using $\delta_{t,m}\neq 0$). Are these differences intended for privacy protection or to improve regret performance?
>
> **[Answer]** As previously mentioned in line 256, **subtracting $y^2_t$ from the loss can enhance privacy protection** (or please see the answer on Question 1). As previously mentioned in line 269 and line 286, $\delta_{t,m}\neq 0$ can be use to protect the privacy or improve the regret. To be specific, **we set $\delta_{t,m}\neq 0$ in the AMRO algorithm for improving the regret bound**, while **we redefine $\delta_{t,m}\neq 0$, such as random noises, in the FedAMRO algorithm for privacy protection.**

---

### Author Response · Authors · 2024-11-22
**Response to All Reviewers**

We thank the reviewers for taking the time to review our paper and offering valuable comments and constructive suggestions. **We have revised the manuscript and marked the revised parts suggested by Reviewers in red.** To be specific, our revised parts contain:

(1) We gave more explanations on “why subtracting $y^2_t$ from the loss can enhance privacy” posed by Reviewer VNSb.

(2) We simplified the problem definition, provided high-level explanations on Algorithm 2 and Algorithm 3, and explained the differences between the analysis of Algorithm 2 and previous work (Appendix E due to space limitations), as recommended by Reviewer 8VCG.

(3) We provided a comparison between our problem and a related yet fundamentally different problem called distributed online convex optimization, as suggested by Reviewer MVgQ, **thereby addressing the crucial concerns raised by the Reviewer regarding our work**.

The reviewer MVgQ raised several concerns regarding the definition of regret and the motivation behind this paper. These two concerns might stem from the absence of explanations on the differences between online learning with decentralized data and distributed online convex optimization in our manuscript. We have provided comprehensive explanations addressing the concerns.

---

### Author Response · Authors · 2024-12-02
**Response to Area Chair and All Reviewers**

Dear Area Chair and Reviewers,

As there is still a disagreement between us and Reviewer MVgQ, we kindly ask you to concentrate more on the concerns raised by the Reviewer and our responses, during the Reviewers’ discussion period. For clarity, we summarize the main concerns and our responses as follows.

The Reviewer believes that the definition of regret in this paper is not reasonable and should follow the definition in distributed online convex optimization (DOCO) [1,2,3]. In this case, the motivation of this work is questionable. The regret in this paper is defined by
$$
\forall w\in W,\quad
Reg(w)=\sum^T_{t=1}\sum^M_{j=1}[\ell(\langle w^j_t,x^j_t\rangle,y^j_t)
-\ell(\langle w,x^j_t\rangle,y^j_t)].
$$
The regret in DOCO is defined by
$$
\forall i\in[M], \forall w\in W,
Reg^i(w)=\sum^T_{t=1}\sum^M_{j=1}[\ell(\langle w^i_t,x^j_t\rangle,y^j_t)
-\ell(\langle w,x^j_t\rangle,y^j_t)].
$$
DOCO requires that the models on each client $i$ maintain a small regret for data across all clients. **Both of the two definitions are valid, as the objectives of DOCO and our specific focus, online learning with decentralized data (also known as federated online learning), are distinct**. **The regret defined in this paper aligns with the previous work on federated online learning [4-7] and distributed online learning [8-9]**. Next we explain why our definition is valid.

**(1)** Due to the privacy constraint, it is infeasible to evaluate a model learned on client $i$, denoted by $w^i_t$, on the other clients, as the data from other clients are not visible for $w^i_t$. In this scenario, computing $\sum^M_{j=1}\ell(\langle w^i_t,x^j_t\rangle,y^j_t)$ is unattainable. **The Reviewer agrees with the privacy constraint but still believes we should follow the defintion in DOCO. This argument insufficient to justify rejecting our definition.**

**(2)** The ultimate goal of online learning algorithms is to minimize the cumulative losses suffered along its run (please see Page 109 in [10]). A natural extension of the goal to online learning with decentralized data (OL-DecD) is to minimize the following cumulative losses
$$
\sum^T_{t=1}\sum^M_{j=1}\ell(\langle w^j_t,x^j_t\rangle,y^j_t).
$$
We introduce the notation of regret in the unrealizable case (please see Page 110 in [10]) and restate the goal as minimizing the regret.

**(3)** The goals of DOCO and OL-DecD are different**.

In DOCO (please see Eq. (4) in [1], and Eq. (2) in [2]), the goal is to solve $P_1$,
$$
P_1: \min_w\sum^T_{t=1}\sum^M_{j=1}\ell(\langle w, x^j_t\rangle, y^j_t).
$$
We denote by $L^\ast$ the minimum and $w^\ast$ the optimal solution. Algorithms for DOCO aim to output a $w^i$ that is a good approximation of $w^\ast$ and approximate $L^\ast$.

**For OL-DecD, by the definition of regret, we can rewrite the goal as approximating $L^\ast$, rather than producing an approximation of $w^\ast$.**

**Note that the difference is non-trivial. The collaboration among clients is necessary in DOCO, as we must approximate $w^\ast$. Otherwise, it is not necessary**. For example, we can independently run online gradient descent on each client, and obtain an approximation of $L^\ast$, i.e.,
$$
\sum^T_{t=1}\sum^M_{j=1}\ell(\langle w^j_t,x^j_t\rangle,y^j_t)
=L^\ast+O(M\sqrt{T}).
$$
In contrast, the optimal algorithm for DOCO [3] obtains a worse approximation of $L^\ast$, i.e.,
$$
\forall i\in [M],\quad
\sum^T_{t=1}\sum^M_{j=1}\ell(\langle w^i_t,x^j_t\rangle,y^j_t)
=L^\ast+O(\rho^{-0.25}M\sqrt{T\ln{T}}), ~\rho <1.
$$

It has been proved that collaboration is unnecessary for OL-DecD in full information setting. In this work, **we aim to explore whether it is necessary in partial information setting, such as, online sparse linear regression with decentralized data (OSLR-DecD). We also answer this question affirmatively**. **OSLR-DecD is a new, well-defined, and challenging problem and is valuable for understanding both the limitations and capabilities of federated learning.**


Thanks.

**Reference**

[1] Yan et al. Distributed Autonomous Online Learning: Regrets and Intrinsic Privacy-Preserving Properties. TKDE, 2013.

[2] Hosseini et al. Online distributed optimization via dual averaging. CDC, 2013.

[3] Wan et al. "Nearly Optimal Regret for Decentralized Online Convex Optimization". COLT, 2024.

[4] Kwon et al. Tighter Regret Analysis and Optimization of Online Federated Learning. TPAMI, 2023.

[5] Dubey et al. Differentially-Private Federated Linear Bandits. NeurIPS, 2020.

[6] Li et al. Communication Efficient Federated Learning for Generalized Linear Bandits. NeurIPS, 2022.

[7] Huang et al. Federated Linear Contextual Bandits. NeurIPS, 2021.

[8] Wang et al. Distributed Bandit Learning: Near-Optimal Regret with Efficient Communication. ICLR, 2020.

[9] Kamp et al. Communication-Efficient Distributed Online Prediction by Dynamic Model Synchronization. ECML, 2014.

[10] Shai Shalev-Shwartz. Online Learning and Online Convex Optimization. FTML, 2011.

---

### Meta-Review · Area_Chair_vc77 · 2024-12-21

**Metareview:**

This paper addresses the problem of online sparse linear regression with distributed data, proposing an algorithm along with regret upper and lower bounds. The main concerns with this paper are as follows: the design and analysis of the algorithm lack sufficient intuitive explanation, and the necessity of federated learning is demonstrated under a rather restrictive condition ($d \ll M$). Furthermore, there are concerns about the accuracy of the content, as detailed below. To meet the standards expected of papers presented at ICLR, these issues need to be addressed appropriately.

1. **Lines 187–194**: The paper states that $S_i$ is a disjoint subset, which can be interpreted as $i \neq i' \implies S_i \cap S_{i'} = \emptyset$. However, under this condition, the optimal subset is not necessarily included in $\\{S_i\\}$.
2. **Table 1**: Are the labels for Algo2 and Algo3 correct? Upon reviewing the referenced literature, I could not find regret upper bounds matching those described in the table.
3. **Lemma 6**: The statement seems to omit the assumption $\beta_{t} > \beta_{t+1}$. Additionally, I could not verify the correctness of part (b). Specifically,
$$
   \frac{1 - \frac{N - |A_t|}{N}\beta_{t+1}}{\eta_{t+1}} - \frac{1 - \frac{N - |A_t|}{N}\beta_{t}}{\eta_{t}} = \frac{1}{\eta_{t+1}} - \frac{1}{\eta_{t}} + \frac{N - |A_t|}{N} \left(\frac{\beta_{t}}{\eta_{t}} - \frac{\beta_{t+1}}{\eta_{t+1}}\right),
$$
   which does not always seem to be less than $(\frac{1}{\eta_{t+1}} - \frac{1}{\eta_{t}})$.
4. **Binomial coefficients**: The notation $\binom{b}{d}$ appears in multiple locations, but it seems to be a typographical error for $\binom{d}{b}$.

Moreover, while the paper invests significant effort in analyzing OMD with a time-variant learning rate, the analysis could potentially be greatly simplified by adopting an FTRL-type update instead. For instance, referring to Corollary 7.9 and Section 7.6 in [Orabona, Francesco. "A modern introduction to online learning." arXiv preprint arXiv:1912.13213 (2019).] is recommended.

**Additional Comments On Reviewer Discussion:**

Concerns have been raised regarding the lack of intuitive explanations for the problem setting, algorithm design, and analysis. While the problem setting—particularly the definition of regret—is argued to be natural and consistent with prior work, there seems to be a lack of specific and intuitive explanations about the practical applications where this definition is appropriate and the motivations that justify its use.

---

### Decision · Program_Chairs · 2025-01-22

Reject